# Ultrack: pushing the limits of cell tracking across biological scales

Jordão Bragantini [1] ✉, Ilan Theodoro [1,2], Xiang Zhao [1], Teun A.P.M. Huijben [1], Eduardo Hirata-Miyasaki[1], Shruthi VijayKumar[1], Akilandeswari Balasubramanian[1], Tiger Lao[1], Richa Agrawal[3], Sheng Xiao [1], Jan Lammerding [3,4], Shalin Mehta [1], Alexandre X. Falcão [2], Adrian Jacobo [1], Merlin Lange [1] & Loïc A. Royer [1] ✉

Tracking live cells across two-dimensional, three-dimensional (3D) and multichannel time-lapse recordings is crucial for understanding tissue-scale biological processes. Despite advancements in imaging technology, accurately tracking cells remains challenging, particularly in complex and crowded tissues where cell segmentation is often ambiguous. We present Ultrack, a versatile and scalable cell tracking method that tackles this challenge by considering candidate segmentations derived from multiple algorithms and parameter sets. Ultrack leverages temporal consistency to select optimal segments, ensuring robust performance even under segmentation uncertainty. We validate our method on diverse datasets, including terabyte-scale developmental time-lapse recordings of zebrafish, fruit fly and nematode embryos, as well as multicolor and label-free cellular imaging. We demonstrate that Ultrack achieves superior or comparable performance in the cell tracking challenge, particularly when tracking densely packed 3D embryonic cells over extended periods. Moreover, we propose an approach to tracking validation via dual-channel sparse labeling that enables high-fidelity ground-truth generation, pushing the boundaries of long-term cell tracking assessment. Our method is freely available as a Python package with Fiji and Napari plugins and can be deployed in a high-performance computing environment, facilitating widespread adoption by the research community.

Advancements in live imaging technologies[1–7] have enabled the visualization of cellular dynamics with unprecedented spatiotemporal resolution[8] over large fields of view[6,7] encompassing thousands of cells[5,9], producing vast amounts of multidimensional data[10]. However, one key challenge remains: the accurate reconstruction of cell trajectories and lineages in complex biological systems[11]. This ability unlocks deeper insights into cellular state and behavior[9,12,13], tissue mechanics[14], morphogenesis[2,15,16] and regeneration[17].

Cell segmentation and tracking have been persistent challenges in bioimage analysis[11]. While cell segmentation has advanced rapidly with the advent of deep-learning-based methods[18–20], cell tracking continues to be an open problem[11]. The primary challenge stems from the association of spurious segmentations over time, hindering long-term tracking, especially in dense and dynamic cellular environments. Most automatic tracking methods use a two-step approach: first segmenting cells, then linking them across time frames[21–24]. While computationally

[1]Chan Zuckerberg Biohub, San Francisco, CA, USA. [2]Institute of Computing - State University of Campinas, Campinas, Brazil. [3]Weill Institute for Cell and Molecular Biology - Cornell University, Ithaca, NY, USA. [4]Meinig School of Biomedical Engineering - Cornell University, Ithaca, NY, USA. ✉e-mail: jordao.bragantini@czbiohub.org; loic.royer@czbiohub.org

efficient, this approach struggles with the compounding of errors over time, particularly in dense tissues[25] or when cells divide rapidly.

Simultaneous segmentation and tracking offers a promising alternative[26–30], wherein multiple candidate segments and/or inter-segment links are generated, and then optimal candidates are selected by satisfying biological constraints (that is, cells divide but do not merge) and optimizing specific criteria (that is, only keep links best supported by the data). However, existing methods in this category are constrained by their inability to utilize arbitrary segmentation inputs[28,31], as well as limitations in data scale or dimensionality due to high computational costs[29,30]. These factors have limited their applicability to tissue-scale problems.

To address these challenges, we developed Ultrack, a robust and scalable method for large-scale cell tracking that excels even under segmentation uncertainty. Ultrack can track cells (or nuclei) in two-dimensional (2D), three-dimensional (3D) and multichannel datasets, accommodating a wide range of biological contexts. Ultrack seamlessly integrates with various segmentation algorithms, including state-of-the-art deep-learning-based cell segmentation tools[18–20,32–34] (Fig. 1a). By jointly evaluating candidate segmentations and tracks, Ultrack uses temporal consistency to select the most accurate segments (Fig. 1b). This approach leverages information from adjacent time points to resolve cell segmentation and tracking ambiguities, enhancing performance, particularly in complex, densely packed tissues where cell boundaries are often uncertain (Supplementary Video 1).

We demonstrate the utility and robustness of Ultrack through a comprehensive set of benchmarks and real-world applications, including top-scoring performances on the cell tracking challenge (CTC)[11]. Our method shows improved tracking performance across various use cases and scales, from multicolor and label-free 2D datasets to large-scale 3D time-lapse sequences of developing embryos. Additionally, we introduce a new dual-channel sparse labeling approach that enables the generation of high-fidelity ground truth for tracking validation. This method combines ubiquitous fluorescence labeling with sparse, random labeling at a distinct wavelength, allowing for the creation of annotations that would be impossible to produce manually. This innovative validation strategy pushes the limits of cell tracking assessment, enabling the evaluation of tracking performance over extended time periods and in increasingly complex tissue environments.

To ensure broad accessibility and ease of use, we provide intuitive interfaces in Fiji[35], Napari[36], Python and command line, catering to researchers with diverse computational backgrounds. We also made efforts to ensure interoperability with existing popular cell tracking frameworks such as TrackMate[37] and Mastodon[38] (Fig. 1b). Thus, by enabling more accurate, scalable and accessible cell tracking, Ultrack can accelerate research in developmental biology, cancer research and regenerative medicine, offering new insights into cellular behavior and tissue dynamics.

## Results

### Avoiding premature mistakes with segmentation hypotheses
Cell tracking accuracy is compromised when cell segmentation (or detection) fails. To address this challenge, Ultrack considers multiple segment hypotheses instead of prematurely committing to a particular choice of segments (Fig. 1a). These hypotheses can be obtained from the original imaging data pixel intensities, multiple algorithms (for example, watershed plus Cellpose) or from one or several algorithms configured with different parameters. This approach allows the integration of various segmentation methods, maintaining compatibility with the growing array of novel off-the-shelf segmentation models[39,40]. However, the main challenge of this approach and previous joint segmentation and tracking methods, as well as the main obstacle for their practical application, is the potentially large number of possible segmentation hypotheses, especially for large multidimensional datasets.

### Efficient representation of segmentation hypotheses
To address the challenge of managing multiple segmentation hypotheses, we use ultrametric contour maps (UCMs)[41–43], which provide a compact representation of these hypotheses. Intuitively, UCMs are multilevel contour maps that represent a hierarchy of possible cell boundaries, where the strength of each boundary reflects its likelihood of being a true cell edge (Fig. 1c). This hierarchical structure allows for the efficient encoding of multiple possible segmentations within a single map. More formally, UCMs partition the image space into nested segments, where valid segments do not overlap, or are contained within another. This property enables Ultrack to efficiently consider multiple segmentation possibilities while tracking cells, even in terabyte-scale datasets (Fig. 1e), that are challenging for existing joint segmentation and tracking solutions[26,28–30].

Hence, the canonical Ultrack input consists of two maps for each time frame: (i) a foreground map distinguishing potential cells from the background and (ii) a grayscale image representing the multilevel contour map (that is, UCM; Fig. 1c). These two input maps can be generated from any method, making Ultrack compatible with a wide range of existing workflows (see Methods for technical details). Therefore, the same algorithm is always applied in every application presented here, and the varying factor is the process of obtaining the multilevel contour map.

### Tracking by maximizing temporal consistency
Once all candidate segments and links between segments are determined, tracking involves solving a complex combinatorial problem of selecting and associating the correct cell segments across time from the set of multiple segmentation and linkage hypotheses (Fig. 1b and Supplementary Video 1). As shown in Fig. 1d, we formulate this combinatorial problem as an integer linear program (ILP)[28,44] that simultaneously solves for the optimal selection of temporally consistent segments while adhering to biological constraints. These constraints include cell division, cells entering (for example, moving into the frame) or exiting (for example, cell death or moving out of the field of view), and ensuring that a pixel is not assigned to more than one cell[45], denoted by auxiliary variables $x_\delta$, $x_\alpha$ and $x_\beta$, respectively (Fig. 1d). This efficient and versatile mathematical formulation leverages highly optimized solvers (Gurobi[46], Coin[47]) and thus can handle tens of millions of segments from terabyte-scale datasets (Fig. 1e). Once solved, the ILP formulation not only selects the most appropriate segments but also encodes the linkage data necessary to reconstruct the cell lineages from the optimal association between the selected segments (see Methods for association score definition and ILP formulation details). A 3D image of a developing zebrafish embryo recapitulating cell lineages and migration patterns is shown in Fig. 1f and Supplementary Video 1.

### Optimal parameter selection by temporal consistency
Image segmentation techniques, from classical methods like watershed to advanced deep-learning models, face considerable challenges in parameter tuning due to several factors: (i) computational: extensive resources are required to explore a large parameter space; (ii) practical: users often lack the expertise to train or tune parameters effectively; and (iii) technical: a single optimal parameter set may not fit all aspects of a diverse dataset, such as when segmenting cells of varying sizes.

Ultrack addresses these challenges by integrating multiple segmentation labels derived from different parameter settings into a single multilevel contour map. This approach leverages the temporal consistency of the tracking process to select the most appropriate segments for each cell (Fig. 1d), resulting in more effective segmentation than what could be achieved with any single parameter setting—even a single optimal one.

To demonstrate this key feature of Ultrack, we tracked human hepatocarcinoma-derived cells expressing YFP-TIA-1 (ref. 48) in their nucleus and cytoplasm from the CTC[11] using the Cellpose 'cyto2r'

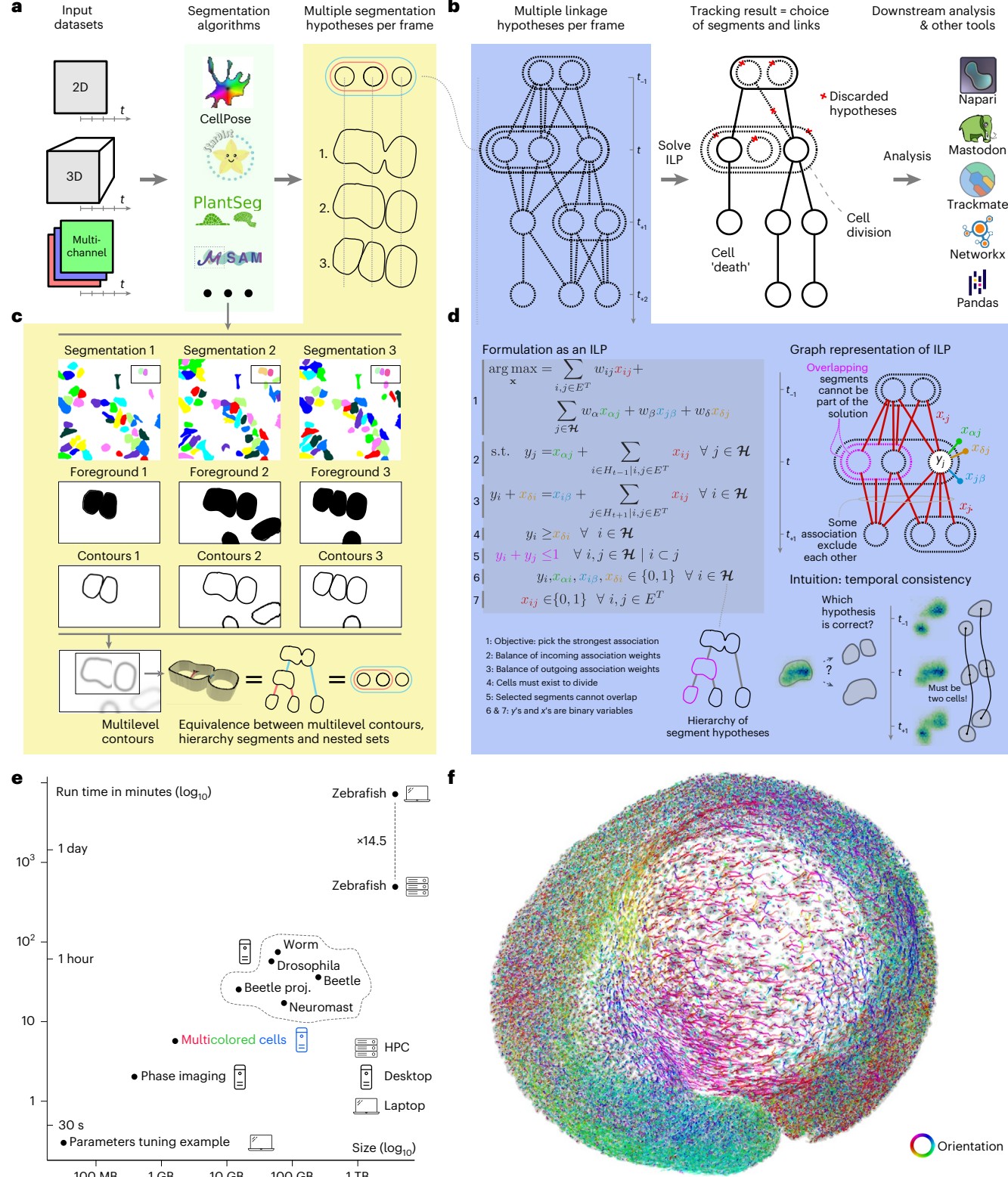

**Fig. 1 | Ultrack overview. a**, The Ultrack pipeline can process a variety of input images, including 2D, 3D and multicolor datasets. These images are then processed by any segmentation algorithm or a combination of them. **b**, Ultrack builds segmentation hypotheses between frames for tracking and solves an ILP problem to identify cell segments and their trajectories. The resulting tracks can be exported in various formats compatible with multiple platforms. **c**, Illustration of how segmentation hypotheses are built using ultrametric contours: multiple segmentations are provided (first row), their binary contours are extracted (second row), and these are combined into a multilevel contour, equivalent to a hierarchy (third row). **d**, Joint segmentation and tracking is performed using an ILP formulation, modeling cell behaviors (for example, cell appearance $x_\alpha$, division $x_\delta$, death or exit $x_\beta$) while finding nonoverlapping cells with maximal association over time. **e**, Ultrack can process arbitrarily sized datasets and scales from a laptop to a HPC cluster. **f**, Projection of a 3D image of a developing zebrafish embryo, with cell tracks overlaid and colored to indicate track orientation in the $xy$ plane. Logos in **b** reproduced with permission from: TrackMate, Jean-Yves Tinevez; Napari, Juan Nunez-Iglesias.

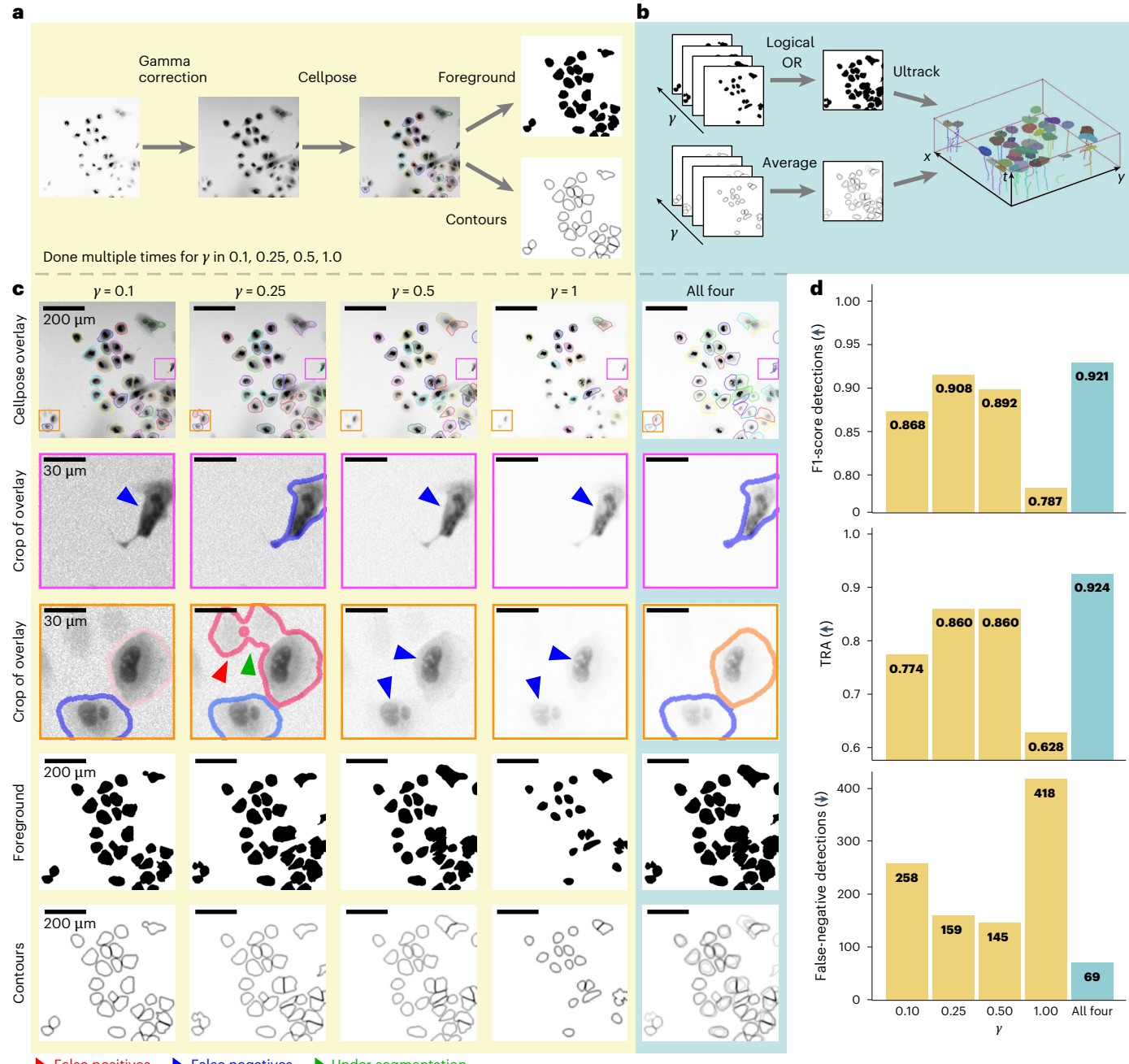

▶ False positives  ▶ False negatives  ▶ Under segmentation

**Fig. 2 | Multiple segmentation hypotheses alleviate the curse of parameter tuning. a**, Segmentation pipeline using gamma correction with varying settings ($\gamma$ = 0.1, 0.25, 0.5 and 1.0) to generate diverse segmentation hypotheses. **b**, Ultrack's approach: integrating multiple foreground and contour maps for joint segmentation and tracking. **c**, Impact of gamma correction on segmentation quality: original images (top row), segmentation results (second and third rows) with errors highlighted by colored arrows, foreground maps (fourth row) and contour maps (fifth row). **d**, Quantitative comparison of F1 score of detections, TRA and false-negative cell detections for individual gamma settings and Ultrack's combined approach (all four). Arrows within parenthesis indicate whether lower (↓) or higher (↑) values are better.

weights[32] without any fine-tuning (that is, deep-learning retraining). In this dataset, cells have varying intensities, impairing the model's ability to consistently segment every cell correctly. For example, using original image intensities ($\gamma$ = 1.0), the segmentation model fails to detect dimmer cells (Fig. 2c). Conversely, altering the dynamic range by taking the power of the image intensities with values closer to zero ($\gamma$ = 0.1, 0.25, 0.5) improved the segmentation of dimmer cells but led to incorrect segmentation or non-detection of now-saturated bright cells (Fig. 2c).

To segment and track the combination of all parameterizations of gamma, we extracted the binary foreground and contour maps from the Cellpose labels (Fig. 2a). Next, we combined them into a single binary foreground and a multilevel contour map for tracking (Fig. 2b and Supplementary Video 2).

We evaluated the tracking and segmentation performance for each individual gamma setting, as well as for the combined contour map approach. To assess performance, we compared the results against the publicly available ground truth provided by the CTC. Figure 2d summarizes the average results obtained from the two publicly available datasets. We used three metrics for evaluation: the F1 score of detections, the tracking accuracy (TRA)[49] and false-negative detection rate.

For false-negative detection, lower scores indicate better performance, while for F1 and TRA, higher scores are better (Methods).

These results demonstrate that Ultrack achieves more accurate tracking by leveraging multiple segmentation hypotheses, even outperforming the optimal single parameter setting ($\gamma = 0.5$ or $0.25$ depending on the metric). This indicates that combining diverse segmentation inputs is often more effective in practice, thus liberating the need for precise parameter tuning and offering a more robust approach to cell tracking.

### Enhanced tracking through multichannel integration

Multicolor labeling has been shown to enhance cell segmentation and tracking capabilities, particularly in complex cellular environments[50,51]. However, applying state-of-the-art deep-learning models to such data often requires training on dataset-specific annotations, as pre-trained weights are typically optimized for single-channel data. Ultrack addresses this challenge by combining multiple segmentation inputs in two ways: by varying parameters, as demonstrated earlier, and by integrating outputs from different segmentation algorithms applied to separate color channels.

We demonstrate this key feature of Ultrack on a three-channel 'multicolor' dataset of metastatic breast adenocarcinoma cells (MDA-MB-231) with red, green and blue markings from a lentiviral gene ontology (LeGO) vector system[52], and show that we can effectively use models like Cellpose's 'cyto2'[32] alongside classical algorithms by applying them independently to each channel and then combining the outputs into a single multilevel contour representation. This approach circumvents the need for retraining typically required by other approaches[37,53]. We combined six segmentation outputs from three image channels—three from Cellpose and three from Otsu[54] with watershed[55,56]—into a single contour and detection map for Ultrack (Fig. 3a). While classical image processing approaches like watershed are generally less precise at splitting cell instances, they provide greater control and allow for the detection of dimmer cells, complementing the more sophisticated Cellpose output. Furthermore, Ultrack can use the three color channels as a feature when associating segments between frames, which helps connect the correct segments across time (Fig. 3b). See Methods for a description of the color features association.

We assessed the effectiveness of our multichannel multi-algorithm approach using progressively more sophisticated strategies: (i) individual pipelines using Cellpose and watershed algorithms on a grayscale version (maximum intensity across multiple channels); (ii) each algorithm applied independently on multiple channels; (iii) combination of outputs from both algorithms in the multicolor configuration (Fig. 3a); and (iv) integration of color features into the association scores to improve linking of segments.

We manually curated 413 cell tracks, creating a gold-standard dataset for benchmarking our results. Demonstrating several key improvements. Incorporating color channels enhances cell distinction compared to grayscale versions, notably reducing under-segmentation (Fig. 3b). The combination of watershed and Cellpose algorithms proves particularly effective, improving the detection of dimmer cells while maintaining Cellpose's accurate segmentation. Furthermore, color-feature linking improves tracking in ambiguous scenarios (Fig. 3b). Overall, Ultrack steadily increased accuracy as additional features (that is, segmentation labels, color information) were incorporated. This demonstrates its effectiveness in creating valid lineages (Fig. 3c and Supplementary Video 3) from a large pool of segmentation hypotheses without requiring a fine-tuned segmentation model, highlighting its versatility and robustness in multicolor cell tracking scenarios.

### Intensity-based tracking from label-free virtual staining

As demonstrated in the previous section, off-the-shelf segmentation models face challenges when datasets deviate greatly from the training distribution. Users then often resort to time-consuming tasks such as annotating multiple segmentation labels to fine-tune a new model or implementing custom algorithms based on their expertise. This is particularly common because of the high diversity of imaging modalities and datasets typically encountered in biology. Here, we show that our approach can avoid these requirements by directly tracking image intensities as if they were multilevel contours, thus circumventing the need for segmentation labels as input.

Our method relies on a single assumption: the intensity map must highlight cell boundaries in some way. This could be achieved through membrane labeling, which delineates cells as a contour map, or through nuclear labeling, which highlights cell interiors. By inverting the latter, we obtain an image where boundaries have higher intensity values than cell interiors.

To evaluate this approach, we tracked A549 cells using quantitative phase imaging (QPI) data acquired with recOrder[57] on a widefield transmission microscope. Traditional segmentation models like Cellpose perform poorly on this data because QPI was not part of their training set[58]. We obtained the intensity map using VSCyto2D[58], a convolutional neural network for virtual staining that predicts membrane and nuclear markers from label-free data.

To create the multilevel contour map, we combined both the nuclear and membrane channels by subtracting the normalized and filtered nuclear channel from the filtered membrane channel (Extended Data Fig. 1a). The foreground was computed by manually thresholding the membrane channel and applying morphological filtering to close holes and remove small objects. We then applied Ultrack to the foreground and the intensity map encoding cell contours (Supplementary Video 4).

For benchmarking, we tracked the virtual stained nuclear channel and manually curated a set of 247 tracklets of lineages starting in the first frame to the last frame. The centroids of the tracked cells were used to create reference gold-standard datasets following the CTC format.

We compared these data against Cellpose with 'livecell_cp3' weights[59] applied to the QPI channel, followed by tracking. With Ultrack applied to the combined membrane and nuclear virtual staining channels, we detected most of the cells (Extended Data Fig. 1b,c), while the off-the-shelf model failed to recognize them. We missed 2,802 fewer associations between cells (that is, false-negative edges) and 3,014 cell detections (that is, false-negative detections) than virtual staining with Ultrack, at the cost of detection of 1,000 false-positive cells yielding a much higher F1 score (0.812 versus 0.347).

This shows that tracking from image intensity directly, in this case from virtual staining, offers a competitive alternative to supervised segmentation and tracking methods, which often fail when presented with unseen imaging modalities, such as label-free quantitative polarization, Zernike phase contrast, differential interference contrast or for different cell types, thus eliminating the need for extensive human annotation to train segmentation models.

### Improving tracking by temporal registration

Imaging systems must balance multiple factors, including the field of view size, spatial resolution, imaging speed and depth[60]. A critical limitation often encountered is acquisition speed (that is, temporal resolution), which hampers cell tracking performance. This is especially problematic when cells move rapidly and have similar appearances, complicating the assignment of cells between consecutive frames.

Registration of adjacent frames often compensates for limited time resolution. However, collective cell motion within growing or deforming tissues frequently demonstrates local coherence that does not conform to simple linear or affine transformations. This is due to the complex dynamics of some biological processes. In such cases, nonlinear registration, such as movement vector fields[61,62], is more appropriate. This approach effectively minimizes apparent motion when cells present distinct migration patterns.

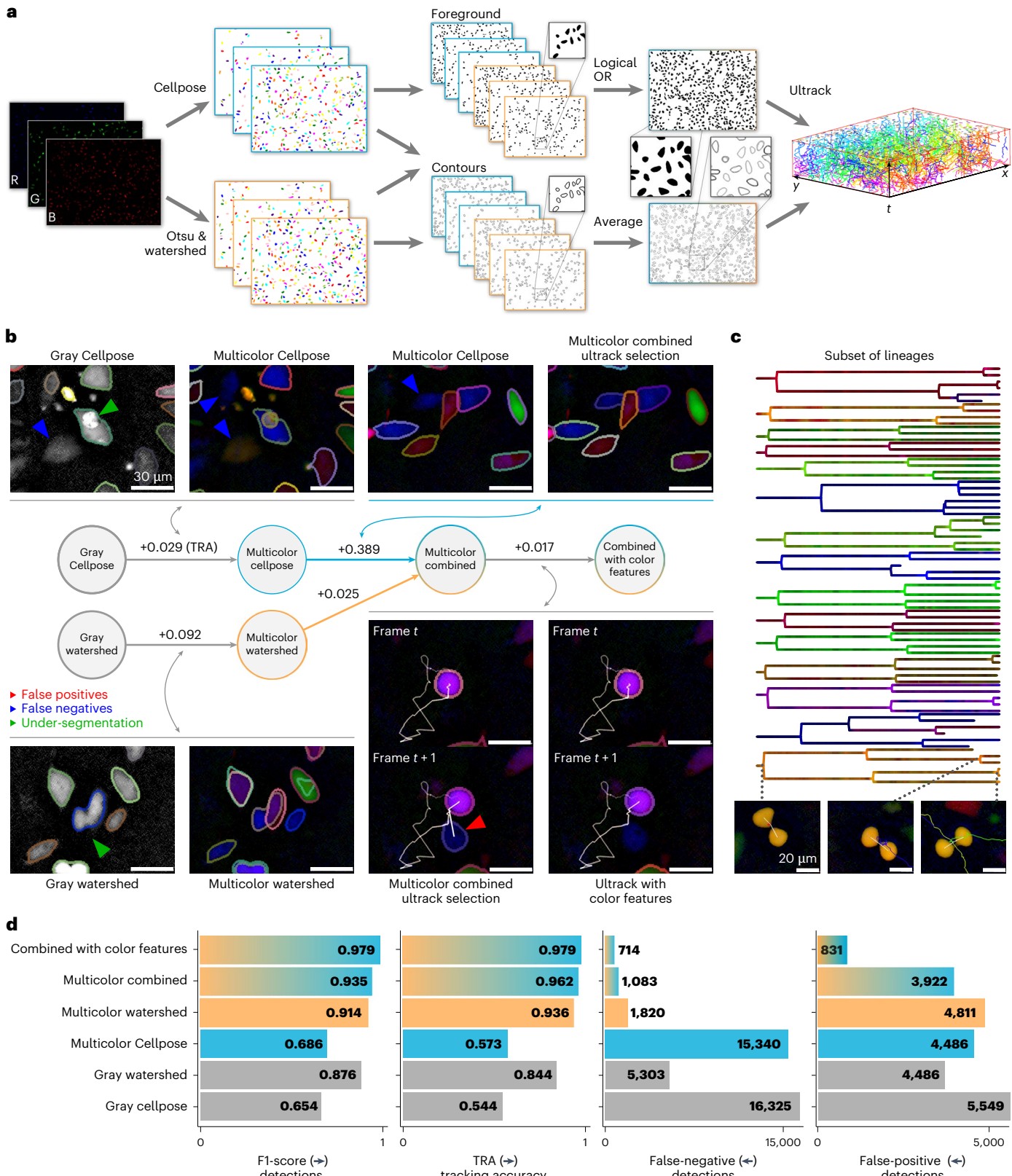

**Fig. 3 | Ultrack enhances multicolor cell tracking by integrating diverse segmentation algorithms. a**, Multichannel, multi-algorithm tracking pipeline: individual labels are generated for each color channel (R, red; G, green; B, blue), from which foregrounds and contours are derived and subsequently integrated for joint segmentation and tracking. **b**, Stepwise improvement of segmentation strategies (left to right): (i) individual algorithms applied to grayscale images, (ii) applications to multicolor data, (iii) combination of labels from both algorithms and colors and (iv) enhancement with color features in association scores. The central diagram illustrates the progression of methods, with numbers indicating improvements in the TRA metric. Colored arrows highlight specific error types. **c**, Representative subset of lineages recovered by Ultrack, showcasing cells' respective colors, and examples of captured division events. **d**, Quantitative evaluation of tracking accuracy for the approaches described in **b**. Final combined results are available at https://t.ly/K-hA7.

Computing nonlinear registration can be extremely time-consuming or require complex, specialized algorithms. To address this, Ultrack provides its own graphics-hardware-accelerated (GPU) routines to compute vector fields for 2D and 3D time-lapse recordings. These routines are based on automatic differentiation from deep-learning models[63], offering efficient computation without sacrificing accuracy (Methods). The movement vector fields can be integrated into the tracking step, applying local movement to each candidate segmentation. This approach avoids altering the original image data. Moreover, Ultrack accepts any vector fields represented as a grid, allowing users to utilize their preferred registration methods.

We evaluated the effectiveness of using vector fields with Ultrack on the *Tribolium castaneum* cartography dataset[64] from the CTC. This dataset features nuclear-labeled embryo cells imaged using selective plane illumination microscopy. The epithelial cells' ellipsoidal surface was transformed into planar cartography using the ImSAnE toolbox[65] to facilitate downstream analysis. However, this cartographic transformation introduced distortion and excessive motion on the polar boundaries (Extended Data Fig. 2a and Supplementary Video 5).

We compared the tracking accuracy with and without flow registration (Extended Data Fig. 2b) at various temporal resolutions, starting by skipping ten frames ($\Delta t = 10$) up to the original frame rate ($\Delta t = 1$; Extended Data Fig. 2c,d). The results show: (i) at $\Delta t = 10$ tracking accuracy improved from 0.443 to 0.623 with flow registration, in this scenario the most motile cells move more than 150 voxels between frames, a substantial distance given that the median cell diameter of approximately 19.64 voxels; (ii) at $\Delta t = 5$ accuracy increased from 0.571 to 0.756; (iii) at original time resolution ($\Delta t = 1$) similar results were obtained (0.927 versus 0.929). Extended Data Fig. 2c provides a specific example where the flow field achieves the correct result, while a cell identity switch occurs in the unregistered case.

These results demonstrate that Ultrack's temporal registration capability enhances tracking accuracy, particularly when dealing with complex cell movements or lower temporal resolution datasets.

Concurrent work has shown that Ultrack remains effective even in scenarios that prioritize speed over resolution, such as neural dynamics imaging, where *z*-resolution is often reduced to enable faster acquisition. The WormID benchmark[66] demonstrates Ultrack's robustness across a range of imaging resolutions, from $1 \times 0.1604 \times 0.1604\ \mu m$ to $1.5 \times 0.32 \times 0.32\ \mu m$ per voxel[67].

All these features—including its ability to handle multiple segmentation hypotheses, integrate multichannel data, leverage intensity-based tracking and apply temporal registration—collectively enable robust cell tracking in complex biological contexts. These capabilities are particularly valuable when confronting the challenges posed by dense, 3D tissues with dynamic cellular behaviors. To demonstrate the power and versatility of this approach, we next evaluated Ultrack's performance on one of the highly competitive CTCs[11].

### Ultrack's top performance at cell tracking in developing embryos

The CTC is a long-standing benchmark for cellular segmentation and tracking methods[11]. It provides participants with training datasets to train and tune their algorithm parameters. However, the test set annotations used for the competitive assessment and final scores are concealed and only accessible to the organizers. This structure prevents information leakage between the training and testing phases and offers an unbiased evaluation of each algorithm's performance. We assessed our method using five 3D datasets: breast cancer cells, *Caenorhabditis elegans* (worm)[68], *Drosophila melanogaster* (fly)[31] and *T. castaneum* (beetle), as well as cartographic projections of *T. castaneum* datasets[64].

The worm and breast cancer cells datasets include manually curated lineage data and cell coordinates, as well as silver-standard segmentation annotations derived from a consensus among prior challenge submissions[69]. Using these annotations, we trained a U-Net model[45,70] that predicts two outputs per voxel: (i) whether the voxel is on a cell contour and (ii) whether that voxel is in the foreground. We optimized these predictions using three dice losses[71]: (i) binary foreground prediction, (ii) binary cells' contour prediction and (iii) U-Net's intermediate cell contour predictions[72] (Methods). The cells' contours and foreground were then used as inputs for Ultrack. As shown in Extended Data Table 1, Ultrack achieves a top combined segmentation and tracking score of 0.844 for the worm dataset. For the cancer cells dataset, our approach underperformed, most likely because of over-fitting during training the segmentation model on limited data: our cross-validation reported an average TRA of 0.905 while achieving a much lower score of 0.818 on the test set.

In general, sufficient training data for segmentation are not always available or easily obtained, which is a typical situation users face. This is the case for the fly, beetle and beetle projection datasets, which lack ground truth to train segmentation models. To circumvent this limitation, the leading deep-learning-based submissions are based on point coordinates and movement, relying only on track annotations. Instead, we asked whether we could forgo deep learning and use classical image processing techniques for segmentation. We opted for a straightforward method. First, we detected the foreground using a difference of Gaussians filter and applied Otsu's threshold[54] to select the foreground voxels. Next, we used image intensity as a proxy for cell boundaries by inverting and normalizing these intensities between 0 and 1, so dimmer voxels, which are more likely to be the cells' contours, represent the highest values.

Combining this simple image processing approach with Ultrack's handling of multiple hypotheses implicit in the contour representation yields noteworthy results compared to competing approaches that require training data. As shown in Extended Data Table 1, for the fly dataset, Ultrack achieved a combined segmentation and tracking score of 0.708, compared to 0.617 for the next best method. Similarly, for the beetle dataset, Ultrack scored 0.841, while the next best result was 0.804 (see Methods for additional details). For the projection dataset, we obtained a lower score of 0.754—likely because of the distortions introduced by the projection.

Overall, these results demonstrate Ultrack's strong performance even without deep-learning models, achieving more first-place rankings on 3D datasets than any other method. The next best performer, MPI, achieved first place only twice on the TRA metric, while never ranking higher than third place in segmentation scores across their submissions (Extended Data Table 2). See Supplementary Video 6 for recordings of the results.

### High-quality cell tracking ground-truth via sparse labeling

While the CTC provides a valuable unbiased benchmark for evaluating segmentation and tracking, it relies primarily on human-generated annotations. These annotations, while crucial, are inherently limited in scale and potentially in quality due to the challenge of manually annotating the latest-generation 3D live microscopy datasets[1,2,6,7,9,15,16]. Such datasets often span hundreds of gigabytes to terabytes, recording thousands of cells over hundreds or more time points, making manual annotation a task that is not only time-consuming but often beyond human capability. Moreover, in densely labeled or multidimensional datasets, distinguishing individual cells can be highly uncertain and sometimes impossible[15,16,24,31,64].

Encouraged by Ultrack's performance on the CTC, we developed a higher-fidelity ground-truth dataset that could overcome these limitations and potentially provide a more challenging benchmark of a crowded and dynamic cellular environment from a developing zebrafish embryo. We applied a dual-labeling protocol that combines ubiquitous fluorescence nuclear labeling Tg(ef1α:H2B-mNeonGreen) with sparse random nuclear labeling at a distinct wavelength (pMTB-ef1-H2B-mCherry). Sparse labeling was accomplished by

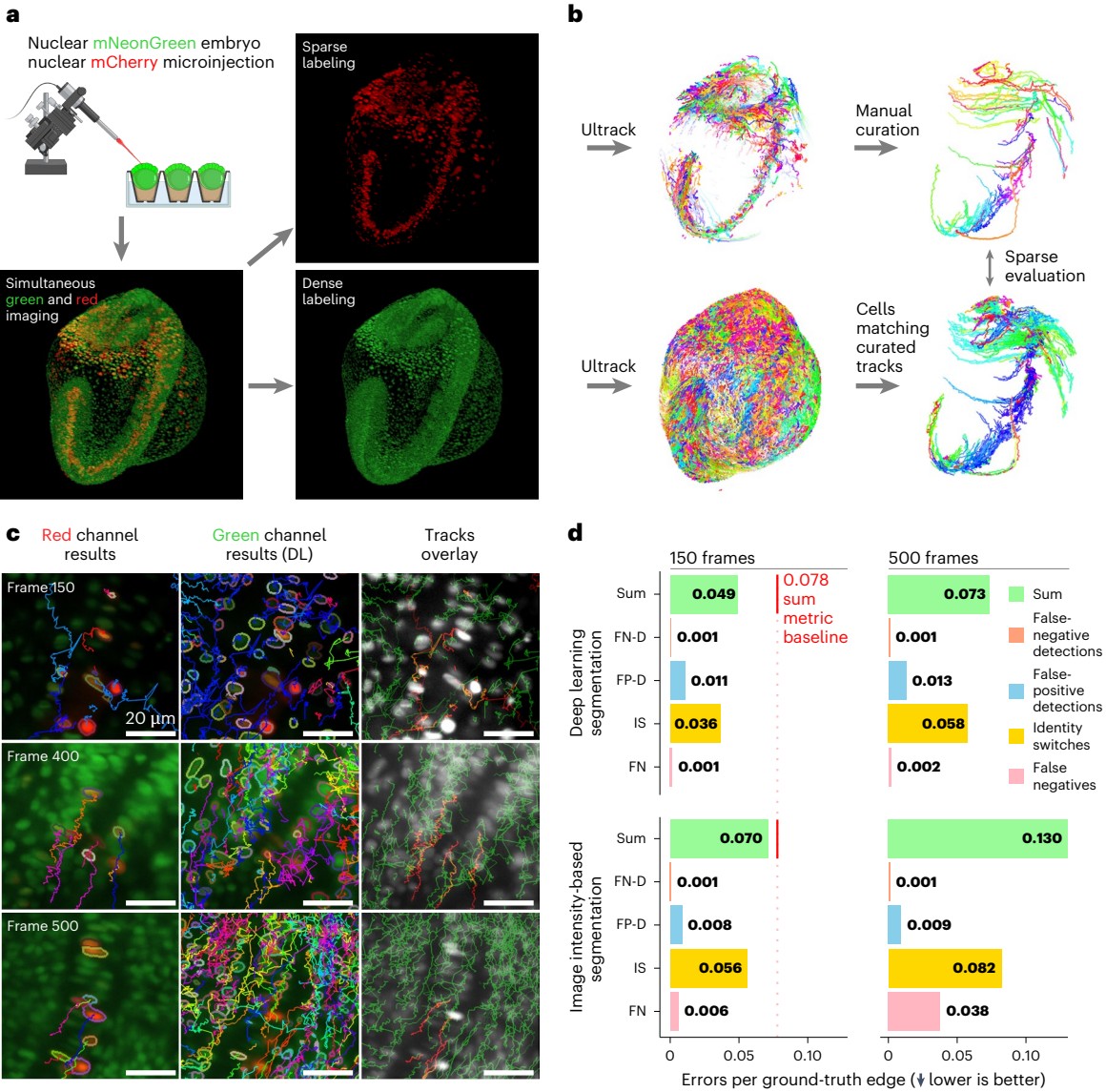

**Fig. 4 | Sparse fluorescence labeling enables high-fidelity tracking validation over extended time-lapse recordings. a**, Validation pipeline using sparse fluorescence labeling: (i) Embryos with ubiquitous green nuclear labeling Tg(ef1α:H2B-mNeonGreen) are injected with DNA plasmids expressing red fluorescent protein pMTB-ef1-H2B-mCherry; a single embryo is then selected for imaging. (ii) Simultaneous imaging of red and green channels. **b**, (iii) Independent tracking of cell nuclei in both channels. (iv) Selection and manual curation of a subset of cell nuclei tracks from the red channel. (v) Comparison of curated tracks to those obtained from ubiquitous labeling. **c**, Comparative visualization of red (first column) and green (second column) channel segmentation and tracking results at different time points, and combined results overlaid grayscale intensity colored in red and green (last column); cell density and tracking difficulty increase over time (top to bottom). **d**, Quantitative evaluation of tracking accuracy under different conditions and durations: using deep learning for cell contour estimation (first row) versus direct image intensity-based segmentation (second row); measurements taken up to the first 150 frames (first column) and 500 frames (second column). Metrics shown are proportions of false-negative edges (FN), identity switches (IS), false-positive divisions (FP-D), false-negative divisions (FN-D) and their sum. The red vertical line indicates the baseline sum error rate for 150 frames of 0.078 for comparison with Malin et al.[24]. Tracking results are available at https://t.ly/WZV-g. Schematic in **a** created with BioRender.com.

early-stage microinjection of a marker (Tol2 transposase-mediated integration; Methods) that propagates to daughter cells upon division, effectively recapitulating their lineage (Fig. 4a).

The sparsity of this second channel greatly simplifies tracking and allows for rapid curation into a set of high-quality, or 'platinum-level', annotations (Fig. 4b). Cells in the sparse channel were automatically tracked using Ultrack and then manually curated to ensure the highest quality standards (Fig. 4b). This curation process involved selecting long, green-overlapping, high-quality lineages, conducting visual inspections and making necessary corrections, resulting in 152 annotated tracklets spanning from 85 to 521 frames. This methodology not only simplifies the manual curation process but also reveals cell trajectories that are challenging to annotate manually using conventional methods on dense datasets (Supplementary Video 7).

As evident in Fig. 4c, tracking is easier at earlier stages of development when cells are less numerous, less densely packed, and when the optical properties of the tissue are more favorable for imaging. Despite these challenges, cell nuclei in the sparse channel are consistently easier to track compared to the dense channel (Fig. 4c).

We used these high-quality sparse-channel-derived tracking labels to evaluate Ultrack's accuracy on the ubiquitously labeled channel. We used two segmentation approaches as in ref. 45: (i) image processing to produce foreground maps and nuclei boundaries directly and (ii) a convolutional neural network to generate foreground and contour

maps. Hence, the Ultrack setup was kept the same, and only the input foreground and contour maps were modified. Figure 4c compares the tracking results using the sparse channel (red) and the ubiquitous channel (green) using the deep-learning-based contour map, approach (ii), for three frames (150, 400 and 500), showcasing the disproportionate difficulty of tracking cells in the dense channel versus sparse channel.

To quantitatively assess the performance, we evaluated our automatic dense tracking results against the manually curated sparse tracking over two time periods (Fig. 4d): the first 150 frames as the previous work[24] and an extended period of 500 frames, nearly covering the entire time-lapse recording. Assessing tracking performance with five metrics[24], all expressed as proportions relative to the total number of annotated edges: (i) false-negative edges (that is, missing connections between cells), (ii) identity switches (that is, incorrect association between cells), (iii) false-positive divisions, (iv) false-negative divisions and (v) the sum error rate that combines the first four metrics.

As expected, tracking accuracy drops for longer imaging periods (500 versus 150 frames; Fig. 4d), reflecting the growing complexity of the developing organism already apparent in Fig. 4c. Notably, Ultrack with a deep-learning-based input, which benefits from available training data, consistently outperforms plain image processing contour and foreground maps, maintaining comparable accuracy over 500 frames to what the image intensity-based approach achieves in just 150 frames.

To contextualize our results within recent advances in cell tracking for embryo-scale 3D datasets, we compared our findings to those reported by Malin et al.[24]. Their study, which used a similar multi-view light-sheet microscope setup[73], established a baseline sum error rate of 0.078 for tracking neuronal cells in zebrafish embryos during 150 frames. As shown in Fig. 4d, Ultrack achieves a lower proportion of sum errors for both deep-learning and intensity-image-based contour maps on 150 frames (0.049 and 0.070, respectively). Moreover, when using deep learning, we maintain accuracy (0.073) comparable to the baseline (0.078) even over a much more challenging 500-frame period. This extended performance is particularly noteworthy, as it underscores the robustness of our method in maintaining high tracking accuracy over prolonged periods of embryonic development despite increasing tissue complexity, cell density and imaging artifacts. These results not only validate the effectiveness of our dual-channel approach but also highlight Ultrack's potential to push the boundaries of long-term cell tracking in complex biological systems.

## Ultrack scales to terabytes of high-resolution data over large fields of view

To demonstrate Ultrack's scalability and performance on massive datasets, we used it to recover cell lineages of nuclear-labeled zebrafish embryos[9] acquired using the fast (one stack per minute), high-resolution (×20 magnification and 1.0-NA Olympus objective) and large-imaging-volume (≥1 mm³) DaXi light-sheet microscope[6], producing uncompressed time-lapse data ranging from 1.7 to 3.7 terabytes, capturing 8.6 h to 13.2 h of zebrafish development (Fig. 5a,b).

Our workflow began with training a U-Net to predict nuclei pixels (foreground) and their contours (Fig. 5a,b). We then applied Ultrack to these network predictions. Leveraging Ultrack's command-line interface (CLI) and SLURM[74], we processed these massive datasets within hours using distributed computing in a high-performance computing (HPC) environment. For example, a 3.7-TB dataset (791 × 448 × 2,174 × 2,423 voxels, T × Z × Y × X) was processed in approximately 8.2 h (Fig. 1e).

Ultrack's computational scaling extends across computing environments, from high-performance clusters to off-the-shelf laptops. Ultrack supports out-of-core processing: it can process arbitrarily large datasets, even those substantially larger than available memory (RAM)—as long as a single image (2D or 3D) can fit into memory (Methods). We could process the same 3.7-TB dataset on a 32-GB RAM laptop in 5 days, accessing data remotely via a 1-Gbps connection (Fig. 1e;

see https://github.com/royerlab/ultrack_supplementary/tree/main/hardware/ for hardware details).

The tracking resulted in millions of segments across hundreds of time frames. To assess performance, we manually evaluated time-lapse recordings until the first tracking mistake, recording the duration of error-free tracks. We used a stratified sampling strategy, manually annotating distinct regions (head, body, tail bud and presomitic mesoderm) at various time points and image qualities. From these strata, we randomly sampled approximately 140 lineages per embryo (n = 3). Figure 5d shows each fish's cumulative proportion of error-free tracks and their mean. On average, it took 62 frames for a tracking error to appear in 50% of the lineages, meaning that for a period of approximately 1 h of development (60 frames), half of the tracks are completely error-free, demonstrating Ultrack's robust performance over extended periods. Figure 5c and Supplementary Video 8 provide a detailed view of an automatically reconstructed lineage, capturing two rounds of cell divisions and illustrating Ultrack's ability to maintain accurate tracking through multiple cell generations.

These results underscore Ultrack's capacity to efficiently process and accurately track cells in multi-terabyte, high-resolution datasets utilizing HPC resources or standard hardware. This scalability and versatility make Ultrack a powerful tool for analyzing large-scale developmental processes at unprecedented resolution and scale.

## Ultrack delivers high-accuracy organ 3D tracking

Cell tracking is often crucial in scenarios with fewer cells, where perfect lineage reconstruction is necessary to uncover precise details of cell behavior, such as division, migration and death[75,76]. To evaluate Ultrack's performance in such a context, we chose the zebrafish neuromast as our model system, given its well-defined structure and importance in studying sensory organ development. Previous efforts in segmenting and tracking cells in this organ included manual tracking of cells[77] or semiautomated segmentation of single frames[78]. To our knowledge, no other study has reported the fully automatic segmentation and tracking of cells in this complex 3D organ for extended time periods.

We imaged a zebrafish neuromast for 42 h using membrane and nuclei markers. The nuclei marker, Tg(she:H2B-EGFP), only labels the neuromast cells (hair, support and mantle cells), while the membrane marker, Tg(cldnb:lyn-mScarlet), labels the neuromast, surrounding skin cells and ionocytes. Our segmentation model was trained to detect only cells that express both markers.

Imaging was performed on an Olympus IX83 microscope equipped with a microlens-based, super-resolution spinning-disk confocal system (VT iSIM, VisiTech International), using a ×60 1.3-NA silicone-oil objective. This setup yielded a dataset of 500 frames in two colors, with volumes of 73 × 1,024 × 1,024 voxels, at a resolution of 250 × 76.6 × 76.6 µm per voxel. Frames were captured every 5 min.

To ensure optimal segmentation accuracy, we fine-tuned a 3D Cellpose model[32] to segment cells using both color channels. We then converted the Cellpose predictions into foreground and contour maps for input into the Ultrack algorithm (Fig. 6a). This approach allowed us to reconstruct lineages spanning the entire time-lapse recording, capturing diverse cellular events such as migration within the organ, cell division and cell death (Fig. 6b,c and Supplementary Video 9).

To assess Ultrack's accuracy, we visually inspected the generated lineages and manually corrected any errors, creating ground-truth reference lineages. Using the CTC evaluation routines, we quantified Ultrack's performance (Fig. 6d). Ultrack achieved a TRA score of 0.9989, demonstrating its high precision in this challenging scenario.

We compared Ultrack's performance against TrackMate[37], using identical Cellpose segmentations as input. To ensure a fair comparison, we first optimized both Ultrack's and TrackMate's parameters using a different 50-frame dataset. Key parameters tuned included minimum and maximum cell size and the maximum movement. We then applied these optimized settings to the full 500-frame dataset reported here.

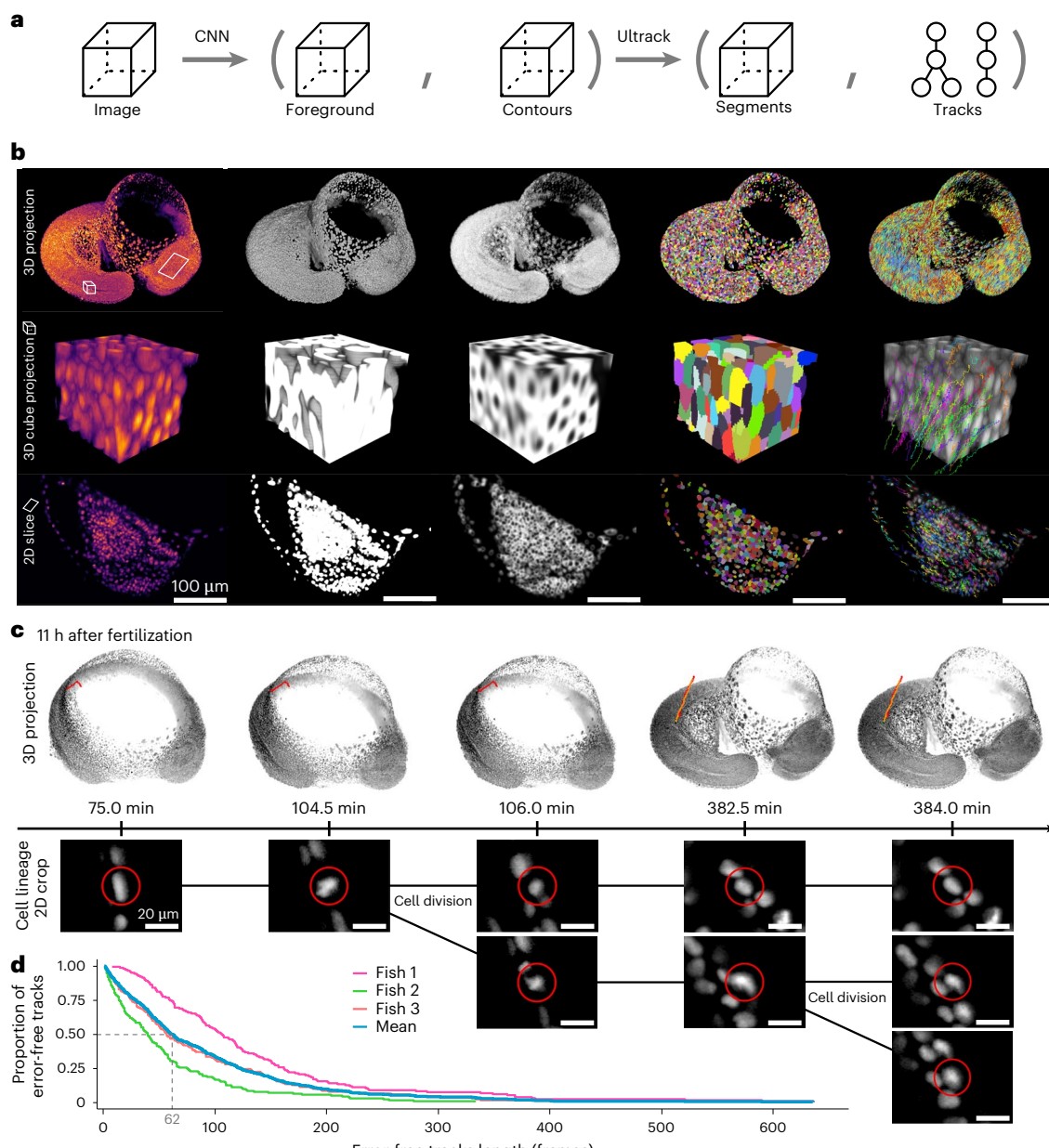

**Fig. 5 | Multi-terabyte cell tracking of zebrafish embryo. a**, Pipeline for joint segmentation and tracking using a convolutional neural network (CNN) and Ultrack. **b**, Intermediate results at each stage: original image (first column); predicted foreground (second column) and contour probabilities (third column); Ultrack segmentation (fourth column) and tracking results (fifth column). **c**, Detailed views of fully automated lineage reconstructions, showcasing two rounds of cell divisions and their corresponding 2D slices. **d**, Distribution of the proportion of error-free tracks for varying lengths across three embryos. This indicates that, on average, 50% of the lineages are error-free for at least 62 frames. Tracking data results are available at https://t.ly/vXFyb.

In this case, Ultrack's tracking results exhibit higher F1 scores of 0.9951 versus 0.9895 and more than twofold reduction in errors (279 versus 721), with 442 fewer mistakes needing manual correction when compared to TrackMate. The primary source of errors in TrackMate's results stemmed from imperfect Cellpose segmentations (Fig. 6d). Despite TrackMate's segmentation filtering options, it struggled to correct issues such as over-segmentation and false merges. In contrast, Ultrack's approach naturally incorporates these imperfect segments into its hypotheses, allowing for more accurate merging and tracking (Fig. 6e).

These results demonstrate Ultrack's ability to maintain high tracking accuracy in this complex, long-term imaging scenario and, thus, its utility for detailed studies of cellular behavior and tissue dynamics.

## Democratizing high-performance cell tracking at all levels

Cell tracking software often represents one of the most complex components in image analysis pipelines, primarily due to the necessity of handling the time dimension—a factor frequently overlooked in segmentation and other image processing operations. Moreover, it operates on non-matrix data (that is, track lineages), where efficient data processing routines are less prevalent. This complexity has historically restricted accessibility to a broad user base, with only a few long-standing solutions like TrackMate[37] and others[21,22,31], alongside specialized solutions for handling larger datasets[24].

Ultrack addresses these challenges by offering a versatile, user-friendly solution that caters to a wide range of users, from biologists with minimal programming experience to machine learning experts requiring customizable inputs. By providing multiple

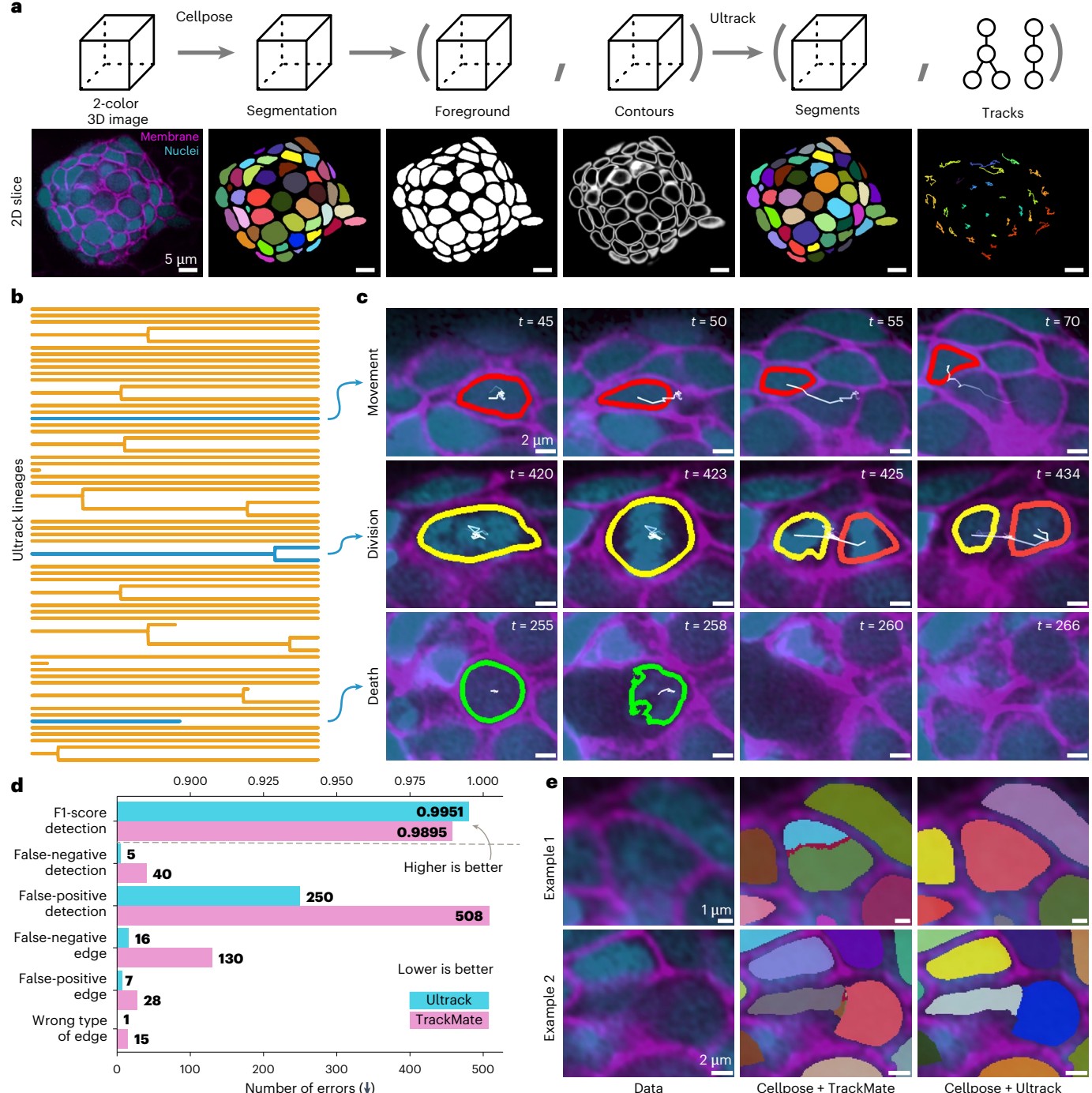

**Fig. 6 | Near-perfect nuclear-based and membrane-based 3D tracking of zebrafish neuromast cells. a**, Pipeline for joint segmentation and tracking using Cellpose and Ultrack, showing 3D schematics (top) and 2D slice examples (bottom) at each stage: original two-color image, Cellpose segmentation, foreground and contour probabilities, Ultrack segmentation and tracking results. **b**, Dendrogram recovered by Ultrack for 71 cells over 500 frames (41.7 h), with orange lines representing cell lineages and blue highlights indicating examples detailed next. **c**, Time-lapse images illustrating cell movement (red outlines), division (yellow/orange outlines) and death (green outlines). **d**, Quantitative comparison between Ultrack (blue) and TrackMate (pink). The bar chart shows error counts for various metrics, with Ultrack obtaining 442 fewer mistakes than TrackMate. **e**, Two examples demonstrating Ultrack's robustness to over-segmentation, showing the original image (left), fragmented TrackMate results (middle) and accurate Ultrack segmentation and tracking (right) despite imperfect Cellpose input.

interfaces and scalable processing capabilities, Ultrack democratizes access to advanced cell tracking technology, enabling efficient analysis of datasets of various sizes across different computational resources (Fig. 1e).

At the heart of Ultrack's accessibility is its multi-interface approach. Ultrack offers two options for users who prefer a graphical

interface: Napari and Fiji. The Napari plugin (Extended Data Fig. 3a) integrates transparently with the Python ecosystem, providing a comprehensive set of features for noncoding users. This interface offers graphics-hardware-accelerated preprocessing, vector-field flow computation and full configuration of Ultrack parameters. Complementing this, the Fiji plugin[35] (Extended Data Fig. 3b) exposes Ultrack's

algorithm to Fiji's mature ecosystem and broad user base, allowing seamless integration with existing Fiji workflows and ensuring compatibility with popular tools like TrackMate[37] and Mastodon[38]. Supplementary Video 10 shows them in action.

For users comfortable with programming, Ultrack provides a user-friendly Python API (Extended Data Fig. 3c,d) that exposes all of its functionalities, allowing for easy customization and reproducible workflows. This API accepts various array formats (for example, NumPy, Zarr and Dask), and can export data into multiple file formats, including networkx[79], tables (that is, data frames)[80], CTC[11] and TrackMate[37] formats, enhancing its interoperability with other tools and workflows. Additionally, Ultrack offers a CLI (Extended Data Fig. 3e) designed for batch processing and streamlined workflows, which supports distributed processing in HPC environments. This CLI accommodates any file format supported by existing Napari reader plugins, eliminating the need for data conversion—a crucial feature for handling large datasets.

## Discussion

Ultrack advances cell tracking by offering a robust, scalable solution for analyzing complex cellular dynamics across various scales and dimensions. Its integration of segmentation and tracking, leveraging multilevel contour maps (UCMs), enables effective analysis in challenging scenarios where previous methods often struggled.

The method's versatility is demonstrated through its performance in 3D embryonic CTCs, even when using non-learned classical image processing algorithms. This ability to track without training data or deep fine-tuning addresses a critical challenge in the microscopy field, where 3D annotations are often scarce[11]. While Ultrack can leverage learned models when available for optimal performance, it remains effective even without them. Furthermore, Ultrack's capacity to combine results from parameter sweeps eliminates the common problem of identifying optimal segmentation parameters. This flexibility extends to working directly with image intensities thus circumventing the need for explicit segmentation in certain scenarios, for example, virtual staining of label-free images. This approach is not limited to label-free imaging, such as QPI, but extends to virtually any imaging method—whether conventional fluorescence microscopy, label-free techniques or virtual staining approaches—that can produce intensity maps highlighting cellular structures, such as nuclei or membranes. By leveraging these intensity-based representations and using direct tracking, Ultrack offers an adaptable and efficient solution to avoid extensive annotation or fine-tuning (that is, deep-learning retraining) of segmentation models. Ultrack also enables processing multichannel data by combining multiple single-channel labels, a feature that, when paired with technologies for engineering multicolored cells[50,51], promises a new generation of high-accuracy lineage reconstruction. Finally, Ultrack addresses challenges associated with low temporal resolution through GPU-accelerated flow-field registration between frames, broadening its applicability to more experiments.

The introduction of a dual-channel sparse labeling approach for validation offers a more challenging and heterogeneous benchmark compared to previous studies focused on specific cell subsets or biological contexts[15,16,31,64]. This approach paves the way for more robust validation of tracking algorithms over extended periods in large datasets.

Ultrack's scalability, tested on both HPC clusters and laptops, addresses the growing demand for tools capable of handling increasingly large and complex datasets produced by advanced imaging technologies[1,6,7]. This flexibility in computational requirements democratizes access to high-quality cell tracking across diverse research settings.

Ultrack's integration with popular platforms like Fiji[35] and compatibility with established tracking frameworks such as TrackMate[37] and Mastodon[38] positions it for widespread adoption. Its multiple interfaces, ranging from graphical user interfaces (GUIs) to command-line tools for HPC environments, align with community efforts to make complex computational tools more accessible to biologists[19,20,33,34,36,81].

Despite its strengths, Ultrack has its limitations. First, the segmentation hypotheses are only as good as the provided foreground and contour maps. While Ultrack is robust to occasional segmentation errors, it may fail when presented with an overwhelming number of incorrect segmentations or systematic errors that persist over time—in such cases, the ILP optimization might converge on erroneous solutions. Additionally, as shown in Extended Data Fig. 2, large cell movements between adjacent frames violate our assumption of segmentation consistency and can impair tracking performance unless registration is used.

A few strategies can improve Ultrack's effectiveness. For instance, since Ultrack uses region-merging hierarchical segmentation, contour maps that initially over-segment cells are actually beneficial, as they generate more hypotheses for the algorithm to consider. Additionally, as with other tracking approaches, frame registration enhances tracking performance. To help users optimize these aspects, we maintain comprehensive documentation (https://royerlab.github.io/ultrack/optimizing.html), providing guidelines and practical tips for achieving optimal tracking results across different applications.

One might question why we chose to use foreground and contour map prediction models rather than established instance segmentation approaches like Stardist[18] or MicroSAM[20] in Figs. 4 and 5. The answer lies in the nature of network contour predictions: their probabilistic outputs capture subtle variations in cell boundary likelihood, producing smoothly varying contour values. This continuous representation generates a richer set of segmentation hypotheses compared to instance segmentation methods, which output discrete labels that must then be converted to foreground and contour maps. The probabilistic nature of contour predictions thus provides Ultrack with more information to work with during the optimization process.

Looking ahead, several promising avenues for future development emerge: (i) Integration of advanced deep-learning techniques, particularly transformer-based models for sequence prediction[82], could further improve long-term tracking accuracy. (ii) Expansion of the sparse labeling approach to generate large-scale, high-quality datasets for training and benchmarking future tracking algorithms. (iii) Development of interactive correction tools leveraging Ultrack's multiple segmentation hypotheses to facilitate efficient manual curation of tracking results. (iv) Exploration of Ultrack's potential applications beyond developmental biology, including cancer research, immunology and regenerative medicine.

In conclusion, Ultrack's innovative approach for joint segmentation and tracking, combined with its scalability and accessibility, has the potential to accelerate research across a wide range of biological disciplines by enabling more accurate, efficient and accessible cell tracking. Ultrack opens new avenues for investigating complex cellular behaviors, tissue dynamics and developmental processes at unprecedented scales and resolutions.

## Online content

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

## Methods

### Animals

All zebrafish care and experimental procedures complied with protocols approved by the Institutional Animal Care and Use Committee at the University of California San Francisco (UCSF). Adult fish were maintained at 28.5 °C and fed twice daily using an automatic feeder[83]. Embryos were raised at 28.5 °C, and staged based on hours post fertilization (hpf). For light-sheet microscopy and injection experiments, we used two transgenic lines: Tg(h2afva:h2afva-mCherry; a gift from J. Huisken, University of Göttingen) and Tg(ef1α:H2B-mNeonGreen; a gift from D. Wagner, UCSF). Zebrafish neuromasts were imaged using the transgenic line Tg(cldnb:lyn-mScarlet) crossed with Tg(she:H2B-EGFP)[84,85].

### Multicolored cell line generation and imaging

MDA-MB-231 metastatic breast adenocarcinoma cells (American Type Culture Collection) were maintained in Dulbecco's Modified Eagle Medium (DMEM; VWR) with 10% (vol/vol) fetal bovine serum (FBS; Seradigm VWR), and penicillin and streptomycin (50 U ml$^{-1}$; Thermo Fisher Scientific). Cells were stably modified with a variety of red, green and blue markings with a LeGO vector system, as described in Weber et al.[52]. Lentivirus was prepared as described in Shah et al.[86]. In brief, pseudoviral particles were generated using 293TN cells (System Biosciences, SBI) co-transfected with lentiviral packaging and envelope plasmids (psPAX and pMD2.G, gifts from D. Trono), and one plasmid of interest containing fluorophore eGFP, mCherry or EBFP2 (Addgene, 25917, 27339 and 85213). Supernatants containing lentivirus were collected, passed through a 0.45-μm filter, concentrated by ultracentrifugation at 200,000$g$ for 4 h at 4 °C, resuspended in phosphate-buffered saline (PBS; VWR) and stored at −80 °C. MDA-MB-231 cells were transduced simultaneously with all three constructs at equal viral titer and a range of multiplicity of infections). To optimize color diversity, a fraction of cells expressing fluorophore was assessed at 5 days after transduction using fluorescence microscopy, and a condition with ≈50% transduction rate of all fluorophores (multiplicity of infection ≈ 0.7) was selected for subsequent experiments. Cells were seeded 12 h before imaging at a density of ≈10,000 per well on a glass-bottom plate coated with 50 μg ml$^{-1}$ rat tail type I collagen (Corning) in 0.02 N glacial acetic acid (Sigma) overnight at 4 °C. Imaging was performed on a Keyence BZ-X800 imaging system equipped with temperature, humidity and CO$_2$ control, and images were acquired at a magnification of ×10 at 10-min intervals for 2 days.

### Microinjection for zebrafish sparse labeling

DNA plasmids (pMTB-ef1-H2B-mCherry2 gifted by D. Wagner, UCSF) were extracted and purified with a QIAprep Spin Miniprep Kit (27104). Purified DNA was diluted to 100 ng l$^{-1}$ in water (Invitrogen UltraPure DNase/RNase-free distilled water, 10977015) and stored at −20 °C. Zebrafish *Tol2* mRNA was prepared using pT3TS-Tol2 plasmids shared by the Ekker lab[87]. pT3TS-Tol2 plasmid DNA was linearized by the restriction enzyme BamHI digestion. Linearized plasmids were gel-purified with a QIAquick Gel Extraction Kit (28704). The 5′-capped mRNAs were synthesized using Invitrogen mMESSAGE mMACHINE T3 Transcription Kit and purified by lithium included in the kit (Thermo Fisher, AM1348). The purified RNA was dried and dissolved with 20 l DNase/RNase-free water, diluted to 200 ng l$^{-1}$ and saved as stock at −70 °C. Before the experiment, the stocked DNA and RNA were thawed, and 1 μl DNA plus 1 μl RNA were mixed with 3 μl of DNase/RNase-free water containing 0.05% phenyl red. For each microinjection, a 4-nl mixture was injected into a single cell of a 4–8-cell stage embryo from the Tg(ef1α:H2B-mNeonGreen) strain. All injections were done with an injector (WPI PV830 Pneumatic Pico Pump) and a micromanipulator (Narishige, Tokyo, Japan). Around 50 to 100 embryos were injected for each experiment. The injected embryos are ready for mounting and imaging at approximately 4.5 hpf.

### Sample preparation for single-objective light-sheet microscopy (DaXi)

Embryos are first dechorionated and then gently embedded into a 1% solution of low-gelling-temperature agarose (Sigma, A0701). The samples are then transferred in a glass-bottom cell culture dish (35-mm cell culture dish with glass bottom 20 mm, Stellar Scientific) and positioned at the correct imaging angle using a custom-made capillary. When solid, the agarose surrounding the tail was cut off and removed using a dissection knife and forceps to permit full development and tail elongation. During the time lapses, the embryos are placed in an embryo medium solution with 0.016% tricaine to prevent embryo movement.

### QPI and virtual staining

The A549 cells were cultured at 37 °C and 5% CO$_2$ and maintained between 20% and 90% confluency. The cells were seeded on a 12-well glass-bottom plate (Cellvis, P12-1.5H-N) and imaged using a widefield microscope (Leica Dmi8). The quantitative phase volumes are virtually stained using the VSCyto2D model to highlight landmark channel organelles such as the nuclei and cell plasma[58].

### Simultaneous sparse and ubiquitous labels imaging

Four hours after microinjection, injected embryos (~4 hpf) were screened using a Thunder Imager Model Organism (Leica Microsystems). An embryo with 20–30% H2B-mCherry2-expressing cells (compared to ubiquitous H2B-mNeonGreen cells) was selected and transferred to a glass-bottom petri dish (35-mm cell culture dish with glass bottom 20 mm, Stellar Scientific) containing 0.3× Danieau's embryo medium. The embryo's chorion was removed using sharp forceps under the same microscope. The dechorionated embryo was then gently transferred to a petri dish containing 0.1% low-temperature melting agarose (Sigma, A0701) using a glass pipette. Subsequently, the embryo was transferred to FEP tubes (Valco-TFEP130; outer diameter, 1/16 inch; internal diameter, 0.03 inch; Vici Metronics) with agarose via a micropipette through a 200-μl tip (Eppendorf). Both ends of the FEP tube were sealed with 2% low-temperature melting agarose to secure the embryo. The tube was placed in the chamber of an adaptive multi-view multicolor light-sheet platform[9] using a customized holder insertion tool filled with low-toxicity silicone (Kwik-Sil, World Precision Instrument). The mounting system was similar to that previously described in Zebrahub[9], but with an upside-down mounting direction. The chamber was filled with embryo water, which was circulated with 0.016% tricaine during imaging. For whole-embryo imaging, the multicolor-interleaved mode was used with an exposure time of 20 ms per channel with the OpenSimView[9] microscope. The imaging volume comprised 493 $z$-planes with 1.625-μm $z$-steps. Images were captured at 90-s intervals, starting at the shield stage (6 hpf), and ending after 24 hpf.

### DaXi microscope parameters

We used our DaXi imaging platform[6] to perform high-resolution tail development imaging (1.0 NA). We use two orthogonal, oblique light sheets for each time point to improve the image field of view coverage. During imaging, the exposure was set to 20 ms, the $z$-step to 1.24 ms, the imaging volume was adapted to each sample and the time step was around 60 s. Time-lapse sessions are about 10-h long and usually cover a developmental window from 1 or 2 somites to 27 somites.

### Software

The Ultrack package, including its REST API, and additional image analysis routine and plotting were implemented using `Blosc`, `CuPy`, `FastAPI`, `imagecodecs`, `iohub`, `inTRACKtive`, `httpx`, `ggplot2`, `Napari`, `networkx`, `Numba`, `NumPy`, `Pandas`, `PyTorch`, `scikit-image`, `SciPy`, `SLURM`, `tifffile`, `traccuracy`, `uvicorn`, `websockets` and `Zarr`[36,63,74,79,80,88–95].

## Datasets

Extended Data Table 3 presents the datasets evaluated in this paper. While not a requirement, some images of the CTC were upsampled for improved accuracy.

## Light-sheet imaging data preprocessing

Upon acquisition, the imaging data are converted to the OME-Zarr format[96,97] using the `iohub`[89] Python package. Next, the raw multi-view images are fused. Phase cross-correlation techniques are applied to correct inter-frame movements to ensure stability throughout the time-lapse sequence. In cases where deep-learning methodologies are utilized, image normalization is performed, scaling pixel intensities to the range between 0 and 1 using user-specified lower and upper quantiles of 0.005 and 0.99999, respectively, and their background is subtracted using the `subtract_background` function from `dexpv2` with $\sigma = 20$, which approximates a white tophat transform using morphological reconstruction by dilation. These preprocessing routines and other image processing operations are implemented and accessible via https://github.com/royerlab/dexpv2/.

## GPU-accelerated flow-field estimation

Flow-field estimation of large 3D volumes can be extremely challenging and computationally intensive, often following particle image velocimetry strategies that estimate local translation using phase cross-correlation for small patches of the data[61]. Instead of maximizing cross-correlation, which minimizes the mean-squared error between frames given a field in the coordinate space, we use gradient descent to optimize the coordinate space directly. This choice, justified by its enhanced computational efficiency and simplicity, allows for the integration of any differentiable function within the coordinate space. The loss function is defined according to equation (1):

$$L(\theta) = \| I_{t-1} - \text{grid\_sample}(I_t, C + \theta) \|_1 + \text{TV}(\theta) \tag{1}$$

where $I_t$ denotes the frame at time $t$, $C$ represents the identity coordinates (that is, no movement), $\theta$ indicates the movement flow field we optimize, and TV denotes the L1 total variation loss on the flow field. The `grid_sample` function, a differentiable data sampling method from PyTorch[63], enables this process. This routine is tailored for specialized GPU hardware, greatly enhancing processing speed. The $\theta$ optimization is performed for a fixed number of iterations over multiple image resolutions, starting from the lowest resolution, which is then used to initialize the next $\theta$ optimization stage. For our experiment (Extended Data Fig. 2), the gradient descent learning rate was set to $1 \times 10^{-2}$, the number of scales to 2 and the iterations to 2,000. A detailed Jupyter notebook is available at https://github.com/royerlab/ultrack_supplementary/tree/main/configuration/flow_registration/.

## Label conversion to foreground and contours map

The conversion of multiple segmentation labels into foreground and contour maps involves two main processes. For the foreground map generation: (i) each label is transformed into a binary map, with nonzero pixels assigned a value of one (foreground) and zero pixels remaining as background; (ii) these individual maps are merged into a single foreground map by taking the maximum value across corresponding pixels from all individual foreground maps; (iii) this logical OR operation ensures that any pixel belonging to at least one segment is included in the final foreground map. For the contour map generation: (i) to preserve each label and its dual contour, we process each label map to create a binary contour map, where pixels are set to one if their neighboring pixels have differing labels, indicating a boundary; (ii) the final contour map is then created by averaging these individual binary contour maps—summing would yield the same results because the hierarchical segmentation reconstruction is based on ordering of

values and not their absolute values, the benefit of averaging is to keep pixel values within the 0 to 1 range.

## Foreground and contours prediction with neural networks

We also implemented a deep-learning approach to generate foreground and contour maps, which we used for both the CTC submissions (where 3D segmentation annotations were available) and the large zebrafish dataset (Fig. 5). Our implementation uses a U-Net architecture[70] that processes single-channel grayscale images to produce two-channel outputs: foreground prediction and contour prediction, each with its sigmoid activation layer.

The network architecture consists of four stages of encoding and decoding, with channel depths of 32, 64, 128 and 256, using $5 \times 5$ convolution kernels. During decoding, we upsample intermediate representations using linear interpolation and combine them with corresponding encoder features via skip connections. Network training minimizes a dice loss[71], supplemented with auxiliary supervision at three intermediate resolutions during decoding, following the approach detailed in ref. 72.

All network weights and architectural details are publicly available at our supplementary repository (https://github.com/royerlab/ultrack_supplementary/), with the complete implementation available at our CTC submission repository (https://github.com/royerlab/ultrack_CTC_submission/).

## Candidate hierarchical (nested) segmentation

The goal of hierarchical segmentation is to compute a set of nested partitions (that is, segmentations) from a multilevel contour map, where each level corresponds to a different segmentation threshold. While a naive approach resorts to binarizing the contour map at each level and computing the connected components of the binary images, Ultrack uses a more sophisticated and computationally efficient method: the hierarchical watershed algorithm[43,98]. This algorithm represents the image as a graph $G = (V, E)$, where $V$ are the pixels or voxels, and $E$ denotes the edges connecting adjacent voxels (four neighbors for 2D and six neighbors for 3D). It defines a non-decreasing order of edge weights as shown in equation (2):

$$w(p, q) = \frac{I(p) + I(q)}{2} \tag{2}$$

where $p, q \in E$ and $I(p)$ is the image intensity at a pixel $p$ of a contour map. The algorithm then recovers nested segmentations using a region-merging approach based on Kruskal's minimum-spanning tree[99] and Tarjan's union-find algorithms[100]. The process begins with an initial segmentation where each pixel forms its own segment. It then iteratively: (i) selects the minimum unseen edge $(p, q)$; (ii) merges the segments of $p$ and $q$ if they are not already in the same segment; (iii) continues until all edges have been processed. This segmentation process operates within each connected component of the binary foreground map, allowing the algorithm to ignore background regions and thereby greatly speeding up the process in non-dense labeling (for example, nuclei). For further details on hierarchical segmentation processing and implementation, see ref. 45.

## Candidate segmentation association

For each candidate segmentation, we compute the potential temporal associations (that is, tracks). Let $\mathcal{H}$ represent the set of all segmentations derived from hierarchical segmentations. We define $E^T$ as the set of pairs representing potential associations between these segmentations, with $w(\cdot, \cdot)$ denoting their respective weights. The association process involves the following steps: (i) For each segmentation $i \in \mathcal{H}$, where $t$ denotes the time point of segment $i$, we compute its candidates $2k$-nearest neighbors within a predefined radius using their centroid coordinates at time $t - 1$; (ii) Optional step for additional filtering of

neighbors, for example, color-feature association section. (iii) The association score between every segment $i$ and their candidate neighbor $j$ is calculated as $w(i,j) = \text{IoU}(i,j)^\gamma$. Where

$$\text{IoU}(i,j)^\gamma = \left(\frac{S_i \cup S_j}{S_i \cap S_j}\right)^\gamma$$

and $S_i$ is the binary mask of region $i$ (that is, segmentation hypothesis) and $\gamma$ is consistently set to four throughout our study; (iv) Include into $E^T$ the $k$ pairs per segment in $t$ with the largest IoU, using their distance as a tie-breaker.

## Segmentation selection and tracking ILP

Given $\mathcal{H}$ and $E^T$, as defined previously, we compute the joint segmentation and tracking by solving an ILP optimization problem. This ILP framework aims to find the optimal pairing between segments that maximizes their association while adhering to biological constraints. The ILP is formulated as follows in equations (3)–(9):

$$\arg\max_{\mathbf{x}} = \sum_{i,j \in E^T} w_{ij} x_{ij} + \\ \sum_{j \in \mathcal{H}} w_\alpha x_{\alpha j} + w_\beta x_{j\beta} + w_\delta x_{\delta j} \tag{3}$$

$$\text{s.t.} \quad y_j = x_{\alpha j} + \sum_{i \in H_{t-1} | i,j \in E^T} x_{ij} \; \forall j \in \mathcal{H} \tag{4}$$

$$y_i + x_{\delta i} = x_{i\beta} + \sum_{j \in H_{t+1} | i,j \in E^T} x_{ij} \; \forall i \in \mathcal{H} \tag{5}$$

$$y_i \geq x_{\delta i} \; \forall \, i \in \mathcal{H} \tag{6}$$

$$y_i + y_j \leq 1 \quad \forall \, i,j \in \mathcal{H} \,|\, i \subset j \tag{7}$$

$$y_i, x_{\alpha i}, x_{i\beta}, x_{\delta i} \in \{0,1\} \; \forall \, i \in \mathcal{H} \tag{8}$$

$$x_{ij} \in \{0,1\} \; \forall \, i,j \in E^T \tag{9}$$

In this formulation, $y_i$ indicates whether a segment is selected, and $x_{ij}$ indicates the selected associations, $x_{\alpha i}, x_{i\beta}$ and $x_{\delta i}$ are slack variables indicating an appearing, disappearing and dividing cell, respectively. Equation (3) is the objective function to maximize the association between segmentations. The first term in the objective function represents the weight of segmentation associations, whereas the subsequent terms account for appearance, disappearance and division events, which are user-defined non-positive penalization weights. Equations (4) and (5) ensure flow conservation, stipulating that an appearance, division or incoming association must correspond to at least an outgoing association or a cell disappearance. Equation (6) restricts cell division to preexisting cells, and equation (7) prevents selecting overlapping segmentations, thereby enforcing that each pixel is uniquely selected.

## Tracking evaluation metrics

The evaluation scores were calculated using official binaries from the CTC for datasets submitted to the competition. Additional calculations were performed using the `traccuracy` Python package[101]. The TRA metric quantifies the cost-effectiveness of editing a predicted lineage into the reference lineage, divided by the cost of constructing the lineage from scratch, $\text{AOGM}_0$. This is mathematically defined according to equations (10) and (11):

$$\text{AOGM} = w_{\text{NS}} n_{\text{NS}} + w_{\text{FN}} n_{\text{FN}} + w_{\text{FP}} n_{\text{FP}} \\ + w_{\text{ED}} n_{\text{ED}} + w_{\text{EA}} n_{\text{EA}} + w_{\text{EC}} n_{\text{EC}} \tag{10}$$

$$\text{TRA} = 1 - \frac{\min(\text{AOGM}, \text{AOGM}_0)}{\text{AOGM}_0} \tag{11}$$

where $n_{\text{NS}}$ represents the number of node splits (that is, undersegmented cells), $n_{\text{FN}}$ and $n_{\text{FP}}$ are false-negative and false-positive nodes respectively, $n_{\text{ED}}$ and $n_{\text{EA}}$ denote false-positive and false-negative edges, and $n_{\text{EC}}$ represents nodes with incorrect semantics. The weights are set as follows: $w_{\text{NS}} = 5$, $w_{\text{FN}} = 10$, $w_{\text{FP}} = 1$, $w_{\text{ED}} = 1$, $w_{\text{EA}} = 1.5$ and $w_{\text{EC}} = 1$, emphasizing the greater complexity in correcting missing units compared to deleting extra ones. For further details, refer to Matula et al.[49]. The combined CTC score averages the TRA and SEG scores. The SEG score calculates the average intersection over union (Jaccard index) across all segmentation instances. A ground-truth ($gt$) segmentation instance, $G_j$, is paired to a predicted instance $P_i$, as in equation (12), if:

$$|P_i \cup G_j| > 0.5|G_j| \tag{12}$$

This criterion ensures that the overlap between the predicted and $gt$ instances is more than half the size of the $gt$ instance, guaranteeing a single match for each $gt$ instance. The same criterion matches nodes in the AOGM metric calculation.

From the segmentation detections obtained through the CTC procedure, we also computed the F1 score, where

$$\text{F1 score} = \frac{2n_{\text{TP}}}{2n_{\text{TP}} + n_{\text{FN}} + n_{\text{FP}}}$$

which provides a less-biased assessment of performance[102] that combines the evaluation of both $n_{\text{FN}}$ and $n_{\text{FP}}$.

In the multicolor and label-free cell culture experiments where not all lineages are annotated, we applied the recommended CTC protocol, where the first frame of the ground truth is provided to indicate which lineages should be kept, all the remaining lineages are removed, and the CTC metric is computed between this subset of lineages and the annotated data.

## Zebrafish sparse tracking evaluation metrics

The sparse tracking data evaluation uses implementation from the Python package `linajea`[24]. The metrics are the proportion of different types of mistakes over the total number of edges (that is, connections between cells of adjacent frames) of the ground-truth data. Because the data are sparse, only a subset of metrics are valid estimates for the remaining unlabeled data:

- FN: Proportion of false negative (that is, missing edges);
- IS: Proportion of identity switches (that is, edges associating incorrect cells);
- FP-D: Proportion of false-positive divisions (that is, fake division) within a one-frame tolerance;
- FN-D: Proportion of false-negative divisions (that is, missing division) within a one-frame tolerance;
- Sum: the sum of all the above.

## Color features association

The image intensity information can be used to filter out segments between adjacent frames that are unlikely to represent the same cell. To do that, we compute a normalized distance between the average intensities of segments, akin to a standard normal distribution $z$-score; if this distance exceeds a predefined threshold $z$, their association is excluded from the ILP formulation. Specifically, the relationship must satisfy the following conditions as in equation (13):

$$\max \left|\frac{\mu_i - \mu_j}{\sigma_j}\right| \leq z \tag{13}$$

where $\mu_j$ is the mean intensity vector within the mask of a segment $S_j$ (represented as set) at time $t$ and $\sigma_j$ is the standard deviation. The operations are done element-wise per channel, and max results in a scalar from it, here, compared to a neighboring segment $S_i$ at $t-1$. Therefore, $j$ is the reference node and $i$ is the candidate neighbor node. We formalized the elements in the color vectors according to equations (14) and (15):

$$\mu_i = \frac{1}{|S_i|} \sum_{p \in S_i} I_p \qquad (14)$$

$$\sigma_i = \sqrt{\frac{1}{|S_i|} \sum_{p \in S_i} (I_p - \mu_i)^2} \qquad (15)$$

where $I_p$ is the intensity values at voxel $p$, such that for a three-channeled image $I_p$ is a 3D vector.

The motivation behind this formulation is the need for an intensity-invariant distance between colored cells. Hence, using a $z$-score formulation was a natural choice because the threshold $z$ can be interpreted as how many standard deviations that a color distribution within a cell differs from other distributions of cells. The use of max is a conservative choice, so that if any channel independently exceeds the threshold $z$, the candidate link is rejected.

This can be efficiently computed by making the pairwise comparison only between the $2k$ candidate pairs of each segment because the default linking already excluded the rest, thus reducing the $N_{t-1} \times N_t$ comparisons to $N_t \times 2k$, where $k$ is the number of candidate links defined at the linking step.

The threshold $z$ was set to 3.0 throughout the multicolor experiments, a symmetric quantile threshold equivalent to a 0.0027 probability under our $z$-score estimation.

### Fiji plugin and Python interoperability
Fiji[35] is a Java-based software that can be extended through plugins developed in the same language. To integrate Ultrack with Fiji, we developed a RESTful API[103] using the FastAPI framework[91], facilitating communication between Java and Python. This API uses websockets[104] for persistent communication, enabling real-time event logging and output streaming from Ultrack to Fiji. We created a GUI in Java that serves as a Fiji plugin, leveraging the API provided with the Ultrack Python package. This interface grants access to features and functionalities of both systems. The GUI components and user experience are implemented using the Bootstrap 5 (ref. [105]) front-end framework. To promote broader interoperability, Ultrack adheres to API specifications detailed in our documentation (https://royerlab.github.io/ultrack/rest_api.html), allowing integration with various programming languages beyond Java and Python.

### Napari plugin
Napari[36] is a Python library for $n$-dimensional image visualization, annotation and analysis. It features a GUI built with Qt[106], which is extensible with Qt primitives in Python. We developed a custom plugin for Napari integration that still scales with large datasets. In addition to the tracking standard workflow, the plugin offers basic preprocessing operations to fit most users' needs, which can benefit from GPU speedups, if any are available. Ultrack's parameters can be tuned through the GUI. The processing runs in the background in a separate thread and, consequently, the intermediate results can be visualized using the native Napari viewer. By the tracking execution finish, the results can be exported to several formats, such as TrackMate[37], NetworkX graph[79] and Zarr array.

### Reporting summary
Further information on research design is available in the Nature Portfolio Reporting Summary linked to this article.

## Data availability
The imaging data and network weights are accessible at https://public.czbiohub.org/royerlab/ultrack/ and https://public.czbiohub.org/royerlab/zebrahub/imaging/single-objective/.

## Code availability
The repository for the Ultrack Python package and its updated versions are available at https://github.com/royerlab/ultrack/. Additional supplementary scripts are available at https://github.com/royerlab/ultrack_supplementary/. The Fiji plugin can be found at https://github.com/royerlab/ultrack-imagej/.

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

## Acknowledgements

We extend our gratitude to D. Wagner and J. Huisken for generously sharing their zebrafish lines. We thank A. Abbas and P. Swoboda for their insightful discussions on Integer Linear Programming and optimization. We are grateful to B. Moretti, V. Pfannenstill and J. Ko for their valuable feedback and for being early adopters of Ultrack. We appreciate I. Sbalzarini and S. Schmid for their mentorship and constructive feedback on the paper. A.X.F. and I.T. were funded by CNPq 304711/2023-3 and FAPESP 2022/16491-2, 2022/07877-4 and 2023/14427-8. J.L. was funded by National Institutes of Health awards R01GM137605 and R35GM153257. This work was funded by the Chan Zuckerberg Biohub San Francisco (CZB SF). We are deeply grateful to CZB SF donors P. Chan and M. Zuckerberg for their generous support, which made this research possible.

## Author contributions

J.B., M.L. and L.A.R. conceived the research. J.B. wrote the first version of the Ultrack software, performed the computational experiments and curated the data except for neuromast and label-free images. I.T. wrote the user interfaces, implemented the REST backend, maintained the software and trained the model for the sparse/dense-labeled embryo. X.Z. prepared and imaged the sparse/dense-labeled embryo. M.L. prepared and imaged the zebrahub embryo. T.L. prepared and imaged the neuromast. T.A.P.M.H. and A.B. analyzed the neuromast data. A.J. and A.B. curated the neuromast data. E.H.-M. curated and analyzed the label-free data. S.X. provided support for the microscopy. R.A. prepared and imaged the multicolor data. J.B., I.T. and L.A.R. wrote the paper. J.L., S.M., A.X.F. and A.J. provided guidance. L.A.R. supervised the research. All authors contributed to editing the paper.

## Competing interests

The authors declare no competing interests.

## Additional information

**Extended data** is available for this paper at https://doi.org/10.1038/s41592-025-02778-0.

**Correspondence and requests for materials** should be addressed to Jordão Bragantini or Loïc A. Royer.

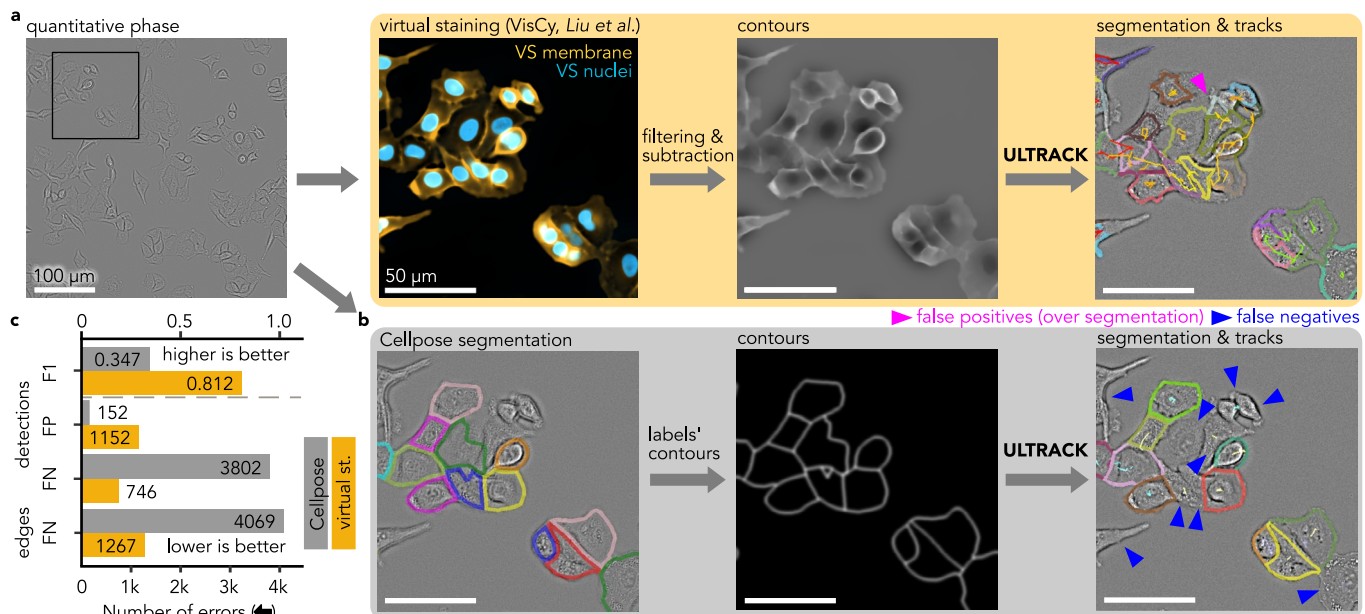

**Extended Data Fig. 1 | Ultrack improves cell tracking in label-free imaging by leveraging virtual staining. a**, Pipeline for intensity-based joint segmentation and tracking from label-free quantitative phase imaging: Quantitative phase image (top left) is processed by VisCy[58] to generate virtual staining of nuclei and membrane (second from left). These virtual stains are used to derive contours (third from left) and foreground maps (not shown) through filtering, subtraction, and thresholding. Ultrack uses these inputs to perform tracking (right). **b**, We compare the above against first applying a Cellpose segmentation to the quantitative phase image (bottom left), then deriving contours (and foreground map, not shown), and then applying Ultrack (right). False positives (over-segmentation) are indicated in magenta, and false negatives in blue. **c**, Quantitative comparison of tracking errors using Cellpose[59] segmentations (gray) versus virtually stained membranes (orange) as input. The bar chart shows the F1-score of detections, upper y-axis and the number of false negative (FN) edges, false negative (FN) detections, and false positive (FP) detections for each method, lower y-axis, from 247 tracklets spanning 48 frames.

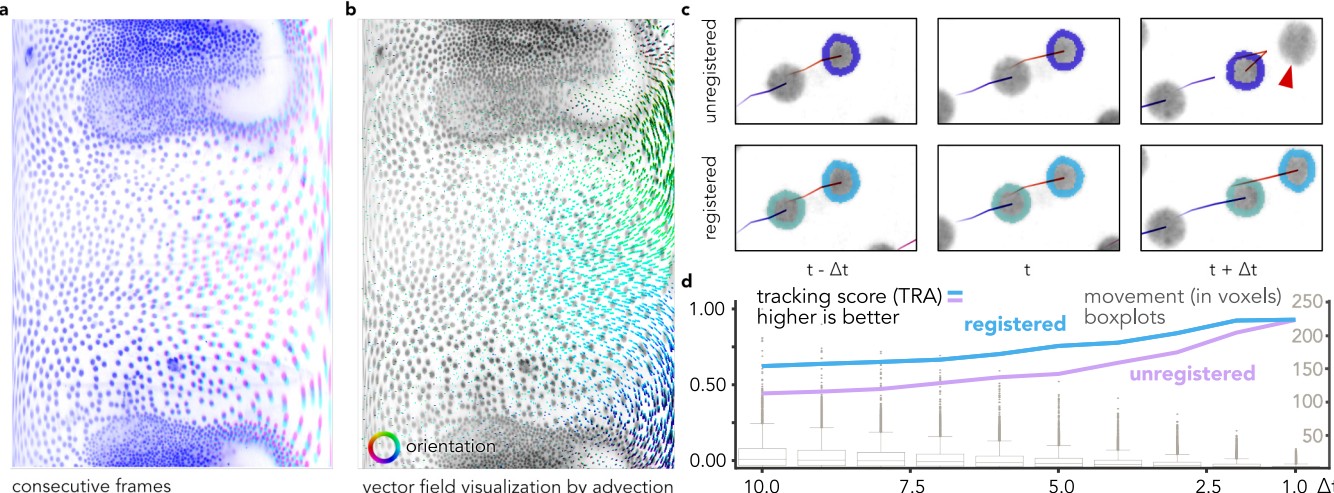

**Extended Data Fig. 2 | Enhancing tracking accuracy through temporal registration. a**, Consecutive frames, t (magenta) and t + 1 (cyan), highlighting divergent motion along the right boundary. **b**, Vector field generated by non-linear registration, colored by orientation. **c**, Qualitative comparison of cell identity preservation: top row shows a cell identity switch in unregistered frames, identity switch indicated by red arrow, bottom row demonstrates improved consistency with flow registration. **d**, Comparative analysis of tracking accuracy for registered and unregistered data across different cell velocities ($\Delta t$), illustrating enhanced performance with registration. The left y-axis corresponds to the colored lines showing TRA scores, while the right y-axis and boxplots show cell movement distributions between adjacent frames in voxels. At $\Delta t = 10.0$, some cells move more than 150 voxels between frames - a substantial distance given the median cell diameter of approximately 19.64 voxels. Boxplots followed Tukey's original definition[107] using the median for the center line, first (q1) and third quantile (q3) for the box limits and whiskers at q1 - 1.5IQR and q3 + 1.5IQR, where IQR = q3 - q1. Considering a total population size of 100915 and an average size of 10091.5 per group ($\Delta t$).

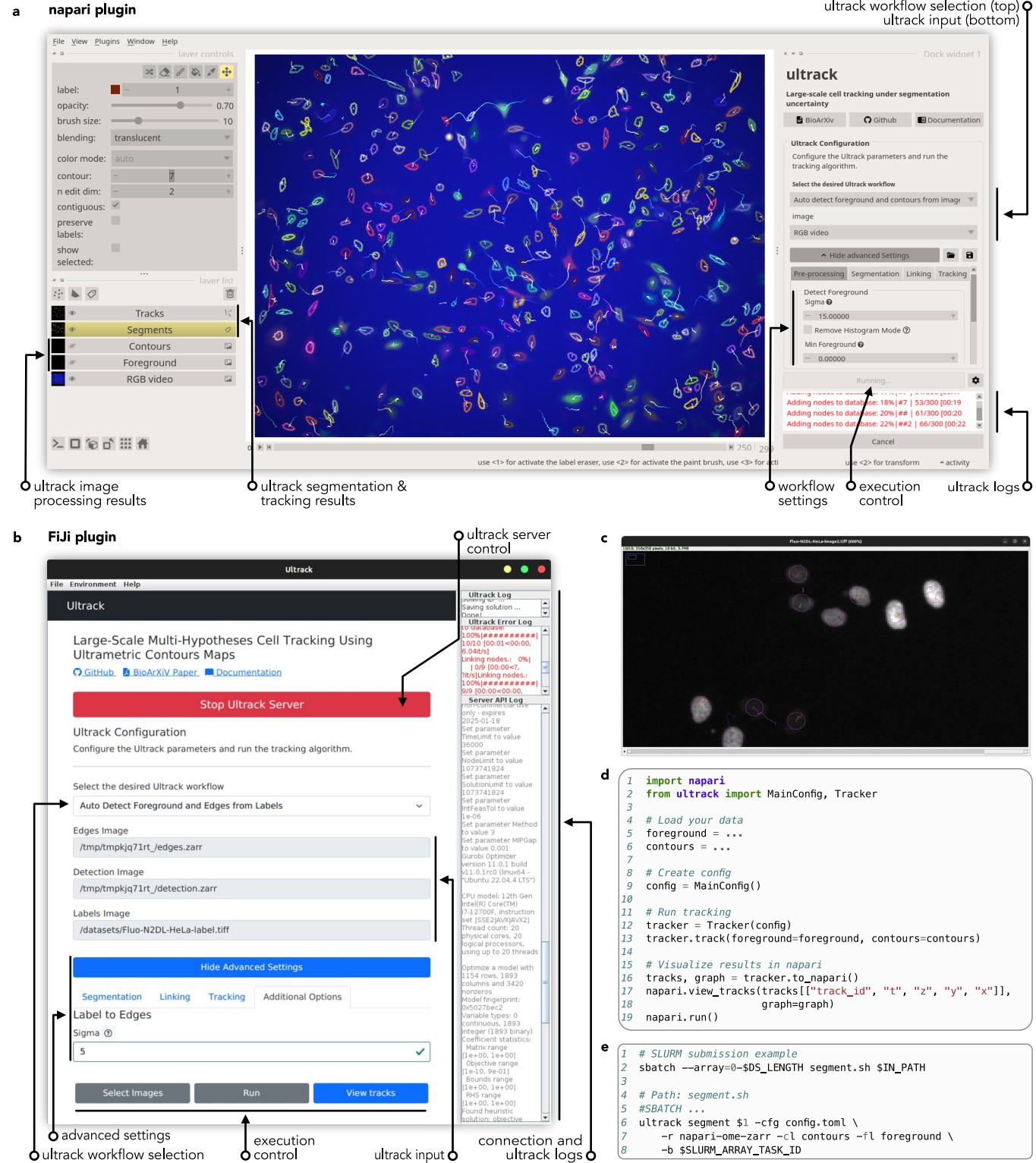

**Extended Data Fig. 3 | Ultrack's multi-interface approach enables cell tracking for diverse user needs and computational environments. a**, Napari plugin interface showcasing multi-color automatic cell tracking results. The interface includes workflow selection, input configuration, image processing results, segmentation and tracking visualization, workflow settings, and execution control. **b**, Fiji plugin interface guiding users through workflow selection, input specification, and advanced settings. It also provides execution control and displays Ultrack server logs. **c**, Example of image data overlaid with cell detections and tracks as viewed in the Fiji interface. **d**, A concise Python API example (18 lines) demonstrating data loading, configuration, tracking execution, and result visualization in napari. **e**, Sample SLURM submission script for Ultrack's CLI, illustrating its integration with high-performance computing environments for distributed processing.

**Extended Data Table 1 | Segmentation and tracking scores for 3D datasets of the Cell Tracking Challenge**

| Kind | Dataset | Rank | combined score (CTB) | tracking score (TRA) | segmentation score (SEG) |
|---|---|---|---|---|---|
| Whole 3D embryo | Worm *C. elegans* | 1st | **0.844**ultrack | 0.987MPI | 0.759MU |
| | | 2nd | 0.829KTH | 0.979JAN | 0.729KIT |
| | | 3rd | 0.808KIT | 0.975IGFL | 0.722KTH |
| | | 4th | - | **0.967**ultrack | **0.722**ultrack |
| | Fly *D. melanogaster* | 1st | **0.708**ultrack | **0.802**ultrack | **0.613**ultrack |
| | | 2nd | 0.617KTH | 0.785JAN | 0.613KTH |
| | | 3rd | 0.591JAN | 0.785MPI | 0.397JAN |
| | Beetle *T. castaneum* | 1st | **0.841**ultrack | 0.955MPI | **0.746**ultrack |
| | | 2nd | 0.804MPI | **0.936**ultrack | 0.684RWTH |
| | | 3rd | 0.785RWTH | 0.886RWTH | 0.654MPI |
| Embryo projection | Beetle projection *T. castaneum* | 1st | 0.867KTH | 0.952MPI | 0.791KTH |
| | | 2nd | 0.816MPI | 0.942KTH | 0.766KIT |
| | | 3rd | 0.787KIT | **0.854**ultrack | 0.680MPI |
| | | 4th | **0.754**ultrack | - | **0.654**ultrack |
| Cells in a dish | Breast cancer cells | 1st | 0.797KIT | 0.884KIT | 0.710KIT |
| | | 2nd | 0.761LEID | 0.882KTH | 0.642LEID |
| | | 3rd | 0.757KTH | 0.880LEID | 0.632KTH |
| | | 7th | **0.683**ultrack | **0.818**ultrack | **0.549**ultrack |

Ultrack results, as of August 2024, are highlighted in bold. Ultrack obtains the highest combined segmentation and tracking score (CTB) for three out of five datasets. Ultrack achieved 1st place more often than any other method, excelling for large and complex 3D whole-embryo cell tracking datasets. See Extended Data Table 2 for competing methods' descriptions.

**Extended Data Table 2 | Description of leading methods in the Cell Tracking Challenge**

| Label | Reference | Description |
|-------|-----------|-------------|
| KTH | 108 | Uses the Gaussian Mixture Probability Hypothesis Density (GM-PHD)[109] filter to detect and estimate the motions of centroids with the Viterbi algorithm to optimally link segmentations across frames while assigning different cell states (*e.g.* migration, mitosis, appearance, disappearance). |
| KIT | 23 | *EmbedTrack* combines pixel-embedding segmentation with motion estimation for joint segmentation and tracking by embedding each pixel in a centroid-offset space and incorporating optical flow. |
| MPI | (unpublished) | Although a detailed description is undisclosed, the source code provided by the challenge organizers suggests a deep-learning-based centroid detection strategy followed by nearest-neighbor linking. |
| JAN | 110, 24 | Builds upon *Linajea*[24],which predicts cell locations as Gaussians' maxima and their motion offset to their position on the previous frame, by classifying cell states (parent, daughter, continuing cells), improving their ILP formulation to predict these states, while asserting that parent cells can only link to daughter cells. A structured SVM is used to learn optimal ILP weights, enhancing automation linking cell trajectories. |
| LEID | 111 | Joint segmentation and tracking by model evolution with level sets and specialized routines for cell division and cell appearances. |
| MU | 112 | Integrates H-minima watershed segmentation with a U-Net model for cell contour and foreground prediction. Subsequent tracking uses an intersection-based heuristic, which pairs cells across frames if they share sufficient overlap. |
| IGFL | 22 | *ELEPHANT* leverages deep-learning-based cell detection, coupled with predicted flow fields (*i.e.* motion offset) for cell advection and linking. This flow-driven approach resembles the MPI and JAN strategy, using learned motion information to guide inter-frame associations. |
| RWTH | 113 | Detects cells via a Laplacian of Gaussian filter and applies nearest-neighbor linking in subsequent frames; detections are then used as seeds for segmentation. Manual curation post-processing is performed to rectify missed detections. |

Scores shown in Extended Data Table 1. Cell tracking challenge leading methods references: KTH[108,109], KIT[23], JAN[24,110], LEID[111], MU[112], IGFL[22], RWTH[113].

**Extended Data Table 3 | List and references of datasets**

| Dataset | T, C, Z, Y, X | Size (GB) | Usage | Ref. |
|---|---|---|---|---|
| YFP-TIA-1 (CTC) | 30, 1, 1, 1024, 1024 | 0.063* | Fig. 2 | 48 |
| MDA-MB-231 | 300, 3, 1, 1440, 1920 | 2.5 | Fig. 3 | this work |
| Label-free | 48, 1, 8, 2046, 2003 | 5.85† | Ext. Data Fig. 1 | 58 |
| Beetle proj. (CTC) | 135, 13, 2447, 1719 | 14.76* | Ext. Data Fig. 2 | 64 |
| Worm (CTC) | 165, 31, 512, 712 | 3.73* | Ext. Data Tab. 1 | 68 |
| Fly (CTC) | 50, 125, 603, 1272 | 9.59* | Ext. Data Tab. 1 | 31 |
| Beetle (CTC) | 105, 983, 1846, 983 | 376* | Ext. Data Tab. 1 | 64 |
| Sparse zebrafish | 522, 2, 505, 2217, 2170 | 5072 | Fig. 4 | this work |
| Late zebrafish | 791, 1, 448, 2174, 2423 | 3733 | Fig. 5 | 9 |
| Neuromast | 500, 1, 73, 1024, 1024 | 71.3 | Fig. 6 | this work |

†: size before 3D to 2D virtual staining. * indicates averaged size and shape. The size presented in Fig. 1f is regarding the input contour map, which can include projections and upsampling.

# Reporting Summary

## Statistics

For all statistical analyses, confirm that the following items are present in the figure legend, table legend, main text, or Methods section.

| n/a | Confirmed | |
|---|---|---|
| ☐ | ☒ | The exact sample size (*n*) for each experimental group/condition, given as a discrete number and unit of measurement |
| ☒ | ☐ | A statement on whether measurements were taken from distinct samples or whether the same sample was measured repeatedly |
| ☒ | ☐ | The statistical test(s) used AND whether they are one- or two-sided<br>*Only common tests should be described solely by name; describe more complex techniques in the Methods section.* |
| ☒ | ☐ | A description of all covariates tested |
| ☒ | ☐ | A description of any assumptions or corrections, such as tests of normality and adjustment for multiple comparisons |
| ☐ | ☒ | A full description of the statistical parameters including central tendency (e.g. means) or other basic estimates (e.g. regression coefficient) AND variation (e.g. standard deviation) or associated estimates of uncertainty (e.g. confidence intervals) |
| ☒ | ☐ | For null hypothesis testing, the test statistic (e.g. *F*, *t*, *r*) with confidence intervals, effect sizes, degrees of freedom and *P* value noted<br>*Give P values as exact values whenever suitable.* |
| ☒ | ☐ | For Bayesian analysis, information on the choice of priors and Markov chain Monte Carlo settings |
| ☒ | ☐ | For hierarchical and complex designs, identification of the appropriate level for tests and full reporting of outcomes |
| ☒ | ☐ | Estimates of effect sizes (e.g. Cohen's *d*, Pearson's *r*), indicating how they were calculated |

*Our web collection on statistics for biologists contains articles on many of the points above.*

## Software and code

Policy information about availability of computer code

| Data collection | Images were collected as indicated in the material and methods. |
|---|---|
| Data analysis | The online repositories https://github.com/royerlab/ultrack and https://github.com/royerlab/ultrack_supplementary contains the codes, docker images and python environment files used for the data analysis in the paper. |

For manuscripts utilizing custom algorithms or software that are central to the research but not yet described in published literature, software must be made available to editors and reviewers. We strongly encourage code deposition in a community repository (e.g. GitHub). See the Nature Portfolio guidelines for submitting code & software for further information.

## Data

Policy information about availability of data

All manuscripts must include a data availability statement. This statement should provide the following information, where applicable:

- Accession codes, unique identifiers, or web links for publicly available datasets
- A description of any restrictions on data availability
- For clinical datasets or third party data, please ensure that the statement adheres to our policy

Data is available at https://public.czbiohub.org/royerlab/ultrack and https://public.czbiohub.org/royerlab/zebrahub/imaging/single-objective

# Research involving human participants, their data, or biological material

Policy information about studies with human participants or human data. See also policy information about sex, gender (identity/presentation), and sexual orientation and race, ethnicity and racism.

| | |
|---|---|
| Reporting on sex and gender | N/A |
| Reporting on race, ethnicity, or other socially relevant groupings | N/A |
| Population characteristics | N/A |
| Recruitment | n?A |
| Ethics oversight | N/A |

Note that full information on the approval of the study protocol must also be provided in the manuscript.

# Field-specific reporting

Please select the one below that is the best fit for your research. If you are not sure, read the appropriate sections before making your selection.

☒ Life sciences  ☐ Behavioural & social sciences  ☐ Ecological, evolutionary & environmental sciences

For a reference copy of the document with all sections, see nature.com/documents/nr-reporting-summary-flat.pdf

# Life sciences study design

All studies must disclose on these points even when the disclosure is negative.

| | |
|---|---|
| Sample size | No measure was taken to estimate the sample size. The sample size was determined by the number of cells identified on their respective imaging data. Representing a typical imaging dataset of biological phenomena. |
| Data exclusions | N/A |
| Replication | Unless specified, all experiments one done once. We are evaluating the performance of an exact computational algorithm and not investing a biological phenomena. |
| Randomization | Ground-truth tracks were annotated completely (whole population no sampling required) or using their respective policy for tracks selection. |
| Blinding | Analyses and evaluation were performed following standardized algorithms for accuracy assessment so no blinding was not necessary. |

# Behavioural & social sciences study design

All studies must disclose on these points even when the disclosure is negative.

| | |
|---|---|
| Study description | Briefly describe the study type including whether data are quantitative, qualitative, or mixed-methods (e.g. qualitative cross-sectional, quantitative experimental, mixed-methods case study). |
| Research sample | State the research sample (e.g. Harvard university undergraduates, villagers in rural India) and provide relevant demographic information (e.g. age, sex) and indicate whether the sample is representative. Provide a rationale for the study sample chosen. For studies involving existing datasets, please describe the dataset and source. |
| Sampling strategy | Describe the sampling procedure (e.g. random, snowball, stratified, convenience). Describe the statistical methods that were used to predetermine sample size OR if no sample-size calculation was performed, describe how sample sizes were chosen and provide a rationale for why these sample sizes are sufficient. For qualitative data, please indicate whether data saturation was considered, and what criteria were used to decide that no further sampling was needed. |
| Data collection | Provide details about the data collection procedure, including the instruments or devices used to record the data (e.g. pen and paper, computer, eye tracker, video or audio equipment) whether anyone was present besides the participant(s) and the researcher, and whether the researcher was blind to experimental condition and/or the study hypothesis during data collection. |
| Timing | Indicate the start and stop dates of data collection. If there is a gap between collection periods, state the dates for each sample cohort. |
| Data exclusions | If no data were excluded from the analyses, state so OR if data were excluded, provide the exact number of exclusions and the |

| Data exclusions | rationale behind them, indicating whether exclusion criteria were pre-established. |
|---|---|
| Non-participation | State how many participants dropped out/declined participation and the reason(s) given OR provide response rate OR state that no participants dropped out/declined participation. |
| Randomization | If participants were not allocated into experimental groups, state so OR describe how participants were allocated to groups, and if allocation was not random, describe how covariates were controlled. |

# Ecological, evolutionary & environmental sciences study design

All studies must disclose on these points even when the disclosure is negative.

| Study description | Briefly describe the study. For quantitative data include treatment factors and interactions, design structure (e.g. factorial, nested, hierarchical), nature and number of experimental units and replicates. |
|---|---|
| Research sample | Describe the research sample (e.g. a group of tagged Passer domesticus, all Stenocereus thurberi within Organ Pipe Cactus National Monument), and provide a rationale for the sample choice. When relevant, describe the organism taxa, source, sex, age range and any manipulations. State what population the sample is meant to represent when applicable. For studies involving existing datasets, describe the data and its source. |
| Sampling strategy | Note the sampling procedure. Describe the statistical methods that were used to predetermine sample size OR if no sample-size calculation was performed, describe how sample sizes were chosen and provide a rationale for why these sample sizes are sufficient. |
| Data collection | Describe the data collection procedure, including who recorded the data and how. |
| Timing and spatial scale | Indicate the start and stop dates of data collection, noting the frequency and periodicity of sampling and providing a rationale for these choices. If there is a gap between collection periods, state the dates for each sample cohort. Specify the spatial scale from which the data are taken |
| Data exclusions | If no data were excluded from the analyses, state so OR if data were excluded, describe the exclusions and the rationale behind them, indicating whether exclusion criteria were pre-established. |
| Reproducibility | Describe the measures taken to verify the reproducibility of experimental findings. For each experiment, note whether any attempts to repeat the experiment failed OR state that all attempts to repeat the experiment were successful. |
| Randomization | Describe how samples/organisms/participants were allocated into groups. If allocation was not random, describe how covariates were controlled. If this is not relevant to your study, explain why. |
| Blinding | Describe the extent of blinding used during data acquisition and analysis. If blinding was not possible, describe why OR explain why blinding was not relevant to your study. |

Did the study involve field work? ☐ Yes ☐ No

# Field work, collection and transport

| Field conditions | Describe the study conditions for field work, providing relevant parameters (e.g. temperature, rainfall). |
|---|---|
| Location | State the location of the sampling or experiment, providing relevant parameters (e.g. latitude and longitude, elevation, water depth). |
| Access & import/export | Describe the efforts you have made to access habitats and to collect and import/export your samples in a responsible manner and in compliance with local, national and international laws, noting any permits that were obtained (give the name of the issuing authority, the date of issue, and any identifying information). |
| Disturbance | Describe any disturbance caused by the study and how it was minimized. |

# Reporting for specific materials, systems and methods

We require information from authors about some types of materials, experimental systems and methods used in many studies. Here, indicate whether each material, system or method listed is relevant to your study. If you are not sure if a list item applies to your research, read the appropriate section before selecting a response.

## Materials & experimental systems

| n/a | Involved in the study |
|-----|----------------------|
| ☒ | ☐ Antibodies |
| ☐ | ☒ Eukaryotic cell lines |
| ☒ | ☐ Palaeontology and archaeology |
| ☐ | ☒ Animals and other organisms |
| ☒ | ☐ Clinical data |
| ☒ | ☐ Dual use research of concern |
| ☒ | ☐ Plants |

## Methods

| n/a | Involved in the study |
|-----|----------------------|
| ☒ | ☐ ChIP-seq |
| ☒ | ☐ Flow cytometry |
| ☒ | ☐ MRI-based neuroimaging |

## Antibodies

| | |
|--|--|
| Antibodies used | *Describe all antibodies used in the study; as applicable, provide supplier name, catalog number, clone name, and lot number.* |
| Validation | *Describe the validation of each primary antibody for the species and application, noting any validation statements on the manufacturer's website, relevant citations, antibody profiles in online databases, or data provided in the manuscript.* |

## Eukaryotic cell lines

Policy information about cell lines and Sex and Gender in Research

| | |
|--|--|
| Cell line source(s) | The MDA-MB-231 cells were purchased directly from ATCC. |
| Authentication | We had them authenticated through ATCC in 2020 using Short Tandem Repeat (STR) analysis. |
| Mycoplasma contamination | We routinely check our cells for mycoplasma contamination and all the cells used in the experiments were negative for mycoplasma. |
| Commonly misidentified lines (See ICLAC register) | There are no other commonly misidentified cell lines listed for MDA-MB-231 cells in the ICLAC register. |

## Palaeontology and Archaeology

| | |
|--|--|
| Specimen provenance | *Provide provenance information for specimens and describe permits that were obtained for the work (including the name of the issuing authority, the date of issue, and any identifying information). Permits should encompass collection and, where applicable, export.* |
| Specimen deposition | *Indicate where the specimens have been deposited to permit free access by other researchers.* |
| Dating methods | *If new dates are provided, describe how they were obtained (e.g. collection, storage, sample pretreatment and measurement), where they were obtained (i.e. lab name), the calibration program and the protocol for quality assurance OR state that no new dates are provided.* |

☐ Tick this box to confirm that the raw and calibrated dates are available in the paper or in Supplementary Information.

| | |
|--|--|
| Ethics oversight | *Identify the organization(s) that approved or provided guidance on the study protocol, OR state that no ethical approval or guidance was required and explain why not.* |

Note that full information on the approval of the study protocol must also be provided in the manuscript.

## Animals and other research organisms

Policy information about studies involving animals; ARRIVE guidelines recommended for reporting animal research, and Sex and Gender in Research

| | |
|--|--|
| Laboratory animals | Zebrafish EKW strain |
| Wild animals | This study did not involve any wild animals. |
| Reporting on sex | We used zebrafish embryos younger than 2 day post fertilization. At that time, zebrafish embryos have not yet sexually differentiated. |
| Field-collected samples | The study did not included any field-collected samples. |

| Ethics oversight | All animal experiments were approved by the Institutional Animal Care and Use Committe (IACUC) at the University of California, San Francisco, USA. |

Note that full information on the approval of the study protocol must also be provided in the manuscript.

# Clinical data

Policy information about clinical studies

All manuscripts should comply with the ICMJE guidelines for publication of clinical research and a completed CONSORT checklist must be included with all submissions.

| Clinical trial registration | *Provide the trial registration number from ClinicalTrials.gov or an equivalent agency.* |
| Study protocol | *Note where the full trial protocol can be accessed OR if not available, explain why.* |
| Data collection | *Describe the settings and locales of data collection, noting the time periods of recruitment and data collection.* |
| Outcomes | *Describe how you pre-defined primary and secondary outcome measures and how you assessed these measures.* |

# Dual use research of concern

Policy information about dual use research of concern

## Hazards

Could the accidental, deliberate or reckless misuse of agents or technologies generated in the work, or the application of information presented in the manuscript, pose a threat to:

No | Yes

- ☐ ☐ Public health
- ☐ ☐ National security
- ☐ ☐ Crops and/or livestock
- ☐ ☐ Ecosystems
- ☐ ☐ Any other significant area

## Experiments of concern

Does the work involve any of these experiments of concern:

No | Yes

- ☐ ☐ Demonstrate how to render a vaccine ineffective
- ☐ ☐ Confer resistance to therapeutically useful antibiotics or antiviral agents
- ☐ ☐ Enhance the virulence of a pathogen or render a nonpathogen virulent
- ☐ ☐ Increase transmissibility of a pathogen
- ☐ ☐ Alter the host range of a pathogen
- ☐ ☐ Enable evasion of diagnostic/detection modalities
- ☐ ☐ Enable the weaponization of a biological agent or toxin
- ☐ ☐ Any other potentially harmful combination of experiments and agents

# Plants

| Seed stocks | *Report on the source of all seed stocks or other plant material used. If applicable, state the seed stock centre and catalogue number. If plant specimens were collected from the field, describe the collection location, date and sampling procedures.* |
| Novel plant genotypes | *Describe the methods by which all novel plant genotypes were produced. This includes those generated by transgenic approaches, gene editing, chemical/radiation-based mutagenesis and hybridization. For transgenic lines, describe the transformation method, the number of independent lines analyzed and the generation upon which experiments were performed. For gene-edited lines, describe the editor used, the endogenous sequence targeted for editing, the targeting guide RNA sequence (if applicable) and how the editor was applied.* |
| Authentication | *Describe any authentication procedures for each seed stock used or novel genotype generated. Describe any experiments used to assess the effect of a mutation and, where applicable, how potential secondary effects (e.g. second site T-DNA insertions, mosiacism, off-target gene editing) were examined.* |

# ChIP-seq

## Data deposition

☐ Confirm that both raw and final processed data have been deposited in a public database such as GEO.

☐ Confirm that you have deposited or provided access to graph files (e.g. BED files) for the called peaks.

| | |
|---|---|
| **Data access links** *May remain private before publication.* | *For "Initial submission" or "Revised version" documents, provide reviewer access links. For your "Final submission" document, provide a link to the deposited data.* |
| **Files in database submission** | *Provide a list of all files available in the database submission.* |
| **Genome browser session** (e.g. UCSC) | *Provide a link to an anonymized genome browser session for "Initial submission" and "Revised version" documents only, to enable peer review. Write "no longer applicable" for "Final submission" documents.* |

## Methodology

| | |
|---|---|
| **Replicates** | *Describe the experimental replicates, specifying number, type and replicate agreement.* |
| **Sequencing depth** | *Describe the sequencing depth for each experiment, providing the total number of reads, uniquely mapped reads, length of reads and whether they were paired- or single-end.* |
| **Antibodies** | *Describe the antibodies used for the ChIP-seq experiments; as applicable, provide supplier name, catalog number, clone name, and lot number.* |
| **Peak calling parameters** | *Specify the command line program and parameters used for read mapping and peak calling, including the ChIP, control and index files used.* |
| **Data quality** | *Describe the methods used to ensure data quality in full detail, including how many peaks are at FDR 5% and above 5-fold enrichment.* |
| **Software** | *Describe the software used to collect and analyze the ChIP-seq data. For custom code that has been deposited into a community repository, provide accession details.* |

# Flow Cytometry

## Plots

Confirm that:

☐ The axis labels state the marker and fluorochrome used (e.g. CD4-FITC).

☐ The axis scales are clearly visible. Include numbers along axes only for bottom left plot of group (a 'group' is an analysis of identical markers).

☐ All plots are contour plots with outliers or pseudocolor plots.

☐ A numerical value for number of cells or percentage (with statistics) is provided.

## Methodology

| | |
|---|---|
| **Sample preparation** | *Describe the sample preparation, detailing the biological source of the cells and any tissue processing steps used.* |
| **Instrument** | *Identify the instrument used for data collection, specifying make and model number.* |
| **Software** | *Describe the software used to collect and analyze the flow cytometry data. For custom code that has been deposited into a community repository, provide accession details.* |
| **Cell population abundance** | *Describe the abundance of the relevant cell populations within post-sort fractions, providing details on the purity of the samples and how it was determined.* |
| **Gating strategy** | *Describe the gating strategy used for all relevant experiments, specifying the preliminary FSC/SSC gates of the starting cell population, indicating where boundaries between "positive" and "negative" staining cell populations are defined.* |

☐ Tick this box to confirm that a figure exemplifying the gating strategy is provided in the Supplementary Information.

# Magnetic resonance imaging

## Experimental design

| | |
|---|---|
| **Design type** | *Indicate task or resting state; event-related or block design.* |

| | |
|---|---|
| Design specifications | *Specify the number of blocks, trials or experimental units per session and/or subject, and specify the length of each trial or block (if trials are blocked) and interval between trials.* |
| Behavioral performance measures | *State number and/or type of variables recorded (e.g. correct button press, response time) and what statistics were used to establish that the subjects were performing the task as expected (e.g. mean, range, and/or standard deviation across subjects).* |

## Acquisition

| | |
|---|---|
| Imaging type(s) | *Specify: functional, structural, diffusion, perfusion.* |
| Field strength | *Specify in Tesla* |
| Sequence & imaging parameters | *Specify the pulse sequence type (gradient echo, spin echo, etc.), imaging type (EPI, spiral, etc.), field of view, matrix size, slice thickness, orientation and TE/TR/flip angle.* |
| Area of acquisition | *State whether a whole brain scan was used OR define the area of acquisition, describing how the region was determined.* |

Diffusion MRI ☐ Used ☐ Not used

## Preprocessing

| | |
|---|---|
| Preprocessing software | *Provide detail on software version and revision number and on specific parameters (model/functions, brain extraction, segmentation, smoothing kernel size, etc.).* |
| Normalization | *If data were normalized/standardized, describe the approach(es): specify linear or non-linear and define image types used for transformation OR indicate that data were not normalized and explain rationale for lack of normalization.* |
| Normalization template | *Describe the template used for normalization/transformation, specifying subject space or group standardized space (e.g. original Talairach, MNI305, ICBM152) OR indicate that the data were not normalized.* |
| Noise and artifact removal | *Describe your procedure(s) for artifact and structured noise removal, specifying motion parameters, tissue signals and physiological signals (heart rate, respiration).* |
| Volume censoring | *Define your software and/or method and criteria for volume censoring, and state the extent of such censoring.* |

## Statistical modeling & inference

| | |
|---|---|
| Model type and settings | *Specify type (mass univariate, multivariate, RSA, predictive, etc.) and describe essential details of the model at the first and second levels (e.g. fixed, random or mixed effects; drift or auto-correlation).* |
| Effect(s) tested | *Define precise effect in terms of the task or stimulus conditions instead of psychological concepts and indicate whether ANOVA or factorial designs were used.* |

Specify type of analysis: ☐ Whole brain ☐ ROI-based ☐ Both

| | |
|---|---|
| Statistic type for inference (See Eklund et al. 2016) | *Specify voxel-wise or cluster-wise and report all relevant parameters for cluster-wise methods.* |
| Correction | *Describe the type of correction and how it is obtained for multiple comparisons (e.g. FWE, FDR, permutation or Monte Carlo).* |

## Models & analysis

| n/a | Involved in the study |
|---|---|
| ☐ | ☐ Functional and/or effective connectivity |
| ☐ | ☐ Graph analysis |
| ☐ | ☐ Multivariate modeling or predictive analysis |

| | |
|---|---|
| Functional and/or effective connectivity | *Report the measures of dependence used and the model details (e.g. Pearson correlation, partial correlation, mutual information).* |
| Graph analysis | *Report the dependent variable and connectivity measure, specifying weighted graph or binarized graph, subject- or group-level, and the global and/or node summaries used (e.g. clustering coefficient, efficiency, etc.).* |
| Multivariate modeling and predictive analysis | *Specify independent variables, features extraction and dimension reduction, model, training and evaluation metrics.* |

