## [Peer Review File · Nature Methods]

Ultrack: pushing the limits of cell tracking across biological scales

Corresponding Author: Dr Loic Royer

Version 0:

Decision Letter:

29th Oct 2024

Dear Loic,

Your Article, "Ultrack: pushing the limits of cell tracking across biological scales", has now been seen by three reviewers. As you will see from their comments below, although the reviewers find your work of considerable potential interest, they have raised a number of concerns. We are interested in the possibility of publishing your paper in Nature Methods, but would like to consider your response to these concerns before we reach a final decision on publication. We therefore invite you to revise your manuscript to address these concerns.

We ask that you focus your revision on emphasizing general applicability, better explaining the technical details of the algorithm, and making code installation as smooth as possible (we suggest Code Ocean as a way to dockerize the software, which was suggested by ref 1).

Link Redacted

We hope to receive your revised paper within 2-3 months. If you cannot send it within this time, please let us know. In this event, we will still be happy to reconsider your paper at a later date so long as nothing similar has been accepted for publication at Nature Methods or published elsewhere.

OPEN SCIENCE REQUIREMENTS

REPORTING SUMMARY AND EDITORIAL POLICY CHECKLISTS

DATA AVAILABILITY

All novel DNA and RNA sequencing data, protein sequences, genetic polymorphisms, linked genotype and phenotype data, gene expression data, macromolecular structures, and proteomics data must be deposited in a publicly accessible database, and accession codes and associated hyperlinks must be provided in the "Data Availability" section.

CODE AVAILABILITY

Please include a "Code Availability" subsection in the Online Methods which details how your custom code is made available. Only in rare cases (where code is not central to the main conclusions of the paper) is the statement "available upon request" allowed (and reasons should be specified).

MATERIALS AVAILABILITY

Authors reporting new chemical compounds must provide chemical structure, synthesis and characterization details. Authors

reporting mutant strains and cell lines are strongly encouraged to use established public repositories.

ORCID

Sincerely,
Rita

Rita Strack, Ph.D.
Senior Editor
Nature Methods

Reviewers' Comments:

Reviewer #1 (Remarks to the Author):

A. Summary of Key Results

The Authors have created an assemblage of image analysis methods, models, and pipelines into a single suite of tools called Ultrack that aids researchers to do tracking of cells at scale. The Authors have created multiple data benchmarks where they hand annotate data to provide ground truth, incorporated cutting edge methods and models from the community, and also implement a variety of high value add simplifications and streamlined capabilities to make the user experience when using Ultrack dramatically improved regardless of if its from a GUI, programming interface, or command line. The amount of work, data, and integration of methods is quite impressive.

B. Originality and significance:

Scalable 3D cell tracking in dense/complex tissues has been a holy grail in image analysis for decades. The originality of the paper lies less in any single dataset/model/approach the authors propose and more in the accumulation of all the data, benchmarks, and tools and how all of them are implemented for scalability (or in the case of the datasets at such a large dataset size and complexity), across a variety of interfaces. This combination of large data, human annotated benchmarks, many tools, scalable implementation, and diverse user interfaces makes the work quite significant in the field. I would guess this will serve as the new benchmark for all other cell tracking work to compare against.

C. Data and Methodology

All imaging data is shared in the latest scalable ome.zarr format which is fantastic, for imaging specialists. However, I am mildly concerned that the greater community who want to do cell tracking are still using tiff and similar formats. Providing the image data in both formats might make the benchmarks more relevant to a larger audience. Conversely, all cell tracks are provided in community accepted/standard formats. Here, using cutting edge metadata standards like Microjson in addition to these common community standards would help future proof the tools.

The Author's use of UNet for their custom segmentation networks is strange given that in Figure 1 the authors list two models - StarDist and microSAM that are easily tunable/trainable and have been repeatedly shown in the literature to perform more robustly (and quickly) than UNet models.

Further, microSAM has a great Napari interface to aid in generating ground truth to then fine-tune it on. The authors should justify the use of this old technology or at least explain why they do not use more modern networks that are freely available and relatively easy to fine tune.

In the methods authors need to list all package version, operating system, CUDA version (if relevant), and hardware they did their work on and what figures/data were generated with which. Further the Authors reference a library called iohub in the methods section but the citation for this python package is not complete and I could not find it after digging through the authors GitHub. It is referenced in 1-2 places in their documentation but never linked and therefore what this library is, what version was used, and where it is, is needed.

The methods don't contain nearly enough detail to both understand what was done algorithmically, or repeat the work proposed. Examples include: No discussion of UNet was found in the methods but I believe multiple UNet models were trained. Values of user set variables/hyperparameters for most/all algorithms used were not documented (examples include the fixed number of iterations for theta optimization in the flow field estimation, user specified lower and upper quantiles for light-sheet data preprocessing, etc.), nor was the existence of these parameters – as an example because I've used the top-hat transform function from OpenCV I know it has a parameter to define the kernel. Not only was this not mentioned, but what value was set was not documented. Given the inherent complexity of the pipelines and libraries used in Ultrack I assume there are many such parameters that are not mentioned/documented but that were used. The Authors need to document these. The "Candidate Segmentation Association" seems to be missing a formula. The code is available at the links provided but actually using this massive codebase to DO what the authors did is not possible at the level of detail currently in the methods section. I would strongly encourage the Authors to reference more than just the topline repo in their various methods sections to enable users to actually recapitulate the Authors work.

The "Color features association" section in the Methods section is of particular concern. $\mu(i)$ was not defined, I assume $\mu(i)$ is the mean intensity vector of neighboring segment $S(i)$ at time $t-1$. But it should be explicitly defined. Why was a threshold of 3 chosen? As described, you're saying for every object in a reference frame, you want to subtract the mean intensity of a candidate in the next frame (However, because there's no specification on candidate here I assume all objects at $t-1$ are candidates?) But then the max value doesn't make sense, so perhaps it's all time steps, but if that is true there is no time variable anywhere in the equation?

I think what the Authors are trying to say is that they are taking a difference in means of an object and a candidate, and dividing that by the frame of reference object intensity standard deviation. The Authors are then taking the absolute value of that number and determining if it is less than 3, if it is then that object is kept as a candidate in the ILP solve. Otherwise the Authors exclude it. I think the Authors are further doing this pairwise comparison across all objects at $t-1$ with each object in the reference frame. This is quite computationally in-efficient as its a one to many comparison. Wouldn't it make more sense to set a distance threshold of candidate options in frame $t-1$ so that EVERY object in the frame isn't compared? Also, I'm not confident my guess is correct because the written equation is still taking the max of an absolute value, which indicates to me that it is a vector of values. This might be for a given object and a given candidate at all pairwise time steps? But then the Authors make the assumption that the candidate is the reference object for any time step $t-2$ and beyond. Which further complicates/expands the computational inefficiency of the calculation. No matter what, the Authors need to dramatically rewrite this formula and explanation.

Further, even if the above is clarified, it is unclear to me why the Authors divide the change in average intensity only by the reference frame intensity standard deviation? Its close to a z-score, but a delta mean intensity divided by the standard deviation of pixels that might or might not be related to one of the object means (in the case where the candidate object is, in fact, not the reference) has no physical meaning/reference that I can think of. Had you of taken the z-score of each object, subtracted them, and looked at the absolute value of that and set it less than 3 I think there's distributional relationships we could make based off known z-score relationships to probability distribution functions. But as it stands, I can't understand why you chose this approach and why it's meaningful.

Code and data availability seems to include all code and data needed to do the analyses discussed, but not the code to generate the figures from the specific datasets that generated those figures (which I was not able to find those specific datasets, but I think they exist, there are just many nested directories without clear documentation on what is the appropriate place to look for the data/graphing code for each figure.)

In the "Label conversion to contour and foreground map" section the Authors state "this averaging approach, as opposed to summation, keeps pixel values within the 0 to 1 range." However, so would summing and normalizing. As would many other approaches. Why did the Authors choose this approach? What was the rationale? What else was tried and how did it perform?

D. Appropriate use of statistics and treatment of uncertainties

1. Throughout the manuscript the authors use relatively biased measures of performance (false negative rates, explicit definitions of region based false positive/negative, edges errors, etc.). Interestingly, from the methods it appears as if they use less biased segmentation assessment tools to train their networks (things like Jaccard) but in figures rely on a small number of biased measures. I would strongly encourage the authors to look into not only accuracy, but less biased region-based assessments like Fowlkes–Mallows index, IoU/Jaccard, and/or F1 score. These have been reported in many publications but a comprehensive set of them can be found at <https://doi.org/10.1186/s12859-023-05486-8>. Additionally, in Figure 6D it seems there are three objectives that could be evaluated in a confusion matrix (1) Correct identification of connections between cells Y/N (2) Division Y/N (3) Correct identity Y/N. Then you could do unbiased assessments (Jaccard/F1/etc.) for each of those and take averages of those across the time periods you indicate using the deep vs not segmentation approaches.

2. All assessments were repeated one time (albeit on MANY cells within that one run). While I understand the HUGE computational complexity and thus associated costs, repeatedly in the literature it has been shown that deep learning is probabilistic not deterministic and therefore any workflow that uses deep learning (which I believe is all of them for this manuscript) should be repeated multiple times to gain a confidence interval for performance. Ideally, re-training networks from scratch using the same training data would happen to make even more robust estimates of performance/generalizability of approach, however I think at the scales of data discussed in this manuscript that is infeasible. The ILP method proposed relies heavily on these candidate segmentations and therefore, I think the Authors will need to, at the minimum, run the segmentation pipelines multiple times with their deep learning models (both trained and community models.)

E. Conclusions: robustness, validity, reliability

In several locations (abstract, introduction, and discussion) in the document the Authors highlight the utility/value of the multi-level contours so that training/fine tuning of deep learning models is not needed. But ironically, for all large 3D cases the Authors train their own custom UNet for segmentation and show superior performance to using intensity contour proxies. I think these areas should be reworded to better reflect the results found in this manuscript.

F. Suggested improvements: experiments, data for possible revision

Figure 1 needs to have an example of a foreground map. And the vernacular used for referring to foreground maps should be made consistent both in the figure and throughout the manuscript. The Authors refer to it in several different ways (segmentation map, foreground mask, etc.).

G. References: appropriate credit to previous work?

Generally, the Authors do a good job of citing relevant work. However, a few references 36, 38, 46, 47, 59, 78, 87, 88, 98, and 101 are not well enough specified/defined to be able to find/know what the Authors are referring to.

H. Clarity and context: lucidity of abstract/summary, appropriateness of abstract, introduction and conclusions

The manuscript was very well written. A very minor correction was that a lagging “l” was left on page 8 the very first sentence. Other than that, the grammar and writing was very clear (with exceptions in the methods described above).

On page 5 the Authors use the term “color features” in the Enhancing cell tracking accuracy through multi-channel integration and color-aware associations.” Section however it is not defined yet. Please define or at least refer to the section of the methods where it is defined.

On page 19 that statement “allowing the algorithm to ignore background regions and thereby greatly speeding up the process” is only true in non-dense object situations. I would ask the Authors to caveat this statement with the conditions under which it is valid.

Reviewer #1 (Remarks on code availability):

Code quality itself is well above average and I was able to (eventually) get many of the libraries and CLIs to run, this was non-trivial for GPU implementations. I would strongly encourage the authors to deliver containerized versions (ideally both docker and aptainer/singularity) of CLIs and Python libraries, especially for GPU versions.

The documentation provided was sufficient for me to run these libraries and CLIs on some of my own data. However, I would strongly suggest the Authors create a repo devoted to JUST the recapitulation of the results from this paper. Trying to connect all the software/steps with what was done in each figure with the correct dataset, ensuring test/train splits, etc. was essentially outside of my capabilities. Its not that I couldn't do it eventually (I think) but data, code, and tooling is so distributed and naming not consistent between the manuscript and the repos that I just ran out of patience. I believe most/all of it exists in what the Authors have provided, but its just too difficult to recapitulate the actual results shown at the current documentation level.

But to reiterate, the documentation provided is sufficient for me to use these tools and apply them to my own data, just not to recapitulate the results of this paper.

Reviewer #2 (Remarks to the Author):

In this study, Bragantini et al. developed Ultrack, a scalable cell tracking tool that simultaneously performs segmentation and tracking. The tool leverages segmentation hypotheses from various algorithms, simplifying parameter tuning while improving accuracy. By solving an Integer Linear Programming (ILP) problem, Ultrack optimizes both segmentation and tracking. It proves effective in densely labeled imaging, as demonstrated by a dual-color labeling approach where one channel is sparsely labeled, facilitating manual curation. This method enhances tracking accuracy, outperforming state-of-the-art techniques with fewer cumulative errors.

Two key in vivo applications were demonstrated: in developing zebrafish embryos, Ultrack scaled up to terabyte-scale data, tracking developing cells with error-free results over dozens of time points using either high-performance computing platforms or a desktop. The second application focused on developing neuromast cells, accurately capturing cellular migrations, divisions, and cell deaths over tens of hours.

The manuscript benchmarks Ultrack's performance across multiple datasets, particularly in comparison to tools like TrackMate. It demonstrates seamless integration with platforms such as Fiji and napari, along with high scalability from personal computers to high-performance computing clusters, which will be a key advantage for users with varying programming expertise. Its innovative approach to handling hard-tuned parameters, uncertainties, and challenges in large datasets makes it a valuable contribution to the field of cell tracking, particularly for extracting meaningful insights from complex biological datasets.

Major Concerns:

1. Although the applications demonstrate superior performance in terms of accuracy, speed, and scalability compared to other methods, the manuscript would benefit from a clearer emphasis on the biological insights enabled by Ultrack. It is not enough to simply showcase improved methodologies; the authors should highlight cases where the enhanced accuracy and throughput of Ultrack reveal biological phenomena that would otherwise go unnoticed. Demonstrating how these capabilities contribute to deeper biological understanding will strengthen the impact of the work.
2. The manuscript's claims regarding the tool's generalizability are not fully convincing, and its application conditions are not described subjectively enough. For instance, in Figure 2b, the "all" condition seems unrealistic. If cells move across the entire frame and the foreground is fully occupied, using logical OR for combining segmentations would not be effective. The authors should better explain the limitations and recommend suitable conditions for its applicability.

Minor Concerns:

1. The description of experimental conditions, such as the sampling rate's impact on tracking quality, could be improved. The absolute imaging conditions may not be informative enough across different microscopy setups and cellular processes. Instead of using absolute measurements (like Δt in Fig. 5), it would be more useful to describe how much spatial change occurs within a sampling interval. For instance, explaining how movements smaller than a certain proportion of cell size do not require registration would provide better intuition.
2. Terms like "without fine-tuning" appear multiple times in the manuscript. It would be helpful to clarify what this means in practice. Does it refer to using default settings, or does it involve some minimal adjustments like threshold tweaking? More precise descriptions would be beneficial.
3. While the supplementary videos effectively convey complex processes, the figures and legends could be clearer to aid navigation. For example, in Figure 3b, the third and fourth columns should be swapped so that the most complete method (multi-color combined Ultrack) appears as the final step, aligning with the flowcharts.
4. In sparse labeling experiments, Tol2 injections could potentially damage transgene expression, leading to fewer H2B-mNeonGreen-expressing cells. The source of H2B-mNeonGreen should be clarified. If the gene was originally inserted via Tol2, the newly introduced Tol2 could disrupt its expression. A fair comparison with Malin-Mayor et al., 2022, should address any differences in the density of cells in densely labeled channels. Additionally, if cells expressing only the red label were observed, were these excluded from the analysis? Clarification on this point would be helpful.
5. In the neuromast tracking experiment, the manuscript should provide more details about the cell types being tracked. Are all the cells in the organ included, and do the two transgenic lines label the same cell populations? This information is important for understanding the scope and accuracy of the tracking.
6. The manuscript does not discuss cell-entering events. While cell divisions and exits are demonstrated in Figures 3c and 8b, the manuscript lacks clarity on cell entry. Are these events categorized as divisions? Clear differentiation between these processes would be useful. Furthermore, if Ultrack can track cells that disappear and later reappear (due to out-of-frame movement), this capability should be emphasized, as it would restore continuous cell information in 2D or 3D imaging without dense z-axis sampling.
7. Is it possible to distinguish between error-prone and error-free tracking in Figure 7? While the data shows that around 1 hour or 62 frames yields 50% error-free tracks, this may not be sufficient for longer experiments spanning hundreds or thousands of frames, or even for leveraging the error-free tracks effectively. Given the impracticality of manual correction for such large datasets, automatically identifying the reliable 50% would be highly beneficial for further analysis and curation. Can Ultrack already categorize tracks based on reliability?

Typos and others:

1. Missing period in "tracking cells in developing embryos".
2. "our baseline (0.078) even over a much more challenging 500-frame period" should be "their baseline".
3. In transgenic line names, it's better to use the format "Tg", e.g., tg(she:h2b-EGFP).

Reviewer #2 (Remarks on code availability):

The code has a clear README, installation instructions, and usage examples. It is easy to install and run locally and is well-documented and accessible for reproducibility and use by the community.

Reviewer #3 (Remarks to the Author):

In this manuscript, Bragatini et al. present an image analysis tool, called Ultrack, for tracking cells in time-lapse microscopy image data of various modalities and dimensionality. From the algorithmic point of view, Ultrack does not bring much novelty to field as its key principles, the hierarchical representation of multiple instance segmentation maps using ultrametric contours and

an ILP-driven multiple hypothesis linking scheme, have already been published by the same authors in their very recent paper (Bragatini et al., ECCV 2024) referenced as [45]. However, from the practical point of view, Ultrack is a valuable contribution for the community as documented by extensive experimental results reported in this manuscript, and by deploying it in the form of a user-friendly add-on to popular Fiji and napari frameworks or of a command-line interface for high-performance computing environments.

Major comments

=====

- In general, the RESULTS section combines a description of the main Ultrack's features with experimental results. Indeed, the first three subsections do not contain any experimental results, and thus shall be moved to the METHODS section.
- RESULTS (Optimal parameter selection...): Please clarify how exactly Ultrack addresses computational and practical challenges mentioned in (i) and (ii), respectively. To this reviewer's understanding and as it is shown by numerous experiments conducted in this work, users must generate multiple segmentation inputs on their own and with their own knowledge of image processing and programming skills. Furthermore, the analysis of image data with hepatocarcinoma-derived cells is a proof of concept rather than a systematic study, based on which one could claim that a hierarchical combination of diverse segmentation inputs is more effective and robust in practice than a single, highly optimized solution. It would be more convincing to support this strong claim using more datasets. For example, one could take top-performing results for different datasets from the Cell Tracking Challenge benchmark and show that Ultrack can outperform all of them when combining their outputs together using ultrametric contours.
- Fig. 3: The visualization given in the panel B is difficult to digest and to map to the experiment description given in the main text. It would be more informative to choose one field of view for all crops to recognize better the differences between individual configurations. The purpose of the panel C is unclear, and it does not convey any important information in its current form. The red-green color scheme used in the panel D is unfortunate because it might indicate a range of colors starting from red (the worst value) and going to green (the best value).
- RESULTS (Intensity-based tracking from label-free...): This subsection again contains proof-of-concept results using a single type of data, and thus it is highly questionable whether Ultrack can broadly be applied for such a type of analysis in daily practice when different label-free modalities, such as QPI, phase contrast, DIC, or brightfield, are exploited.
- RESULTS (Ultrack achieves state-of-the-art cell tracking...): Based on the results reported in [45] and those listed at <https://celltrackingchallenge.net/participants/AC-6/>, it seems that only a subset of results submitted to the Cell Tracking Challenge benchmark has been reported in this manuscript. For the sake of completeness, it would be informative for readers to know that Ultrack does not excel in all situations evaluated and to understand what the reasons for its lower performance in these situations are.
- DISCUSSION: The limitations of Ultrack are discussed very briefly. It would be informative for the reader to know some guidelines on how to prepare reliable segmentation inputs for Ultrack and how much the performance of Ultrack is affected by different types of segmentation errors committed in individual input segmentation masks.

Other remarks:

- page 1, col. 1, 2nd paragraph: Clarify why segmentation errors are accumulated over time when the majority of existing cell segmentation methods process individual images separately, without considering any temporal context.
- page 1, col. 1, 3rd paragraph: It is not always true that cells cannot merge over time. There can be other biological constraints, such as cell fusion in plant cells, not covered in this work. Can Ultrack handle such situations?
- RESULTS (Optimal parameter selection...): Obviously not all parametrizations of gamma were considered (only four of them). Please rephrase. Furthermore, clarify what "manually curated ground truth" refers to and whether the two publicly available datasets refer to the pair of available training sequences.
- page 7, col. 2, 2nd paragraph: "The cell centroids of these tracks" -> "The centroids of the tracked cells" or alike
- page 8, col. 2, 4th paragraph: It seems that the test datasets from the Cell Tracking Challenge dataset repository are freely available, but without reference annotations. Please rephrase.
- page 8, col. 2, last paragraph: It seems that some segmentation annotations are publicly available for the fly and beetle datasets. Please clarify why these cannot be used to train segmentation models.
- Table 1: The superscripts attached to individual scores are undefined.
- page 19, col. 2, 1st paragraph: Specify the neighborhood used and how it is guaranteed that extracted contours of individual segments are closed surfaces in 3D.
- Equation 12: "...between the predicted and gt instances is at least half..." -> "...between the predicted and gt instances is more than half..."

Version 1:

Decision Letter:

Our ref: NMETH-A57608A

4th Apr 2025

Dear Loic,

Thank you for submitting your revised manuscript "Ultrack: pushing the limits of cell tracking across biological scales" (NMETH-A57608A). It has now been seen by the original referees and their comments are below. The reviewers find that the paper has improved in revision, and therefore we'll be happy in principle to publish it in Nature Methods, pending minor revisions to satisfy the referees' final requests and to comply with our editorial and formatting guidelines.

Regarding the remaining request from reviewer 2 to benchmark performance under conditions of varying axial resolution, while we appreciate this would strengthen the work, we do not consider it strictly necessary for publication, and leave it to you if you want to add supplementary data to this effect. We do ask that you at least discuss the possible impact of poor z-resolution on performance.

TRANSPARENT PEER REVIEW

ORCID

Sincerely,
Rita

Rita Strack, Ph.D.
Senior Editor
Nature Methods

Reviewer #2 (Remarks to the Author):

I appreciate the authors' efforts in addressing the suggestions made during the review process. The manuscript has been significantly improved, and the potential utility of this tool within the field is now well articulated.

It appears that the changes discussed for Figure 5d have not been incorporated into the revised manuscript. Specifically, the manuscript would greatly benefit from additional benchmarking information on the tool's performance under varying temporal resolutions (as shown in changes of Figure 5d) and z-resolutions. Recording neural dynamics often requires high temporal resolution, which typically necessitates compromises in z-resolution. While the four preprint papers cited in the manuscript demonstrate fine z-sampling, many applications frequently face a trade-off between temporal and z-resolution, which can complicate 3D segmentation and tracking. I recommend that the authors provide benchmarks on the tool's performance under these compromised low z-resolution conditions. Such benchmarks would be extremely valuable for potential users, as they would help clarify the tool's robustness and limitations in real-world scenarios.

Reviewer #2 (Remarks on code availability):

The code has a clear README, installation instructions, and usage examples. It is easy to install and run locally and is well-documented.

Reviewer #3 (Remarks to the Author):

The authors have adequately addressed the comments and concerns raised by the reviewers. This reviewer does not have any additional remarks to the revised manuscript.

Reviewer #3 (Remarks on code availability):

Version 2:

Decision Letter:

8th Jul 2025

Dear Loic,

I am pleased to inform you that your Article, "Ultrack: pushing the limits of cell tracking across biological scales", has now been accepted for publication in Nature Methods. The received and accepted dates will be August 20, 2024 and July 8, 2025. This note is intended to let you know what to expect from us over the next month or so, and to let you know where to address any further questions.

Over the next few weeks, your paper will be copyedited to ensure that it conforms to Nature Methods style. Once your paper is typeset, you will receive an email with a link to choose the appropriate publishing options for your paper and our Author Services team will be in touch regarding any additional information that may be required. It is extremely important that you let us know now whether you will be difficult to contact over the next month. If this is the case, we ask that you send us the contact information (email, phone and fax) of someone who will be able to check the proofs and deal with any last-minute problems.

Authors may need to take specific actions to achieve compliance with funder and institutional open access mandates.

If your research is supported by a funder that requires immediate open access (e.g. according to [Plan S principles](https://www.springernature.com/gp/open-science/plan-s-compliance) or the [NIH public access policy](https://www.springernature.com/gp/open-science/us-federal-agency-compliance)) then you should select the gold OA route, and we will direct you to the compliant route where possible. Because authors warrant under our subscription licensing terms that they haven't committed to licensing any version of their article under a licence inconsistent with the terms of our agreement – including the applicable embargo period – publication under the subscription model isn't suitable for authors whose funders require no embargo.

If you are active on Twitter/X or Bluesky, please e-mail me your and your coauthors' handles so that we may tag you when the paper is published.

Best regards,
Rita

Rita Strack, Ph.D.

Senior Editor
Nature Methods

Visit the Springer Nature Editorial and Publishing website at http://editorial-jobs.springernature.com?utm_source=ejP_NMeth_email&utm_medium=ejP_NMeth_email&utm_campaign=ejp_Nmeth or www.springernature.com/editorial-and-publishing-jobs for more information about our career opportunities. If you have any questions please click [here](mailto:editorial.publishing.jobs@springernature.com).

Ultrack: pushing the limits of cell tracking across biological scales

Reviewer #1 (Remarks to the Author):

A. Summary of Key Results

The Authors have created an assemblage of image analysis methods, models, and pipelines into a single suite of tools called Ultrack that aids researchers to do tracking of cells at scale. The Authors have created multiple data benchmarks where they hand annotate data to provide ground truth, incorporated cutting edge methods and models from the community, and also implement a variety of high value add simplifications and streamlined capabilities to make the user experience when using Ultrack dramatically improved regardless of if its from a GUI, programming interface, or command line. The amount of work, data, and integration of methods is quite impressive.

→ We appreciate the reviewer's recognition of our work and their encouraging words.

B. Originality and significance:

Scalable 3D cell tracking in dense/complex tissues has been a holy grail in image analysis for decades. The originality of the paper lies less in any single dataset/model/approach the authors propose and more in the accumulation of all the data, benchmarks, and tools and how all of them are implemented for scalability (or in the case of the datasets at such a large dataset size and complexity), across a variety of interfaces. This combination of large data, human annotated benchmarks, many tools, scalable implementation, and diverse user interfaces makes the work quite significant in the field. I would guess this will serve as the new benchmark for all other cell tracking work to compare against.

→ Thanks. This is exactly what we have endeavored to achieve. It was very important for us to make our work broadly valuable for the community, which, as the reviewer rightly noticed, required providing a complete, optimized, and scalable package of datasets, benchmarks, and tools.

C. Data and Methodology

All imaging data is shared in the latest scalable ome.zarr format which is fantastic, for imaging specialists. However, I am mildly concerned that the greater community who want to do cell tracking are still using tiff and similar formats. Providing the image data in both formats might make the benchmarks more relevant to a larger audience.

→ Indeed, we created TIFFs of all the published data, except for the 5TB zebrafish dataset because, from our experience, hosting and working with TIFF at this scale is problematic and nowadays atypical and not recommended.

Conversely, all cell tracks are provided in community accepted/standard formats. Here, using cutting edge metadata standards like Microjson in addition to these common community standards would help future proof the tools.

→ We included OME-TIFF metadata in the new files, matching their respective OME-zarr counterparts. We preferred it over MicroJSON since its usage is more widespread across the microscopy community.

As an example, we show the OME-TIFF zebrafish neuromast dataset and metadata loaded with FIJI.

The Author's use of UNet for their custom segmentation networks is strange given that in Figure 1 the authors list two models - StarDist and microSAM that are easily tunable/trainable and have been repeatedly shown in the literature to perform more robustly (and quickly) than UNet models. Further, microSAM has a great Napari interface to aid in generating ground truth to then fine-tune it on. The authors should justify the use of this old technology or at least explain why they do not use more modern networks that are freely available and relatively easy to fine tune.

→ To perform best, Ultrack requires as input both a foreground and contour prediction. Of course, one can produce these masks by conversion of instance segmentation predictions obtained from other methods such as Stardist, Cellpose, microSAM, or others. However, the label representation and conversion to masks discard important uncertainty information (i.e., fuzzy 'ultrametric' contours) that is used to generate the multiple segmentation hypotheses. Therefore, training a supervised model to return foreground and contours is the best option, assuming that training data is available.

As an example, foreground and contour prediction was illustrated in Figure 7, from left to right column, it shows the original image, foreground and contours predictions at different levels of the same image:

More generally, Stardist (and Cellpose) are actually built upon an UNet architecture, so despite being old for deep learning standards, it remains, in practice, the state-of-the-art Convolutional

Neural Network (CNN) architecture. The main differences among these different approaches – including our own – is in the final layer, loss function, and post-processing used to obtain the instance segmentation. Moreover, these instance segmentation models are not quicker than a vanilla UNet because they have additional post-processing, which can take a considerable amount of time on large 3D datasets.

MicroSAM, which employs a transformer network architecture, is a different story; during our initial research, we also thought SAM-based approaches would be the perfect pairing for our method because it produces multiple instances per object. However, a few experiments showed that MicroSAM generalization capabilities aren't yet on par with other existing methods. Plus, SAM-based segmentation approaches are currently very slow, mainly because they are interactive methods that segment individual objects per prompt. So, currently, there are limitations to the applicability of SAM-based approaches in the context of large embryo-scale datasets. However, we think that with more training and other improvements, microSAM will soon eventually get there, and both of our methods will enjoy its synergy.

Because other readers might ask themselves the same question, we included this in our discussion section:

One might question why we chose to use foreground and contour map prediction models rather than established instance segmentation approaches like Stardist¹⁸ or MicroSAM²⁰ in Fig. 6 and Fig. 7. The answer lies in the nature of network contour predictions: their probabilistic outputs capture subtle variations in cell boundary likelihood, producing smoothly varying contour values. This continuous representation generates a richer set of segmentation hypotheses compared to instance segmentation methods, which output discrete labels that must then be converted to foreground and contour maps. The probabilistic nature of contour predictions thus provides Ultrack with more information to work with during the optimization process.

In the methods authors need to list all package version, operating system, CUDA version (if relevant), and hardware they did their work on and what figures/data were generated with which.

→ We created a repository https://github.com/royerlab/ultrack_supplementary with the packaging version and parameters configuration for each experiment (configuration/) and the resources used (hardware/). Other replies will also provide additional details on this repository. See screenshot below:

Hardware

Here we provide additional information about the hardware used to execute the experiments in the paper.

These details can be useful in terms of reproducibility, to tune your runtime expectations and assess the results when running Ultrack on different platforms.

Three different hardware configurations were used in the experiments,

Desktop

- Intel i9 9900K CPU
- 4x DDR4 3200Hz 32GB RAM (128GB in total)
- Asus Motherboard Z390 PRO
- Corsair power supply unit 1200
- 2x NVIDIA RTX 3090 Ti
- 1TB Samsung NVMe SSD
- 4x 4TB hard drives
- 10Gbit Ethernet card
- Operational System: Ubuntu 20.04 LTS

Cluster

The cluster contains elastic amount of resources, we constrained our jobs to use at most 100-cpu cores and 20 GPUS when measuring our runtime for different steps.

The resources used for each step are as follows:

0. Network prediction: 20x 4-cpu core 100GB RAM 1 A40 GPU nodes
1. Database: 1x 10-cpu core 1000GB RAM node (extremely excessive, today we are using 128GB of RAM)
2. Segmentation (hypotheses generation): 90x 1-cpu core 100GB RAM nodes
3. Linking (hypotheses linking): 90x 1-cpu core 5GB RAM nodes
4. Tracking: 8x 12-cpu cores 400GB RAM nodes
5. Exporting: 1x 90-cpu core 400GB RAM node

Steps are run sequentially, thus each line corresponds to the total resources used for that step, with the exception of the database which is on and available for all steps after item 1.

The cluster runs Rocky Linux 8, Slurm is used for job scheduling, and GPFS file system.

Laptop

- Intel i9 11980HK CPU
- DDR5 3200Hz 32GB RAM
- NVIDIA RTX 3080 Ti
- 2TB NVMe SSD
- 1Gbit Ethernet
- Operational System: Ubuntu 22.04 LTS

Further, the Authors reference a library called `iohub` in the methods section but the citation for this python package is not complete and I could not find it after digging through the authors GitHub. It is referenced in 1-2 places in their documentation but never linked and therefore what this library is, what version was used, and where it is, is needed.

→ Indeed, we did not notice the online references were incomplete. We addressed this with item G from this reviewer's minor comments section. See below for updated reference:

- [87] Z. Liu, I. Ivanov, J. Bragantini, T. Chandler, A. C. Solak, E. Hirata-Miyasaki, C. Foltz, L.-H. Yeh, and S. Mehta, "iohub vo.1.0," February 2024. [Online]. Available: <https://github.com/czbiohub-sf/iohub>

The methods don't contain nearly enough detail to both understand what was done algorithmically, or repeat the work proposed. Examples include: No discussion of UNet was found in the methods but I believe multiple UNet models were trained.

→ We included an additional description of the UNet architecture of CTC and Zebrahub in Methods, and we are addressing the comment regarding repeating our work below.

Foreground and contours prediction with neural networks. We also implemented a deep learning approach to generate foreground and contour maps, which we used for both the Cell Tracking Challenge submissions (where 3D segmentation annotations were available) and the large zebrafish dataset (Fig. 7). Our implementation uses a U-Net architecture⁶⁸ that processes single-channel grayscale images to produce two-channel outputs: foreground prediction and contour prediction, each with its sigmoid activation layer.

The network architecture consists of four stages of encoding and decoding, with channel depths of 32, 64, 128, and 256, using 5×5 convolution kernels. During decoding, we upsample intermediate representations using linear interpolation and combine them with corresponding encoder features via skip connections. Network training minimizes a dice loss,⁶⁹ supplemented with auxiliary supervision at three intermediate resolutions during decoding, following the approach of.⁷⁰

All network weights and architectural details are publicly available at our supplementary repository (https://github.com/royerlab/ultrack_supplementary), with the complete implementation available at our CTC submission repository (https://github.com/royerlab/ultrack_CTC_submission).

Values of user set variables/hyperparameters for most/all algorithms used were not documented (examples include the fixed number of iterations for theta optimization in the flow field estimation, user specified lower and upper quantiles for light-sheet data preprocessing, etc.), nor was the existence of these parameters – as an example because I've used the top-hat transform function from OpenCV I know it has a parameter to define the kernel. Not only was

this not mentioned, but what value was set was not documented. Given the inherent complexity of the pipelines and libraries used in Ultrack I assume there are many such parameters that are not mentioned/documentated but that were used. The Authors need to document these.

→ We created a new repository https://github.com/royerlab/ultrack_supplementary to address this and other comments from this reviewer. This repository's purpose is to organize and share the parameters used through our experiments, this is mostly done through configuration files, tables, figure generation code and compiled neural network weights for our own CNNs.

We also included this information in the manuscript as:

GPU-accelerated flow field estimation. Flow field estimation of large 3D volumes can be extremely challenging and computationally intensive, often following particle image velocimetry strategies that estimate local translation using phase cross-correlation for small patches of the data.⁶¹ Instead of maximizing cross-correlation, which minimizes the mean-squared error between frames given a field in the coordinate space, we employ gradient descent to optimize the coordinate space directly. This choice, justified by its enhanced computational efficiency and simplicity, allows for the integration of any differentiable function within the coordinate space. The loss function is defined as:

$$L(\theta) = \|I_{t-1} - \text{grid_sample}(I_t, C + \theta)\|_1 + TV(\theta) \quad (1)$$

where I_t denotes the frame at time t , C represents the identity coordinates (*i.e.* no movement), θ indicates the movement flow field we optimize, and TV denotes the L1 total variation loss on the flow field. The `grid_sample` function, a differentiable data sampling method from PyTorch,⁶³ enables this process. This routine is tailored for specialized GPU hardware, greatly enhancing processing speed. The θ optimization is performed for a fixed number of iterations over multiple image resolutions, starting from the lowest resolution, which is then used to initialize the next θ optimization stage. **For our experiment, Fig 5, the gradient descent learning rate was set to 1e-2, the number of scales to 2, and the iterations to 2000. A detailed jupyter notebook is available at https://github.com/royerlab/ultrack_supplementary/tree/main/configuration/flow_registration.**

Note that the goal of this repository is not to provide the code to reproduce the experiments because most of them can already be done through the main repository, with some exceptions, where they were done through our command line interface, so their respective ultrack's configuration files are sufficient.

We present screenshots of our resources below, repository front page:

Ultrack supplementary information

This repository provides supplementary information for the paper Ultrack: pushing the limits of cell tracking across biological scales.

Our goals with this, is to provide a centralized location for the following information:

- Hardware: Describes the hardware used in the experiments.
- Figures: Scripts to generate content for the figures in the paper.
- Configuration: Python package and parameters used in Ultrack's experiments.

For Ultrack's execution and usage refer to our main repository Ultrack.

Citing

```
@article{bragantini2024ultrack,  
  title={Ultrack: pushing the limits of cell tracking across biological scales},  
  author={Bragantini, Jordao and Theodoro, Ilan and Zhao, Xiang and Huijben, Teun APM and Hira},  
  journal={bioRxiv},  
  pages={2024--09},  
  year={2024},  
  publisher={Cold Spring Harbor Laboratory}  
}
```

Example of large zebrafish embryo tracking SLURM execution configuration scripts and configuration:

Name	Last commit message	Last commit da...
..		
README.md	updated torch load to use jit	4 days ago
config.toml	adding large zebrafish files	4 days ago
create_server.sh	adding large zebrafish files	4 days ago
link.sh	adding large zebrafish files	4 days ago
main.sh	adding large zebrafish files	4 days ago
requirements.txt	adding large zebrafish files	4 days ago
segment.sh	adding large zebrafish files	4 days ago
solve.sh	adding large zebrafish files	4 days ago

README.md

Large zebrafish tracking

This directory contains the configuration files for tracking the large zebrafish using SLURM.

This requires a conda environment with the `requirements.txt` file installed and it assumes the environment will be named `ultrack`.

To download the UNet weights use:

```
wget https://public.czbiohub.org/royerlab/ultrack/unet_weights/unet-daxi.pt
```

The weights are compiled with torchscript, so to load them you can simply:

```
import torch
model = torch.jit.load('unet-daxi.pt')
```

Tracking execution:

```
bash main.sh
```

Example of Cell Tracking Challenge submission with Docker containers:

1 Branch 0 Tags Go to file t Go to file Code

 **JoOkuma** 2023-10 submission fa6a20a · 5 days ago ↻

 Fluo-C3DL-MDA231	2023-10 submission	5 days ago
 Fluo-N3DH-CE	2023-10 submission	5 days ago
 Fluo-N3DL-DRO	2023-10 submission	5 days ago
 Fluo-N3DL-TRIC	2023-10 submission	5 days ago
 Fluo-N3DL-TRIF	2023-10 submission	5 days ago
 dexp-dl	2023-10 submission	5 days ago
 ultrack	2023-10 submission	5 days ago
 weights	2023-10 submission	5 days ago
 .dockerignore	2023-10 submission	5 days ago
 .gitignore	2023-10 submission	5 days ago
 Dockerfile	2023-10 submission	5 days ago
 Fluo-C3DL-MDA23...	2023-10 submission	5 days ago
 Fluo-C3DL-MDA23...	2023-10 submission	5 days ago
 Fluo-N3DH-CE-01.sh	2023-10 submission	5 days ago
 Fluo-N3DH-CE-02....	2023-10 submission	5 days ago
 Fluo-N3DL-DRO-01...	2023-10 submission	5 days ago
 Fluo-N3DL-DRO-0...	2023-10 submission	5 days ago
 Fluo-N3DL-TRIC-01...	2023-10 submission	5 days ago
 Fluo-N3DL-TRIC-0...	2023-10 submission	5 days ago
 Fluo-N3DL-TRIF-01...	2023-10 submission	5 days ago
 Fluo-N3DL-TRIF-02...	2023-10 submission	5 days ago

📄 README.md	2023-10 submission	5 days ago
📄 clean.sh	2023-10 submission	5 days ago
📄 compute_labels.py	2023-10 submission	5 days ago
📄 download.sh	2023-10 submission	5 days ago
📄 inference.py	2023-10 submission	5 days ago
📄 normalize.py	2023-10 submission	5 days ago
📄 register.py	2023-10 submission	5 days ago
📄 requirements.txt	2023-10 submission	5 days ago
📄 run_all.sh	2023-10 submission	5 days ago
📄 train.py	2023-10 submission	5 days ago
📄 train.sh	2023-10 submission	5 days ago
📄 unet.py	2023-10 submission	5 days ago
📄 utils.py	2023-10 submission	5 days ago
📄 vector_field.py	2023-10 submission	5 days ago
📄 view-submission.py	2023-10 submission	5 days ago

📖 README
☰

Ultrack's CTC submission

This repository contains the code to reproduce the results of the Ultrack submission to the Cell Tracking Challenge.

For more information about the challenge see <https://celltrackingchallenge.net>

For more information about Ultrack see <https://github.com/royerlab/ultrack>

Requirements

Requirements

- Docker: <https://docs.docker.com/get-docker/>
- NVIDIA GPU with at least 12 GB of RAM. Execution tested on 24 GB.
- Gurobi Web License Service (WLS) because it's running in a docker container. See <https://github.com/royerlab/ultrack/tree/main/docker#gurobi-support> for additional instructions.

Setup

1. Clone the repository

```
git clone https://github.com/royerlab/ultrack_CTC_submission
```

2. Change to the repository directory

```
cd ultrack_CTC_submission
```

3. Download pre-trained weights

```
bash download.sh
```

4. Build docker image

```
docker build -t ultrack_ctc .
```

Reproducing CTC submission

Once setup is complete, you can run the experiments by running the docker container.

```
docker run --rm -it --gpus all \  
  -v weights:/app/weights \  
  -v <YOUR_DATA_DIR>:/app/data \  
  -v <LARGE_DATA_STORAGE>:/wkdir \  
  -e WK_DIR=/wkdir \  
  -v <PATH TO YOUR GUROBI WSL LICENSE>:/opt/gurobi/gurobi.lic \  
  ultrack-ctc
```

The variable `$WK_DIR` and `/wkdir` are useful to store intermediate results, for the TRIF dataset this can be more than 500GBs, which otherwise will be saved in your main storage unit.

General usage

For general usage and application of our tracking algorithm on your own data see <https://github.com/royerlab/ultrack>.

Additional notes

To train your own model, see `train.sh` and `train.py`. This requires at least 24 GB of GPU memory depending on the dataset.

The configuration of the paper's remaining experiments can also be found there.

The "Candidate Segmentation Association" seems to be missing a formula.

→ The reviewer might have missed the formula because it is within the text. Here's it highlighted in our original submission below.

Candidate segmentation association. For each candidate segmentation, we compute the potential temporal associations (*i.e.*, tracks). Let \mathcal{H} represent the set of all segmentations derived from hierarchical segmentations. We define E^T as the set of pairs representing potential associations between these segmentations, with $w(\cdot, \cdot)$ denoting their respective weights. The association process involves the following steps: (i) For each segmentation $i \in \mathcal{H}$, where t denotes the time point of segment i , we compute its k -nearest neighbors within a predefined radius using their centroid coordinates at time $t - 1$; (ii) These pairs are then included in E^T ; (iii) The association score between any two segmentations i and j is calculated as $w(i, j) = \text{IoU}(i, j)^\gamma$. Where γ is consistently set to four throughout our study and IoU represents the intersection over union between the segmentation masks of i and j .

We made the formulation more descriptive so it's clearer to every reader:

Candidate segmentation association. For each candidate segmentation, we compute the potential temporal associations (*i.e.*, tracks). Let \mathcal{H} represent the set of all segmentations derived from hierarchical segmentations. We define E^T as the set of pairs representing potential associations between these segmentations, with $w(\cdot, \cdot)$ denoting their respective weights. The association process involves the following steps: (i) For each segmentation $i \in \mathcal{H}$, where t denotes the time point of segment i , we compute its **candidates 2k-nearest** neighbors within a predefined radius using their centroid coordinates at time $t-1$; (ii) **Optional step for additional filtering of neighbors, e.g. Color feature association Section.** (iii) The association score between every **segment i and their candidate neighbor j** is calculated as $w(i, j) = \text{IoU}(i, j)^\gamma$. **Where**

$$\text{IoU}(i, j)^\gamma = \left(\frac{S_i \cup S_j}{S_i \cap S_j} \right)^\gamma$$

and S_i is the binary mask of region i (*i.e.* segmentation hypothesis) and γ is consistently set to four throughout our study; (iv) Include into E^T the k pairs per segment in t with the largest IoU, using their distance as a tie-breaker.

The code is available at the links provided but actually using this massive codebase to DO what the authors did is not possible at the level of detail currently in the methods section. I would strongly encourage the Authors to reference more than just the topline repo in their various methods sections to enable users to actually recapitulate the Authors work.

→ We acknowledge the reviewer's concerns, and we addressed this comment in two ways:

(i) We included additional details of our work, such as an improved description of candidate linking and multi-color linking procedure, and information about the UNet architecture and training setup.

(ii) We set up a new repository that organizes and recapitulates the content produced in this paper: https://github.com/royerlab/ultrack_supplementary. This repository contains information about the hardware used, where to find and the Python and Ultrack configuration to achieve similar results as our experiments, and the scripts we used to generate the data for the figure making.

For example, the repository contains a hyperlinks to the scripts of used for the manuscript figures:

ultrack_supplementary / configuration /		↑ Top
..		
CTC @ fa6a20a	renamed experiments to configuration	5 days ago
flow_registration	flow registration configuration	5 days ago
large_zebrafish	updated torch load to use jit	5 days ago
multi_color	multi color description	5 days ago
parameter_sweep	add parameter sweep README	2 days ago
sparse_zebrafish	updated torch load to use jit	5 days ago
.gitignore	updated torch load to use jit	5 days ago
README.md	Update README.md	yesterday

README.md 
Configuration index

This file and its subdirectories organize the setup and execution configuration of Ultrack's experiments.

Our goal here is to describe how to show the pipeline executed in our experiments, so any user can easily modify them for their own applications. It's not our goal to fully reproduce the experiments as most of them are entangled with the data and the environment in which they were executed.

The experiments being, using our preprint as a reference:

- Table 1
- Figure 2
- Figure 3
- Figure 5
- Figure 6
- Figure 7

As an example, for the multi-color ensemble of Cellpose and watershed tracking, Fig. 3, it refers to Ultrack's main repository documentation, where we provide files to setup a conda environment with or without GPUs and a detailed description of what was done using and why in a Jupyter notebook.

The `README.md` file content is as follows:

Multi-color tracking with ensemble

See jupyter notebook in ultrack's repo for demonstration of multi-color tracking without a multi-color segmentation model, URL: https://github.com/royerlab/ultrack/blob/main/examples/multi_color_ensemble/multi_color_ensemble.ipynb

Two conda environments configurations are provided, a CPU only and GPU accelerated.

For setup instructions refer to our examples instructions, here.

Conda environment file:

main

ultrack / examples / multi_color_ensemble / environment_gpu.yml

ilan-theodoro and JoOkuma Improve API notebook example (#70)

Code

Blame

21 lines (21 loc) · 291 Bytes

```
1 name: ultrack-multi-color
2 channels:
3   - gurobi
4   - rapidsai
5   - conda-forge
6   - defaults
7 dependencies:
8   - cellpose
9   - coin-or-cbc
10  - cucim
11  - cupy!=13.*
12  - gurobi
13  - jupyter
14  - pip
15  - pyqt
16  - seaborn
17  - pip:
18    - napari==0.4.18
19    - traccuracy
20    - napari-arboretum
21  - pyift
```

Section of Jupyter notebook with hyperparameter configuration and descriptions:

```
ultrack / examples / multi_color_ensemble / multi_color_ensemble.ipynb
```

```
10.4 MB
```

```
# Combining watershed results
detection, contours = labels_to_contours(
    [cellpose_labels[...], c] for c in range(cellpose_labels.shape[-1]) + \
    [ws_labels[...], c] for c in range(ws_labels.shape[-1]),
    sigma=5.0,
    detection_store_or_path=zarr.TempStore(),
    edges_store_or_path=zarr.TempStore(),
)

Applying watershed_seg...: 100%|██████████| 900/900 [06:19<00:00, 2.37it/s]
<ipython-input-9-e45de74e19eb>:14: DeprecationWarning: Argument detection_store_or_path is deprecated, please use
foreground_store_or_path instead.
detection, contours = labels_to_contours(
Converting labels to contours: 100%|██████████| 300/300 [00:14<00:00, 21.04it/s]
```

Finally, we can track the cells. Ultrack parameters are defined in the `MainConfig` class. Each step contains its own parameters in a subconfiguration (e.g., `config.linking_config`). These parameters were found by inspecting the tracking results.

The configuration documentation can be found here.

```
In [10]: config = MainConfig()
n_workers = 1 if COLAB else 8

# Candidate segmentation parameters
config.segmentation_config.n_workers = n_workers
config.segmentation_config.min_area = 250
config.segmentation_config.min_frontier = 0.01

# Setting the maximum number of candidate neighbors and maximum spatial distance between cells
config.linking_config.max_neighbors = 5
config.linking_config.max_distance = 50
config.linking_config.n_workers = n_workers

# Tracking integer linear programming (ILP) parameters
config.tracking_config.division_weight = -0.01
config.tracking_config.disappear_weight = -5
config.tracking_config.appear_weight = -0.5

# ILP processing window size.
# It reduces memory usage while asserting continuity in the tracks.
config.tracking_config.window_size = 15
config.tracking_config.overlap_size = 3
config.tracking_config.solution_gap = 0.01

print("ultrack config")
print(config)
```

The “Color features association” section in the Methods section is of particular concern. $\mu(i)$ was not defined, I assume $\mu(i)$ is the mean intensity vector of neighboring segment $S(i)$ at time $t-1$. But it should be explicitly defined. Why was a threshold of 3 chosen? As described, you’re saying for every object in a reference frame, you want to subtract the mean intensity of a candidate in the next frame (However, because there’s no specification on candidate here I assume all objects at $t-1$ are candidates?) But then the max value doesn’t make sense, so perhaps it’s all time steps, but if that is true there is no time variable anywhere in the equation?

I think what the Authors are trying to say is that they are taking a difference in means of an object and a candidate, and dividing that by the frame of reference object intensity standard

deviation. The Authors are then taking the absolute value of that number and determining if it is less than 3, if it is then that object is kept as a candidate in the ILP solve. Otherwise the Authors exclude it. I think the Authors are further doing this pairwise comparison across all objects at t-1 with each object in the reference frame. This is quite computationally in-efficient as its a one to many comparison. Wouldn't it make more sense to set a distance threshold of candidate options in frame t-1 so that EVERY object in the frame isn't compared? Also, I'm not confident my guess is correct because the written equation is still taking the max of an absolute value, which indicates to me that it is a vector of values. This might be for a given object and a given candidate at all pairwise time steps? But then the Authors make the assumption that the candidate is the reference object for any time step t-2 and beyond. Which further complicates/expands the computational inefficiency of the calculation. No matter what, the Authors need to dramatically rewrite this formula and explanation.

Further, even if the above is clarified, it is unclear to me why the Authors divide the change in average intensity only by the reference frame intensity standard deviation? Its close to a z-score, but a delta mean intensity divided by the standard deviation of pixels that might or might not be related to one of the object means (in the case where the candidate object is, in fact, not the reference) has no physical meaning/reference that I can think of. Had you of taken the z-score of each object, subtracted them, and looked at the absolute value of that and set it less than 3 I think there's distributional relationships we could make based off known z-score relationships to probability distribution functions. But as it stands, I can't understand why you chose this approach and why it's meaningful.

→ We appreciate the feedback on this section's lack of clarity. We added more content to this section to avoid the misunderstanding encountered by this reviewer.

Before addressing individual parts of the comment from the reviewer, we would like to highlight that this is an additional filtering step before the ILP-based tracking. Therefore, the goal is not to find the linkage but to remove highly unlikely candidates.

Without further ado:

"I think what the Authors are trying to say is that they are taking a difference in means of an object and a candidate, and dividing that by the frame of reference object intensity standard deviation. ... Further, even if the above is clarified, it is unclear to me why the Authors divide the change in average intensity only by the reference frame intensity standard deviation?" - Review #1

The comment above is related to the phrase from our main text, "... where μ_j is the mean intensity vector within the mask of a segment S_j at time t and σ_j the standard deviation ..."

We improved our text and included the exact mathematical formulation, hopefully clarifying that μ and σ are vectors in the color space and their estimations are done **within** each segmentation hypothesis mask. We only divide by the reference candidate segment standard deviation because we compare how well its neighbor fits into the reference color distribution. For reference, we included below the update section:

Color features association. The image intensity information can be used to filter out segments between adjacent frames that are unlikely to represent the same cell. To do that, we compute a normalized distance between the average intensities of segments, **akin to a standard normal distribution z -score**; if this distance exceeds a predefined threshold z , their association is excluded from the tracking Integer Linear Programming (ILP) formulation. Specifically, the relationship must satisfy the following conditions:

$$\max \left| \frac{\mu_i - \mu_j}{\sigma_j} \right| \leq z \quad (13)$$

where μ_j is the mean intensity vector within the mask of a segment S_j (**represented as set**) at time t and σ_j the standard deviation. The operations are done element-wise **per channel**, and \max results in a scalar from it, here, compared to a neighboring segment S_i at $t-1$. **Therefore, j is the reference node and i is the candidate neighbor node.** We formalized the elements in the color vectors as,

$$\mu_i = \frac{1}{|S_i|} \sum_{p \in S_i} I_p \quad (14)$$

$$\sigma_i = \sqrt{\frac{1}{|S_i|} \sum_{p \in S_i} (I_p - \mu_i)^2} \quad (15)$$

where I_p is the intensity values at voxel p , such that for an 3-channeled image I_p is a 3-dimensional vector.

The motivation behind this formulation is the need for an intensity-invariant distance between colored cells. Hence, using a z -score formulation was a natural choice because the threshold z can be interpreted as how many standard deviation color distribution *within* a cell differs from others. The use of \max is a conservative choice, so that if any channel, *independently* exceeds the threshold z , the candidate link is rejected.

This can be efficiently computed by making the pairwise comparison only between the $2k$ candidate pairs of each segment because the default linking already excluded the rest. Thus reducing the $N_{t-1} \times N_t$ comparisons to $N_t \times 2k$, where k is the number of candidate links defined at the linking step.

The threshold z was set to 3.0 throughout the multi-color experiments, **a symmetric quantile threshold equivalent to a 0.0027 probability of under our z -score estimation.**

“The Authors are then taking the absolute value of that number and determining if it is less than 3, if it is then that object is kept as a candidate in the ILP solve. Otherwise the Authors exclude it. I think the Authors are further doing this pairwise comparison across all objects at t-1 with each object in the reference frame. This is quite computationally in-efficient as its a one to many comparison. Wouldn't it make more sense to set a distance threshold of candidate options in frame t-1 so that EVERY object in the frame isn't compared?” - Review #1

The reviewer is correct, $N_i \times N_j$ comparisons, where N_i is the number of objects in frame i , would be computationally expensive. Therefore, we only evaluate the color space difference between nodes within $2k$ candidate neighbors in the adjacent frames computed from the spatial nearest neighbor linking procedure, greatly reducing it to $N_i \times 2k$ comparisons (k is the number of candidate links defined by the user) as described in the excerpt from the previous reply.

Code and data availability seems to include all code and data needed to do the analyses discussed, but not the code to generate the figures from the specific datasets that generated those figures (which I was not able to find those specific datasets, but I think they exist, there are just many nested directories without clear documentation on what is the appropriate place to look for the data/graphing code for each figure.)

→ Most of the figures were manually composed using vector graphics software. However, we do have scripts to obtain images, screenshots, and movies. We included these Python scripts in the `figures` folder inside the new https://github.com/royerlab/ultrack_supplementary repository. These scripts are:

```
Unset
ultrack_supplementary/figures
├── constants.py
├── flow_registration
│   ├── flow_field_crop_comparison.py
│   ├── flow_field_crop.py
│   ├── flow_field_image.py
│   └── flow_field_video.py
├── large_zebrafish
│   ├── zebrafish_intensity_profile.py
│   ├── zebrafish_lineage.py
│   ├── zebrafish_presentation_figure.py
│   └── zebrafish_thumbnails.py
├── multi_color
│   ├── multi_color_constants.py
│   ├── multi_color_crops.py
│   ├── multi_color_figure.py
│   ├── multi_color_lineage_videos.py
│   ├── multi_color_moving_slice_video.py
│   ├── multi_color_single_track_video.py
│   └── multi_color_video.py
├── neuromast
│   ├── neuromast500_events_script.ipynb
│   └── neuromast500_thumbnails_script.ipynb
```

```

|   └─ neuromast500_TRA_scores.ipynb
├─ parameter_sweep
|   └─ parameter_sweep.py
|   └─ parameter_sweep_video.py
├─ README.md
├─ sparse_zebrafish
|   └─ sparse_fancy_video.py
|   └─ sparse_figure.py
|   └─ sparse_lineage_comparison_video.py
|   └─ sparse_video.py
├─ ui
|   └─ screenshot_ui.py
├─ utils.py
└─ virtual_staining
    └─ virtual_staining_3D_lineage.py
    └─ virtual_staining_crops.py
    └─ virtual_staining_previews.py
    └─ virtual_staining_videos.py

```

In the “Label conversion to contour and foreground map” section the Authors state “this averaging approach, as opposed to summation, keeps pixel values within the 0 to 1 range.” However, so would summing and normalizing. As would many other approaches. Why did the Authors choose this approach? What was the rationale? What else was tried and how did it perform?

→ Indeed, summing with or without normalizing can yield the same results. Therefore, we don’t mention normalization and other transformations in the text.

Regarding other approaches and our rationale for using the average. We chose the average because of its simplicity and because our multi-level contour approach to reconstructing the hierarchical segmentation does not care about the absolute values of the input image topology (i.e., contour map), only their ordering. Therefore, any monotonic function will produce the same result, and the choice of monotonic function does not matter. Therefore, summing (even without normalization) and averaging produce the exact same result.

The benefit of keeping the values within a fixed range, here chosen to be between 0 and 1, is to facilitate downstream analysis or filtering over the contour map.

We now included this in the text:

Label conversion to foreground and contours map. The conversion of multiple segmentation labels into foreground and contours maps involves two main processes. For the foreground map generation: (i) each label is transformed into a binary map, with non-zero pixels assigned a value of one (foreground) and zero pixels remaining as background; (ii) these individual maps are merged into a single foreground map by taking the maximum value across corresponding pixels from all **individual foreground maps**; (iii) this logical OR operation ensures that any pixel belonging to at least one segment is included in the final foreground map. For the contour map generation: (i) to preserve each label and its dual contour, we process each label map to create a binary contour map, where pixels are set to one if their neighboring pixels have differing labels, indicating a boundary; (ii) the final contour map is then created by averaging these individual binary contour maps — **summing would yield the same results because the hierarchical segmentation reconstruction is based on ordering of values and not their absolute values, the benefit of averaging is to keep** pixel values within the 0 to 1 range.

D. Appropriate use of statistics and treatment of uncertainties

1. Throughout the manuscript the authors use relatively biased measures of performance (false negative rates, explicit definitions of region based false positive/negative, edges errors, etc.). Interestingly, from the methods it appears as if they use less biased segmentation assessment tools to train their networks (things like Jaccard) but in figures rely on a small number of biased measures. I would strongly encourage the authors to look into not only accuracy, but less biased region-based assessments like Fowlkes–Mallows index, IoU/Jaccard, and/or F1 score. These have been reported in many publications but a comprehensive set of them can be found at <https://doi.org/10.1186/s12859-023-05486-8>.

→ First, we would like to thank the reviewer for the extremely helpful reference. We were not aware of this work.

Where appropriate, we updated our metrics to include the F1. The Fowlkes-Mallows index and Jaccard results were extremely similar to F1 and did not provide new insights and thus were omitted due to space restrictions. Additionally, we chose not to use IoU/Jaccard (between matches/ROI) to avoid confusion with the IoU which is used between segments (pixel measure as reported by the article), which is used thoroughly used in the paper. This won't lead to less information since F1, Fowlkes-Mallows, and Jaccard strongly correlate. The figures were updated accordingly, and our conclusions remain the same.

Using these metrics for extremely sparse annotations was impossible, as we cannot estimate false positives; see more about this in the following reply.

Thus, the updated plots are shown below, with F1-scores plots highlighted in red.

Fig. 2d

Fig 3d

Fig 4c

Fig 8d

The methods section for metrics was update to include:

From the segmentation detections obtained through the CTC procedure, we also computed the F1-score, where

$$\text{F1-score} = \frac{2n_{TP}}{2n_{TP} + n_{FN} + n_{FP}}$$

which provides a less-biased assessment of performance¹⁰⁰ that combines the evaluation of both n_{FN} and n_{FP} .

Additionally, in Figure 6D it seems there are three objectives that could be evaluated in a confusion matrix (1) Correct identification of connections between cells Y/N (2) Division Y/N (3) Correct identity Y/N. Then you could do unbiased assessments (Jaccard/F1/etc.) for each of those and take averages of those across the time periods you indicate using the deep vs not segmentation approaches.

→ As described by (C. Malin-Mayor et al. 2023), who proposed the sparse annotation metrics, the number of computable metrics is limited because we **cannot estimate false positive edges (FP)**. The reason being that we do not know if they are mistakes or missing edge annotations. Therefore, a confusion matrix that requires true positives, true negatives, **false positives**, and false negatives is not possible. The same is true for Jaccard and F1.

2. All assessments were repeated one time (albeit on MANY cells within that one run). While I understand the HUGE computational complexity and thus associated costs, repeatedly in the literature it has been shown that deep learning is probabilistic not deterministic and therefore any workflow that uses deep learning (which I believe is all of them for this manuscript) should be repeated multiple times to gain a confidence interval for performance. Ideally, re-training networks from scratch using the same training data would happen to make even more robust estimates of performance/generalizability of approach, however I think at the scales of data discussed in this manuscript that is infeasible. The ILP method proposed relies heavily on these candidate segmentations and therefore, I think the Authors will need to, at the minimum, run the segmentation pipelines multiple times with their deep learning models (both trained and community models.)

→ The training of our deep learning segmentation models, as in the large Zebrafish (Figure 7) and Neuromast (Figure 8), was done iteratively by predicting the network results, filtering or curating the annotations, and retraining, which is extremely common in our field (as <https://doi.org/10.1242/dev.202800>). So in a sense, we did retrain the algorithm multiple times, and in general it would not be possible to “repeat training runs” due to its iterative and interactive nature of our “runs”. More generally, we acknowledge the reviewers' concern and the good practices of benchmarking a new neural network architecture or loss function. However, in our work, we are concerned with segmentation and tracking which is a subsequent step to the network inference. Most experiments using off-the-shelf deep learning models can be run in deterministic mode https://pytorch.org/docs/stable/generated/torch.use_deterministic_algorithms.html. Therefore, the stochasticity of the inference is not a problem. More fundamentally, our score margins are typically wide enough and computed on large enough datasets that we don't expect our conclusions to be affected. Finally, our tracking algorithm is deterministic when the ILP is solved up to zero GAP between the estimated upper bound.

E. Conclusions: robustness, validity, reliability

In several locations (abstract, introduction, and discussion) in the document the Authors highlight the utility/value of the multi-level contours so that training/fine tuning of deep learning models is not needed. But ironically, for all large 3D cases the Authors train their own custom UNet for segmentation and show superior performance to using intensity contour proxies. I think these areas should be reworded to better reflect the results found in this manuscript.

→ We thank the review for the attentive comment, we reworded a few sections to avoid claiming that Ultrack with an off-the-shelf or any other source is better than a specialized solution. Our point is more that ultrametric contours are indeed very helpful in making simpler (classical image processing or deep-learning without fine-tuning) approaches more effective, which can in some applications be sufficient and avoid the burden of annotating and training your own deep learning model.

Of course, for more challenging applications where one must ‘pull all stops’, a user might want to take a more specialized route and invest the time to train or fine-tune a custom deep learning model. In that case, Ultrack is also valuable tool, and the benefits of ultrametric contours are also at play. This is typically the case for large 3D datasets for which the availability of pre-trained models is limited.

Interestingly, and at first surprisingly, we would also like to highlight that when training data is not available, segmenting and tracking using mult-level (ultrametric) contours directly derived from image intensities gives results that are comparable to deep learning. The examples being: (i) Figure 6, where our intensity-based method obtained lower error than the deep-learning-based baseline; and (ii) the cell tracking challenge, where the Drosophila and the Tribolium datasets submissions used the intensities directly. Both applications don’t require training/fine-tuning and are 3D datasets – classifying them as ‘large’ is subjective, but the Tribolium is 400G, and Figure 6 is 5 TB.

F. Suggested improvements: experiments, data for possible revision

Figure 1 needs to have an example of a foreground map. And the vernacular used for referring to foreground maps should be made consistent both in the figure and throughout the manuscript. The Authors refer to it in several different ways (segmentation map, foreground mask, etc.).

→ We included a foreground map in the main figure and standardized the vernacular to “foreground map(s)” and, when permitted, only “foreground”.

For example, the caption of Figure 4 shows standardization from “foreground masks” to “foreground maps”:

Fig. 4 | Ultrack improves cell tracking in label-free imaging by leveraging virtual staining. **a**, Pipeline for intensity-based joint segmentation and tracking from label-free quantitative phase imaging: Quantitative phase image (top left) is processed by VisCy⁵⁷ to generate virtual staining of nuclei and membrane (second from left). These virtual stains are used to derive contours (third from left) and foreground **maps** (not shown) through filtering, subtraction, and thresholding. Ultrack uses these inputs to perform tracking (right). **b**, We compare the above against first applying a Cellpose segmentation to the quantitative phase image (bottom left), then deriving contours (and foreground **map**, not shown), and then applying Ultrack (right). False positives (over-segmentation) are indicated in magenta, and false negatives in blue. **c**, Quantitative comparison of tracking errors using Cellpose⁵⁸ segmentations (gray) versus virtually stained membranes (orange) as input. The bar chart shows the **F1-score of detections, upper y-axis** and the number of false negative (FN) edges, false negative (FN) detections, and false positive (FP) detections for each method, **lower y-axis**.

G. References: appropriate credit to previous work?

Generally, the Authors do a good job of citing relevant work. However, a few references 36, 38, 46, 47, 59, 78, 87, 88, 98, and 101 are not well enough specified/defined to be able to find/know what the Authors are referring to.

→ Indeed, some of the references were not very helpful, these were online content that hadn't been published in traditional academic venues. We updated the reference style to include more information, mainly their online URL. We adjusted all mentioned references, as the example below:

Before:

- [36] N. Sofroniew, T. Lambert, K. Evans, J. Nunez-Iglesias, G. Bokota, P. Winston, G. Peña-Castellanos, K. Yamauchi, M. Bussonnier, D. Doncila Pop, A. Can Solak, Z. Liu, P. Wadhwa, A. Burt, G. Buckley, A. Sweet, L. Migas, V. Hilsenstein, L. Gaifas, J. Bragantini, J. Rodríguez-Guerra, H. Muñoz, J. Freeman, P. Boone, A. R. Lowe, C. Gohlke, L. Royer, A. Pierré, H. Har-Gil, and A. McGovern, “napari: a multi-dimensional image viewer for Python.”
- [37] D. Ershov, M.-S. Phan, J. W. Pylvänäinen, S. U. Rigaud, L. Le Blanc, A. Charles-Orszag, J. R. Conway, R. F. Laine, N. H. Roy, D. Bonazzi, *et al.*, “Trackmate 7: integrating state-of-the-art segmentation algorithms into tracking pipelines,” *Nature Methods*, vol. 19, no. 7, pp. 829–832, 2022.
- [38] T. Pietzsch and J.-Y. Tinevez, “Mastodon.”

After:

- [36] N. Sofroniew, T. Lambert, K. Evans, J. Nunez-Iglesias, G. Bokota, P. Winston, G. Peña-Castellanos, K. Yamauchi, M. Bussonnier, D. Doncila Pop, A. Can Solak, Z. Liu, P. Wadhwa, A. Burt, G. Buckley, A. Sweet, L. Migas, V. Hilsenstein, L. Gaifas, J. Bragantini, J. Rodríguez-Guerra, H. Muñoz, J. Freeman, P. Boone, A. R. Lowe, C. Gohlke, L. Royer, A. Pierré, H. Har-Gil, and A. McGovern, “napari: a multi-dimensional image viewer for Python.” [Online]. Available: <https://github.com/napari/napari>
- [37] D. Ershov, M.-S. Phan, J. W. Pylvänäinen, S. U. Rigaud, L. Le Blanc, A. Charles-Orszag, J. R. Conway, R. F. Laine, N. H. Roy, D. Bonazzi *et al.*, “Trackmate 7: integrating state-of-the-art segmentation algorithms into tracking pipelines,” *Nature Methods*, vol. 19, no. 7, pp. 829–832, 2022.
- [38] T. Pietzsch and J.-Y. Tinevez, “Mastodon.” [Online]. Available: <https://github.com/mastodon-sc/mastodon>

H. Clarity and context: lucidity of

abstract/summary, appropriateness of abstract, introduction and conclusions

The manuscript was very well written. A very minor correction was that a lagging “l” was left on page 8 the very first sentence. Other than that, the grammar and writing was very clear (with exceptions in the methods described above).

→ Thank you, we fixed the typo.

On page 5 the Authors use the term “color features” in the Enhancing cell tracking accuracy through multi-channel integration and color-aware associations.” Section however it is not defined yet. Please define or at least refer to the section of the methods where it is defined.

→ We updated the manuscript to refer to the Methods section, as suggested by the reviewer, and improved the description of the color features.

the more sophisticated Cellpose output. Furthermore, UI-track can use the three color channels as a feature when associating segments between frames, which helps connect the correct segments across time (Fig. 3b), **see Methods for description of the color features association.**

On page 19 that statement “allowing the algorithm to ignore background regions and thereby greatly speeding up the process” is only true in non-dense object situations. I would ask the Authors to caveat this statement with the conditions under which it is valid.

→ We included the caveat that this speed-up is greater for non-dense labeling (e.g. nuclei). See below:

have been processed. This segmentation process operates within each connected component of the binary foreground **map**, allowing the algorithm to ignore background regions and thereby greatly speeding up the process **in non-dense labeling (e.g. nuclei)**. For further details on hierarchical segmentation processing and implementation, see.⁴⁵

Code quality and availability:

Code quality itself is well above average and I was able to (eventually) get many of the libraries and CLIs to run, this was non-trivial for GPU implementations. I would strongly encourage the authors to deliver containerized versions (ideally both docker and aptainer/singulairy) of CLIs and Python libraries, especially for GPU versions.

→ We are thankful for the recognition of our implementation efforts so it can used by a broad community. Following the reviewer's suggestion, we created cpu and gpu **docker images**. They are publicly available, so any user can build their own or modify them:

<https://github.com/royerlab/ultrack/tree/main/docker>

They are also available in our dockerhub so they can be easily installed without having to build your own. Its documentation can be found at

<https://royerlab.github.io/ultrack/docker/README.html>

The documentation provided was sufficient for me to run these libraries and CLIs on some of my own data. However, I would strongly suggest the Authors create a repo devoted to JUST the recapitulation of the results from this paper. Trying to connect all the software/steps with what was done in each figure with the correct dataset, ensuring test/train splits, etc. was essentially outside of my capabilities. Its not that I couldn't do it eventually (I think) but data, code, and tooling is so distributed and naming not consistent between the manuscript and the repos that I just ran out of patience. I believe most/all of it exists in what the Authors have provided, but its just too difficult to recapitulate the actual results shown at the current documentation level.

→ We created a new repository, https://github.com/royerlab/ultrack_supplementary, as suggested by the reviewer. This repository covers hardware information, figure generation script, and an experimental configuration index, which organizes where the notebook describing them, their python requirements, and parameters can be found, we show below the main page of the repository:

JoOkuma Update README.md		a15601f · 3 days ago	🕒 25 Commits
📁 configuration	Update README.md		3 days ago
📁 figures	minor changes to neuromast figure scripts		last week
📁 hardware	add hardware description		last week
📄 .gitmodules	renamed experiments to configuration		last week
📄 README.md	updated readme		last week
📖 README			✎ ☰

Ultrack supplementary information

This repository provides supplementary information for the paper Ultrack: pushing the limits of cell tracking across biological scales.

Our goals with this, is to provide a centralized location for the following information:

- Hardware: Describes the hardware used in the experiments.
- Figures: Scripts to generate content for the figures in the paper.
- Configuration: Python package and parameters used in Ultrack's experiments.

For Ultrack's execution and usage refer to our main repository Ultrack.

Citing

```
@article{bragantini2024ultrack,
  title={Ultrack: pushing the limits of cell tracking across biological scales},
  author={Bragantini, Jordao and Theodoro, Ian and Zhao, Xiang and Huijben, Teun APM and Hira},
  journal={bioRxiv},
  pages={2024--09},
  year={2024},
  publisher={Cold Spring Harbor Laboratory}
}
```

But to reiterate, the documentation provided is sufficient for me to use these tools and apply them to my own data, just not to recapitulate the results of this paper.

→ Once again, we are thankful to the reviewer for recognizing our effort.

Reviewer #2 (Remarks to the Author):

In this study, Bragantini et al. developed Ultrack, a scalable cell tracking tool that simultaneously performs segmentation and tracking. The tool leverages segmentation hypotheses from various algorithms, simplifying parameter tuning while improving accuracy. By solving an Integer Linear Programming (ILP) problem, Ultrack optimizes both segmentation and tracking. It proves effective in densely labeled imaging, as demonstrated by a dual-color labeling approach where one channel is sparsely labeled, facilitating manual curation. This method enhances tracking accuracy, outperforming state-of-the-art techniques with fewer cumulative errors.

Two key in vivo applications were demonstrated: in developing zebrafish embryos, Ultrack scaled up to terabyte-scale data, tracking developing cells with error-free results over dozens of time points using either high-performance computing platforms or a desktop. The second application focused on developing neuromast cells, accurately capturing cellular migrations, divisions, and cell deaths over tens of hours.

The manuscript benchmarks Ultrack's performance across multiple datasets, particularly in comparison to tools like TrackMate. It demonstrates seamless integration with platforms such as Fiji and napari, along with high scalability from personal computers to high-performance computing clusters, which will be a key advantage for users with varying programming expertise. Its innovative approach to handling hard-tuned parameters, uncertainties, and challenges in large datasets makes it a valuable contribution to the field of cell tracking, particularly for extracting meaningful insights from complex biological datasets.

→ We are extremely thankful for the positive comments regarding our method.

Major Concerns:

1. Although the applications demonstrate superior performance in terms of accuracy, speed, and scalability compared to other methods, the manuscript would benefit from a clearer emphasis on the biological insights enabled by Ultrack. It is not enough to simply showcase improved methodologies; the authors should highlight cases where the enhanced accuracy and throughput of Ultrack reveal biological phenomena that would otherwise go unnoticed. Demonstrating how these capabilities contribute to deeper biological understanding will strengthen the impact of the work.

→ We agree that new methods have to show their value by pushing the limits of what is currently possible. We do think that our current manuscript does that, showing how we can track with state-of-the-art performance on several datasets. Moreover, Ultrack's other appeal is that it enables tracking at scales which were previously unattainable. This in turn, enables us to push the analysis horizon and enables new biological understanding. The main reason we developed ultrack was to be able to track cells in zebrafish embryos at scale and at a much higher level of accuracy than was previously possible. This is primarily demonstrated in our recently published companion paper:

- <https://doi.org/10.1016/j.cell.2024.09.047> : Lange, Merlin, et al. "A multimodal zebrafish developmental atlas reveals the state-transition dynamics of late-vertebrate pluripotent axial progenitors." *Cell* 187, no. 23 (2024): 6742-6759.

This work (zebrahub) makes use of ultrack to make significant contributions in our understanding of specific progenitor populations (NMPs & hemangioblasts), contributing to reveal previously unknown biological phenomena. In addition, we are happy to report that multiple recent preprints have been using ultrack successfully, and we are aware of more works to come:

1. <https://doi.org/10.1101/2025.01.06.631621> : Adhinarta, Jason, et al. "WormID-Benchmark: Extracting Whole-Brain Neural Dynamics of *C. elegans* At the Neuron Resolution." *bioRxiv* (2025): 2025-01.
2. <https://doi.org/10.1101/2024.01.03.574108> : Eck, Elizabeth, et al. "Single-cell transcriptional dynamics in a living vertebrate." *bioRxiv* (2024).
3. <https://doi.org/10.48550/arXiv.2410.11281> : Pradeep, Soorya, et al. "Contrastive learning of cell state dynamics in response to perturbations." *arXiv preprint arXiv:2410.11281* (2024).
4. <https://doi.org/10.1101/2024.09.23.614351> : Hirata-Miyasaki, Eduardo, et al. "High-speed 3D Imaging with 25-Camera Multifocus Microscope." *bioRxiv* (2024): 2024-09.

In particular, we would like to highlight item 1, which is a work completely independent from us. This preprint shows Ultrack's contribution to an adjacent field of study, namely here the study of *C. elegans* neural activity (screenshots from their preprint):

Method	$d_{th}=3\mu m$		$d_{th}=6\mu m$	
	DET	TRA	DET	TRA
Ultrack Bragantini et al. (2023)	0.70±0.04	0.66±0.05	0.81±0.05	0.78±0.05
3DeeCellTracker Wen et al. (2021)	0.28±0.04	0.26±0.03	0.34±0.05	0.32±0.04
3DeeCellTracker [†] Wen et al. (2021)	0.43±0.07	0.40±0.07	0.51±0.09	0.47±0.08

Table 4. Neuron tracking results (Task 3). We report metrics using two different distance thresholds on both pre-trained and re-trained ([†]) models using 5-fold cross-validation.

Quantitative results. We report our metrics in Tab. 4. **Ultrack** consistently achieves higher tracking and detection accuracies on both distance thresholds compared to 3DeeCellTracker across the datasets, due to a relatively high number of detections in each time frame in line with the number of neurons in the volume. We hypothesize that using the "ensemble"-mode provided by 3DeeCellTracker can increase its performance.

We hope that this is overall enough evidence of the practical value that ultrack's brings to the field.

2. The manuscript's claims regarding the tool's generalizability are not fully convincing, and its application conditions are not described subjectively enough. For instance, in Figure 2b, the "all"

condition seems unrealistic. If cells move across the entire frame and the foreground is fully occupied, using logical OR for combining segmentations would not be effective. The authors should better explain the limitations and recommend suitable conditions for its applicability.

→ It is true that our algorithm relies on some assumptions regarding the time resolution. While we can partially address this concern using flow field registration (see Figure 5), fast moving cells can still be challenging in extreme scenarios. We have included a section on the limitations and assumptions of our method so users can better understand when our approach is the most appropriate. Here's the updated paragraph regarding the limitations related to fast-moving cells:

Despite its strengths, Ultrack has its limitations. First, the segmentation hypotheses are only as good as the provided foreground and contour maps. While Ultrack is robust to occasional segmentation errors, it may fail when presented with an overwhelming number of incorrect segmentations or systematic errors that persist over time — in such cases, the ILP optimization might converge on erroneous solutions. Additionally, as shown in Figure 5, large cell movements between adjacent frames violate our assumption of segmentation consistency and can impair tracking performance unless registration is used.

We are under the impression that the reviewer might have misunderstood what we meant by “All” in Figure 2b. To avoid potential confusion going forward, we renamed “All” to “All four” meaning “All four gamma parametrizations combined”. To clarify a point of potential confusion, we would like to emphasize that the foreground OR operation is done **per frame**, so fast-moving cells will not increase other frames' foreground regions because they are combined independently.

This hypothetical scenario of having the entire image as “foreground” would be an extreme case where most tracking algorithms would fail because that's equivalent to saying every pixel is a candidate cell. For example, tracking-by-assignment methods that do not use hypotheses would also have trouble associating these false-positive detections over time. It is also very important to realize that the input to Ultrack is **both a foreground and a contour mask**. The foreground mask's primary purpose is to distinguish cells from the background, and the contour splits the cells from each other. Hence, you can have large regions covered by the foreground mask, but you can still distinguish the cells from each other using the contours – as well as entertain multiple hypotheses – and background can be ignored when they create uncorrelated segments over time.

Minor Concerns:

1. The description of experimental conditions, such as the sampling rate's impact on tracking quality, could be improved. The absolute imaging conditions may not be informative enough across different microscopy setups and cellular processes. Instead of using absolute measurements (like Δt in Fig. 5), it would be more useful to describe how much spatial change occurs within a sampling interval. For instance, explaining how movements smaller than a certain proportion of cell size do not require registration would provide better intuition.

→ This was a great suggestion; we included a measurement of the cell motility distribution for each Δt in Figure 5 so it can be correlated with the change in accuracy.

Below is the updated Figure 5d, which includes the distribution of movements (in voxels) between frames, shown in gray, and the right-hand side y-axis, left-hand side y-axis, and colored lines are the TRA score as before:

Using the segmentation annotations available, we estimated the median diameter of a cell on the yx -plane to be approximately 19.64 voxels. Our accuracy starts decreasing when the movement distribution exceeds $2x$ these values at $\Delta t = 3$ frames, which is a considerable movement that can result in cell identity switching because the cell is moving more than its own size.

2. Terms like "without fine-tuning" appear multiple times in the manuscript. It would be helpful to clarify what this means in practice. Does it refer to using default settings, or does it involve some minimal adjustments like threshold tweaking? More precise descriptions would be beneficial.

→ We changed "fine-tuning" to "fine-tuning (i.e. deep-learning retraining)" or "deep fine-tuning" to make it more precise. This jargon was used with deep learning in mind, meaning retraining but not from scratch. We hope this disambiguates the usage since tuning in the context of image

processing parameters is often referred to as adjusting or estimating and, in classical machine-learning models as model fitting.

3. While the supplementary videos effectively convey complex processes, the figures and legends could be clearer to aid navigation. For example, in Figure 3b, the third and fourth columns should be swapped so that the most complete method (multi-color combined Ultrack) appears as the final step, aligning with the flowcharts.

→ Thanks for catching this mistake. We fixed the ordering of the third and fourth columns of the first row of Figure 3b.

4. In sparse labeling experiments, Tol2 injections could potentially damage transgene expression, leading to fewer H2B-mNeonGreen-expressing cells. The source of H2B-mNeonGreen should be clarified. If the gene was originally inserted via Tol2, the newly introduced Tol2 could disrupt its expression. A fair comparison with Malin-Mayor et al., 2022, should address any differences in the density of cells in densely labeled channels. Additionally, if cells expressing only the red label were observed, were these excluded from the analysis? Clarification on this point would be helpful.

→ Our Tg(ef1α:H2B-mNeonGreen) transgenic line was generated using Tol2 transposons, and we acknowledge that subsequent Tol2 mRNA injections could, in theory, remobilize integrated elements and potentially affect transgene expression (Kondrychyn et al., 2009). These are extremely rare events (less than 0.005 probability (0.5%) as seen below). Importantly, these cells are excluded from the ground-truth tracks, the reasoning that supports this conclusion are:

1. **Stable Integration and Promoter Context:** The ef1α promoter and the H2B-mNeonGreen coding sequence were cloned together as a transposable cDNA fragment to generate the stable transgenic line. Since we use homozygous Tg(ef1α:H2B-mNeonGreen) fish and their embryos, the transgene—including its promoter—is stably integrated into the genome. A new round of Tol2 mRNA-mediated insertion events is unlikely to readily remove or inactivate the promoter or disrupt the existing transgene's expression.
2. **Low Frequency of Remobilization Events:** Previous work has shown that only about 7% of Tol2 mRNA-injected heterozygous embryos exhibit remobilization-induced changes to transgene expression, and such changes are verifiable through germline transmission (Kondrychyn et al., 2009). Because our experiments were performed in homozygous embryos, the likelihood of remobilization events is even lower (estimated at approximately 0.49% – less than 1%).

5. In the neuromast tracking experiment, the manuscript should provide more details about the cell types being tracked. Are all the cells in the organ included, and do the two transgenic lines label the same cell populations? This information is important for understanding the scope and accuracy of the tracking.

→ We included this information in the manuscript. Yes, all the cells in the organ are included: hair, support, and mantle cells. The nuclei label, Tg(she:H2B-EGFP), is only expressed in the neuromast cells, while the membrane label, Tg(cldnb:lyn-mScarlet), is expressed in the cells in the neuromast, surrounding skin cells, and ionocytes. Therefore, our Cellpose model was trained to detect only the cells that express both the membrane and the nuclei marker. We updated the manuscript accordingly as shown below:

We imaged a zebrafish neuromast for 42 hours using membrane and nuclei markers. The nuclei marker, Tg(she:H2B-EGFP), only labels the neuromast cells (hair, support, and mantle cells), while the membrane marker, Tg(cldnb:lyn-mScarlet), labels the neuromast, surrounding skin cells, and ionocytes. Therefore, our segmentation model was trained to detect only cells that express both markers.

6. The manuscript does not discuss cell-entering events. While cell divisions and exits are demonstrated in Figures 3c and 8b, the manuscript lacks clarity on cell entry. Are these events categorized as divisions? Clear differentiation between these processes would be useful. Furthermore, if Ultrack can track cells that disappear and later reappear (due to out-of-frame movement), this capability should be emphasized, as it would restore continuous cell information in 2D or 3D imaging without dense z-axis sampling.

→ Our approach does handle explicitly cell appearance as described in the Section “Segmentation selection and tracking ILP” on Methods. This is also described in Figure 1. However, the variables were not explicit in the caption for Fig. 1, so we have now included that information, indicating which variables correspond to them and highlighted them in the formula in green this the screenshot of the figure below:

Fig. 1 | Ultrack overview. **a**, The Ultrack pipeline can process a variety of input images, including 2D, 3D, and multi-color datasets. These images are then processed by any segmentation algorithm or a combination of them. **b** Ultrack builds segmentation hypotheses between frames for tracking and solves an Integer Linear Programming (ILP) problem to identify cell segments and their trajectories. The resulting tracks can be exported in various formats compatible with multiple platforms. **c**, Illustration of how segmentation hypotheses are built using ultrametric contours: multiple segmentations are provided (first row), their binary contours are extracted (second row), and these are combined into a multilevel contour, equivalent to a hierarchy (third row). **d**, Joint segmentation and tracking is performed using an Integer Linear Programming (ILP) formulation, modeling cell behaviors (e.g. cell appearance $x_{,\alpha}$, division $x_{,\delta}$, death, or leaving the field of view $x_{,\beta}$) while finding non-overlapping cells with maximal association over time. **e**, Ultrack can process arbitrarily sized datasets and scales from a laptop to a High-Performance Computing (HPC) cluster. **f**, Projection of a 3D image of a developing zebrafish embryo, with cell tracks overlaid and colored to indicate track orientation in the xy-plane.

7. Is it possible to distinguish between error-prone and error-free tracking in Figure 7? While the data shows that around 1 hour or 62 frames yields 50% error-free tracks, this may not be sufficient for longer experiments spanning hundreds or thousands of frames, or even for leveraging the error-free tracks effectively. Given the impracticality of manual correction for such large datasets, automatically identifying the reliable 50% would be highly beneficial for further analysis and curation. Can Ultrac track already categorize tracks based on reliability?

→ That's an excellent question. When the mistakes are evident as excessive motion (jumps) between neighboring frames, they can be easily detected by inspecting the centroids' coordinate progression over time. Our software provides functions to compute these features, such as tracks_df_movement. In datasets with fewer cells, the lineage dendrogram can also indicate fake appearance and disappearance, and this was used to annotate the Neuromast dataset (Figure 8).

However, the general case of classifying correct and incorrect tracks is still an open problem and on-going research. If such method existed, most tracking algorithms could take it into account during tracking and iteratively improve their results.

Typos and others:

1. Missing period in "tracking cells in developing embryos".
2. "our baseline (0.078) even over a much more challenging 500-frame period" should be "their baseline".
3. In transgenic line names, it's better to use the format "Tg", e.g., tg(she:h2b-EGFP).

→ We appreciate the corrections. All of them were fixed

Code quality and availability:

The code has a clear README, installation instructions, and usage examples. It is easy to install and run locally and is well-documented and accessible for reproducibility and use by the community.

→ Thank you. We are glad our efforts are being helpful.

Reviewer #3 (Remarks to the Author):

In this manuscript, Bragatini et al. present an image analysis tool, called Ultrack, for tracking cells in time-lapse microscopy image data of various modalities and dimensionality. From the algorithmic point of view, Ultrack does not bring much novelty to field as its key principles, the hierarchical representation of multiple instance segmentation maps using ultrametric contours and an ILP-driven multiple hypothesis linking scheme, have already been published by the same authors in their very recent paper (Bragatini et al., ECCV 2024) referenced as [45]. However, from the practical point of view, Ultrack is a valuable contribution for the community as documented by extensive experimental results reported in this manuscript, and by deploying it in the form of a user-friendly add-on to popular Fiji and napari frameworks or of a command-line interface for high-performance computing environments.

Major comments

- In general, the RESULTS section combines a description of the main Ultrack's features with experimental results. Indeed, the first three subsections do not contain any experimental results, and thus shall be moved to the METHODS section.

→ We politely disagree. Since this is a methods paper in a **Methods** journal, the method itself (and its features) are the main contributions, and therefore results of our work. This is the standard as related work published in the venue, for example, the Model Design Section in Cellpose (<https://doi.org/10.1038/s41592-020-01018-x>), and others, scGPT (<https://doi.org/10.1038/s41592-024-02201-0>), CryoDRGN-ET (<https://doi.org/10.1038/s41592-024-02340-4>), etc.

Nonetheless, we are open to suggestions and additional input from the journal editors to make the manuscript more readable.

- RESULTS (Optimal parameter selection...): Please clarify how exactly Ultrack addresses computational and practical challenges mentioned in (i) and (ii), respectively. To this reviewer's understanding and as it is shown by numerous experiments conducted in this work, users must generate multiple segmentation inputs on their own and with their own knowledge of image processing and programming skills. Furthermore, the analysis of image data with hepatocarcinoma-derived cells is a proof of concept rather than a systematic study, based on which one could claim that a hierarchical combination of diverse segmentation inputs is more effective and robust in practice than a single, highly optimized solution. It would be more convincing to support this strong claim using more datasets. For example, one could take top-performing results for different datasets from the Cell Tracking Challenge benchmark and show that Ultrack can outperform all of them when combining their outputs together using ultrametric contours.

→ We will copy and break into parts the reviewers' comments to improve the clarity of our reply:

“Please clarify how exactly Ultrack addresses computational and practical challenges mentioned in (i) and (ii), respectively.” – Reviewer 3

→ For reference, the challenges (i) and (ii) referred by the reviewer are:

“Image segmentation techniques ... face significant challenges in parameter tuning due to several factors: **(i) computational**: extensive resources are required to explore a large parameter space; **(ii) practical**: users often lack the expertise to tune parameters effectively;” – from our submission

The presented use case – intentionally – minimally deviates from the standard non-expert user scenario of applying off-the-shelf Cellpose. Running Cellpose multiple times to obtain multiple segmentation hypotheses by changing the gamma parameter is rather trivial and only requires a marginal amount of work to iterate over a range of values (a for loop), which anyone with minimal Python experience can do. Users that are not comfortable with that could also use our napari or FIJI interface.

In contrast, a highly optimized solution as suggested by reviewer (deep learning, in our opinion) would require:

- Manually annotating segmentation and tracking labels;
- Converting their annotations to the adequate format for deep learning training;
- Training a neural network and cross-validating it to avoid overfitting;
- Optimizing the network architecture and training hyper-parameters;
- Testing the generalization on new or hidden test data. If it doesn't work, restart the process.

These steps of the highly optimized solution are exactly the challenges defined in items **(i) computational** and **(ii) practical** that we want to avoid while providing a satisfactory solution with off-the-shelf models, which can be used with minimal friction via our tool.

We hope this clarifies how Ultrack addresses items **(i)** and **(ii)**.

“... users must generate multiple segmentation inputs on their own and with their own knowledge of image processing and programming skills ...” – Review 3

→ As mentioned above, this might not require significant knowledge of image processing. Nonetheless, multiple segmentations are optional, while our experiments shown it benefits the tracking and segmentation performance, users can still provide a single segmentation input as is done with other tracking algorithms. Moreover, we are unique in the ability of generate multiple segmentation hypotheses **directly intensity images** thus avoid computing multiple segmentations altogether.

“... the analysis of image data with hepatocarcinoma-derived cells is a proof of concept rather than a systematic study ...” – Review 3

→ Yes, this individual figure briefly showcase Ultrack’s capabilities. However, the **work as whole** is an in-depth study (Figures 2 to 8, plus the Cell Tracking Challenge submission that is validated by a independent third-party). To our knowledge, given the extent of some of the datasets, this goes beyond any other tracking paper we have seen before, where either they only explore 3D embryos or 2D cells in a dish.

“ ... one could claim that a hierarchical combination of diverse segmentation inputs is more effective and robust in practice than a single, highly optimized solution.” – Reviewer 3

→ We are concerned this comment misses the versatility of our work, for example, if a user has access to highly optimized segmentations, they can provide them to Ultrack, creating a trivial hierarchy and becoming equivalent to a standard ILP tracker.

“For example, one could take top-performing results for different datasets from the Cell Tracking Challenge benchmark and show that Ultrack can outperform all of them when combining their outputs together using ultrametric contours.” – Reviewer 3

→ Most 2D datasets from the Cell Tracking Challenge are almost solved, with several specialized solutions scoring above 0.95 on the TRA metric and often with a minimal difference (0.005 or less) between the leading methods.

Jilly	Bf-C2DL-H5C*	Bf-C2DL-MuSC*	DIC-C2DH-HeLa*	Fluo-C2DL-Huh7	Fluo-C2DL-MSC*	Fluo-C3DH-A549*	Fluo-C3DH-H157*	Fluo-C3DL-MDA231*	Fluo-N2DH-GOWT1*	Fluo-N2DL-HeLa*	Fluo-N3DH-CE*	Fluo-N3DL-CHO*	Fluo-N3DL-DRO	Fluo-N3DL-TRIC	Fluo-N3DL-TRIF	PhC-C2DH-U373*	PhC-C2DL-PSC*	Fluo-C3DH-A549-SIM	Fluo-N2DH-SIM+	Fluo-N3DH-SIM+
OP _{CTB}	0.907 ⁽⁸⁾	0.878 ⁽¹⁾	0.912	0.875 ⁽¹⁾	0.759 ⁽¹⁾	0.938 ⁽²⁾	0.938 ⁽¹⁾	0.797 ⁽¹⁾	0.951 ⁽¹⁾	0.956 ⁽⁵⁾	0.844 ⁽⁶⁾	0.926 ⁽¹⁾	0.708 ⁽⁶⁾	0.867 ⁽²⁾	0.841 ⁽⁶⁾	0.954 ⁽³⁾	0.855 ⁽¹⁾	0.920 ⁽¹⁾	0.905 ⁽¹⁾	0.897 ⁽⁵⁾
	0.906 ⁽⁶⁾	0.875 ⁽⁸⁾	0.909 ⁽²⁾	0.843 ⁽⁵⁾	0.740 ⁽¹⁾	0.931 ⁽¹⁾	0.929 ⁽⁶⁾	0.761	0.945	0.953 ⁽³⁾	0.829 ⁽¹⁾	0.912 ⁽¹⁾	0.617 ⁽²⁾	0.816 ⁽²⁾	0.804 ⁽³⁾	0.951 ⁽¹⁾	0.854 ⁽⁶⁾	0.913	0.903 ⁽⁶⁾	0.865 ⁽⁶⁾
	0.901 ⁽¹⁾	0.870 ⁽⁸⁾	0.904	0.772	0.710 ⁽¹⁾	0.925 ⁽¹⁾	0.925 ⁽²⁾	0.757 ⁽¹⁾	0.941 ⁽⁸⁾	0.953 ⁽¹⁾	0.808	0.911	0.591	0.787 ⁽¹⁾	0.785 ⁽¹⁾	0.951 ⁽²⁾	0.851 ⁽⁸⁾	0.865 ⁽¹⁾	0.896	0.848 ⁽¹⁾
SEG	0.826 ⁽⁸⁾	0.782 ⁽¹⁾	0.873	0.791 ⁽¹⁾	0.655 ⁽¹⁾	0.876 ⁽²⁾	0.889 ⁽²⁾	0.710 ⁽¹⁾	0.931 ⁽³⁾	0.923 ⁽³⁾	0.759 ⁽²⁾	0.917 ⁽¹⁾	0.613 ⁽⁵⁾	0.791 ⁽²⁾	0.746 ⁽⁶⁾	0.924 ⁽³⁾	0.743 ⁽³⁾	0.840 ⁽¹⁾	0.830 ⁽¹⁾	0.820 ⁽⁵⁾
	0.826 ⁽⁸⁾	0.778 ⁽⁸⁾	0.863 ⁽²⁾	0.751 ⁽⁵⁾	0.650	0.863 ⁽¹⁾	0.888 ⁽¹⁾	0.642	0.929 ⁽⁶⁾	0.922 ⁽⁵⁾	0.729 ⁽¹⁾	0.905 ⁽¹⁾	0.567 ⁽²⁾	0.766 ⁽²⁾	0.684 ⁽³⁾	0.923 ⁽¹⁾	0.740 ⁽⁶⁾	0.827	0.829 ⁽⁸⁾	0.759 ⁽⁸⁾
	0.818 ⁽¹⁾	0.774 ⁽⁸⁾	0.853 ⁽²⁾	0.690	0.645 ⁽¹⁾	0.849 ⁽¹⁾	0.878 ⁽⁸⁾	0.632 ⁽¹⁾	0.929 ⁽⁸⁾	0.919 ⁽¹⁾	0.722 ⁽¹⁾	0.902	0.397	0.680 ⁽³⁾	0.654 ⁽³⁾	0.922 ⁽¹⁾	0.738 ⁽⁸⁾	0.730 ⁽¹⁾	0.825 ⁽²⁾	0.746 ⁽¹⁾
TRA	0.988 ⁽⁸⁾	0.974 ⁽¹⁾	0.955	0.960	0.873 ⁽¹⁾	1.000	0.987 ⁽¹⁾	0.884 ⁽¹⁾	0.979	0.993 ⁽¹⁾	0.987 ⁽¹⁾	0.953 ⁽¹⁾	0.802 ⁽⁵⁾	0.952 ⁽¹⁾	0.955 ⁽¹⁾	0.985 ⁽¹⁾	0.968 ⁽¹⁾	1.000	0.979 ⁽¹⁾	0.974 ⁽¹⁾
	0.987 ⁽¹⁾	0.971 ⁽⁸⁾	0.954 ⁽²⁾	0.934 ⁽⁸⁾	0.839 ⁽¹⁾	1.000	0.980	0.882 ⁽¹⁾	0.976 ⁽¹⁾	0.992 ⁽¹⁾	0.979	0.948 ⁽⁶⁾	0.785	0.942 ⁽²⁾	0.936 ⁽⁶⁾	0.982 ⁽¹⁾	0.967 ⁽¹⁾	1.000	0.978 ⁽¹⁾	0.972 ⁽¹⁾
	0.985 ⁽¹⁾	0.971 ⁽¹⁾	0.954	0.865 ⁽¹⁾	0.788	1.000	0.976	0.880	0.967 ⁽⁶⁾	0.991 ⁽¹⁾	0.975	0.935	0.785 ⁽¹⁾	0.854 ⁽⁶⁾	0.886 ⁽¹⁾	0.982 ⁽¹⁾	0.966	1.000	0.977 ⁽¹⁾	0.967

Above are the results where the segmentation is extremely accurate, and most tracking algorithms achieve comparable results. In this case, applying Ultrack would achieve similar results when provided with their segmentation output (trivial segmentation hypotheses as mentioned above). However, this departs from our goals because it falls into the above-mentioned problems, items (i) and (ii) of using a lot of computational and manual resources to reach this point, and again, requiring a lot of expertise to obtain these extremely optimized segmentation models. Additionally, many of the solutions are deep learning models trained from scratch for each dataset with fully annotated ground-truth datasets, an unrealistic scenario for most practitioners, with limited time, data, and resources, and thus out-of-scope for our work.

- Fig. 3: The visualization given in the panel B is difficult to digest and to map to the experiment description given in the main text. It would be more informative to choose one field of view for all crops to recognize better the differences between individual configurations. The purpose of the panel C is unclear, and it does not convey any important information in its current form. The red-green color scheme used in the panel D is unfortunate because it might indicate a range of colors starting from red (the worst value) and going to green (the best value).

→ We initially tried to obtain a single field of view that contained all the individual failure modes. However, we could not find that especially considering some runs are at the 0.93+ accuracy. If we consider the whole frame, the cells were not clearly visible due to their size in proportion to the image.

Panel C is for a qualitative assessment.

We have changed the colors to blue and orange.

Fig. 3 | Ultrack enhances multi-color cell tracking by integrating diverse segmentation algorithms. a, Multi-channel,

- RESULTS (Intensity-based tracking from label-free...): This subsection again contains proof-of-concept results using a single type of data, and thus it is highly questionable whether Ultrack can broadly be applied for such a type of analysis in daily practice when different label-free modalities, such as QPI, phase contrast, DIC, or brightfield, are exploited.

→ The goal of this section is to highlight that Ultrack doesn't need segmentation masks and can be applied to image intensities directly, in this case, **virtual staining (VS)** of cell nuclei and membrane from quantitative phase images. This was repeated in multiple sections where Ultrack uses images' intensities directly, as in the fluorescence imaging datasets of Drosophila and Tribolium from the Cell Tracking Challenge and Figure 6. Therefore, it isn't a proof of concept.

Moreover, extending virtual staining from other label-free modalities is beyond the scope of this paper because we are concerned with the tracking method. Nonetheless, the virtual staining method authors made an online demo available <https://huggingface.co/spaces/compmicro-czb/VirtualStaining>, and the reviewer can verify their generalization performance in other modalities (e.g., brightfield, quantitative phase, and Zernike phase contrast, etc.) for nuclei and membrane prediction (Liu et al., 2024).

Even if this specific method fails, Ultrack is agnostic to the input and can use different methods depending on the user's preference.

- RESULTS (Ultrack achieves state-of-the-art cell tracking...): Based on the results reported in [45] and those listed at <https://celltrackingchallenge.net/participants/AC-6/>, it seems that only a subset of results submitted to the Cell Tracking Challenge benchmark has been reported in this manuscript. For the sake of completeness, it would be informative for readers to know that Ultrack does not excel in all situations evaluated and to understand what the reasons for its lower performance in these situations are.

→ We appreciate the reviewer's concern. As requested, we updated the table and added additional discussion. Our current manuscript, which extends [45], focused on the '**whole embryo 3D datasets**' from CTC. We initially excluded the other two datasets because they are practically 2D datasets embedded in 3D volumes:

- MDA231: cells in a dish (30 pixels in Z);
- TRIC: deformable projection (13 pixels in Z) of the Tribolium, which, in our case, you get superior performance without distortions by tracking the original Tribolium data directly;

Compared to our other submissions, our decreased accuracy on these datasets stems from the fact that in our submission of MDA231 we decided to go for a deep-learning approach since it had ground-truth annotations. However, our training and cross-validation experiment underperformed on the hidden test set, obtaining TRA 0.818 on the hidden test set, while an average of TRA 0.905 ([45] supplementary table) during cross-validation, a more than 0.08 lower score. Thus, we concluded that our UNet predictions failed to generalize, impairing Ultrack's performance.

For the TRIC dataset, the lower performance stems from the distortions of the projections, which violate our spatial consistency assumption. This is discussed in our limitations section and partially addressed by our improved flow-field registration. Moreover, we want to highlight that, two leading solutions in this dataset (MPI and KIT), use the track annotations to learn the flow-field pattern (**supervised learning**), while our approach was fully **unsupervised** (no machine learning fitting or deep learning training) using the image intensities directly.

- DISCUSSION: The limitations of Ultrack are discussed very briefly. It would be informative for the reader to know some guidelines on how to prepare reliable segmentation inputs for Ultrack and how much the performance of Ultrack is affected by different types of segmentation errors committed in individual input segmentation masks.

→ We agree with the reviewer. We updated the discussion to include more limitations and a reference to our documentation, where we discuss how to optimize the performance in different scenarios (e.g. over or under-segmentation, missing links, etc).

Despite its strengths, Ultrack has its limitations. First, the segmentation hypotheses are only as good as the provided foreground and contour maps. While Ultrack is robust to occasional segmentation errors, it may fail when presented with an overwhelming number of incorrect segmentations or systematic errors that persist over time — in such cases, the ILP optimization might converge on erroneous solutions. Additionally, as shown in Figure 5, large cell movements between adjacent frames violate our assumption of segmentation consistency and can impair tracking performance unless registration is used.

A few strategies can improve Ultrack's effectiveness. For instance, since Ultrack uses region-merging hierarchical segmentation, contour maps that initially oversegment cells are actually beneficial, as they generate more hypotheses for the algorithm to consider. Additionally, as with other tracking approaches, frame registration enhances tracking performance. To help users optimize these aspects, we maintain comprehensive documentation, providing guidelines and practical tips for achieving optimal tracking results across different applications.

Other remarks:

- page 1, col. 1, 2nd paragraph: Clarify why segmentation errors are accumulated over time when the majority of existing cell segmentation methods process individual images separately, without considering any temporal context.

→ We agree, the segmentation errors do not accumulate to decrease the total segmentation accuracy as pointed out by the reviewer. What is missing is that they accumulate, decreasing **tracking** accuracy.

As from our excerpt below:

“The primary challenge stems from the accumulation of segmentation errors over time, **hindering long-term tracking**,” – from our submission.

This happens as the segmentation mistakes are compounded at the track level because: (i) The association score of spurious segmentation is detrimental not only to the bad segments themselves but also to linking good segments in adjacent frames; (ii) Single mistakes from missing or bad segments can lead to local mistakes due to the combinatorial nature of matching objects, for example, assuming we have cells (A, B, C, D), and A is missing on frame 1 and D on frame 2, several combinations are possible, (B-A, C-B, D-C) which can be prioritized over the correct (B-B, C-C) matching.

We modified our text to increase clarity that the accumulation of errors happens at the association (tracking) level:

Cell segmentation and tracking have been persistent challenges in bioimage analysis.¹¹ While cell segmentation has advanced rapidly with the advent of deep-learning-based methods,¹⁸⁻²⁰ cell tracking continues to be an open problem.¹¹ The primary challenge stems from the **association of spurious segmentations** over time, hindering long-term tracking, especially in dense and dynamic cellular environments. Most automatic tracking methods employ a two-step approach: first segmenting cells, then linking them across time frames.²¹⁻²⁴ While computationally efficient, this approach struggles with the compounding of errors over time, particularly in dense tissues²⁵ or when cells divide rapidly.

- page 1, col. 1, 3rd paragraph: It is not always true that cells cannot merge over time. There can be other biological constraints, such as cell fusion in plant cells, not covered in this work. Can Ultrack handle such situations?

→ No, Ultrack in its current form cannot, but this limitation is easily addressable, as the ILP formulation can be adapted to include cell fusion. However, this point is more of an exception rather than a rule, as other rare biological phenomena, such as division into three or more cells.

- RESULTS (Optimal parameter selection...): Obviously not all parametrizations of gamma were considered (only four of them). Please rephrase. Furthermore, clarify what "manually curated ground truth" refers to and whether the two publicly available datasets refer to the pair of available training sequences.

→ We clarified in the revised manuscript that ground truth refers to the publicly available data for training. As the reviewer pointed out, we did not evaluate all real numbers (as there are an infinite number of them), only four (0.1, 0.25, 0.5, 1.0), so we replaced "All" with "All four."

- page 7, col. 2, 2nd paragraph: "The cell centroids of these tracks" -> "The centroids of the tracked cells" or alike

→ Fixed, we appreciate the correction.

- page 8, col. 2, 4th paragraph: It seems that the test datasets from the Cell Tracking Challenge dataset repository are freely available, but without reference annotations. Please rephrase.

→ We include "test sets **annotations**" to disambiguate the phrase.

- page 8, col. 2, last paragraph: It seems that some segmentation annotations are publicly available for the fly and beetle datasets. Please clarify why these cannot be used to train segmentation models.

→ These are small subsets of randomly located 2D annotations. They can be used but don't bring much performance for 3D deep learning models. That's why competing deep-learning-based works on these datasets are mainly working at the centroid level – not segments.

- Table 1: The superscripts attached to individual scores are undefined.

→ We included a supplementary table (Table 3, last page of the manuscript) describing the individual algorithms for each subscript. We provide a copy below:

Label	Reference	Description
KTH	104	Uses the Gaussian Mixture Probability Hypothesis Density (GM-PHD) ¹⁰⁵ filter to detect and estimate the motions of centroids with the Viterbi algorithm to optimally link segmentations across frames while assigning different cell states (e.g. migration, mitosis, appearance, disappearance).
KIT	23	EmbedTrack combines pixel-embedding segmentation with motion estimation for joint segmentation and tracking by embedding each pixel in a centroid-offset space and incorporating optical flow.
MPI	(unpublished)	Although a detailed description is undisclosed, the source code provided by the challenge organizers suggests a deep-learning-based centroid detection strategy followed by nearest-neighbor linking.
JAN	106, 24	Builds upon Linajea , ²⁴ which predicts cell locations as Gaussians' maxima and their motion offset to their position on the previous frame, by classifying cell states (parent, daughter, continuing cells), improving their ILP formulation to predict these states, while asserting that parent cells can only link to daughter cells. A structured SVM is used to learn optimal ILP weights, enhancing automation linking cell trajectories.
LEID	107	Joint segmentation and tracking by model evolution with level sets and specialized routines for cell division and cell appearances.
MU	108	Integrates H-minima watershed segmentation with a U-Net model for cell contour and foreground prediction. Subsequent tracking uses an intersection-based heuristic, which pairs cells across frames if they share sufficient overlap.
IGFL	22	ELEPHANT leverages deep-learning-based cell detection, coupled with predicted flow fields (i.e. motion offset) for cell advection and linking. This flow-driven approach resembles the MPI and JAN strategy, using learned motion information to guide inter-frame associations.
RWTH	109	Detects cells via a Laplacian of Gaussian filter and applies nearest-neighbor linking in subsequent frames; detections are then used as seeds for segmentation. Manual curation post-processing is performed to rectify missed detections.

Table 3 | Description of leading methods in the Cell Tracking Challenge, scores shown at Table 1.

- page 19, col. 2, 1st paragraph: Specify the neighborhood used and how it is guaranteed that extracted contours of individual segments are closed surfaces in 3D.

→ We updated the text to indicate they are the immediate neighbors, 4-neighbors for 2D and 6-neighbors for 3D. The surfaces (contours in 2D images) are closed by definition because the algorithm operates by merging regions – and each region has their corresponding closed surface (contour in 2D).

- Equation 12: "...between the predicted and gt instances is at least half..." -> "...between the predicted and gt instances is more than half..."

→ Fixed, thank you.

Ultrack: pushing the limits of cell tracking across biological scales

Reviewer #1:

None

Reviewer #2:

I appreciate the authors' efforts in addressing the suggestions made during the review process. The manuscript has been significantly improved, and the potential utility of this tool within the field is now well articulated.

→ Thank you!

It appears that the changes discussed for Figure 5d have not been incorporated into the revised manuscript. Specifically, the manuscript would greatly benefit from additional benchmarking information on the tool's performance under varying temporal resolutions (as shown in changes of Figure 5d) and z-resolutions. Recording neural dynamics often requires high temporal resolution, which typically necessitates compromises in z-resolution. While the four preprint papers cited in the manuscript demonstrate fine z-sampling, many applications frequently face a trade-off between temporal and z-resolution, which can complicate 3D segmentation and tracking. I recommend that the authors provide benchmarks on the tool's performance under these compromised low z-resolution conditions. Such benchmarks would be extremely valuable for potential users, as they would help clarify the tool's robustness and limitations in real-world scenarios.

The editors have informed us that it is not strictly necessary to address this point, and we concur. The manuscript is already quite long, and in our opinion, there are not many deep insights that can be gathered from such an analysis: the results can be expected, as the axial resolution decreases, the image quality and, therefore, the tracking quality will also decrease. A more interesting question is how far Ultrack retains its edge compared to competing approaches as the axial resolution deteriorates. Conveniently, this was independently evaluated against other algorithms in the WormID benchmark (Adhinarta et al. bioRxiv 2025), which has lower and variable z-resolution. We have added a short mention of that in the discussion:

temporal resolution datasets.

Concurrent work has shown that Ultrack remains effective even in scenarios that prioritize speed over resolution, such as neural dynamics imaging, where z-resolution is often reduced to enable faster acquisition. The WormID benchmark⁶⁶ demonstrates Ultrack’s robustness across a range of imaging resolutions, from $1 \times 0.1604 \times 0.1604$ to $1.5 \times 0.32 \times 0.32 \mu\text{m}/\text{voxel}$.⁶⁷

All these features — including its ability to handle mul-

For reference, here is the table in the paper showing that Ultrack has higher performance despite the substantially lower axial resolution compared to datasets in our manuscript:

Method	$d_{\text{th}}=3\mu\text{m}$		$d_{\text{th}}=6\mu\text{m}$	
	DET	TRA	DET	TRA
Ultrack Bragantini et al. (2023)	0.70±0.04	0.66±0.05	0.81±0.05	0.78±0.05
3DeeCellTracker Wen et al. (2021)	0.28±0.04	0.26±0.03	0.34±0.05	0.32±0.04
3DeeCellTracker [†] Wen et al. (2021)	0.43±0.07	0.40±0.07	0.51±0.09	0.47±0.08

Remarks on code availability:

The code has a clear README, installation instructions, and usage examples. It is easy to install and run locally and is well-documented.

→ Thank you!

Reviewer #3:

Remarks to the Author:

The authors have adequately addressed the comments and concerns raised by the reviewers. This reviewer does not have any additional remarks to the revised manuscript.

→ Thank you!